# A haplotype-resolved pangenome of the barley wild relative *Hordeum bulbosum*

Jia-Wu Feng[1], Hélène Pidon[1,2,3], Maria Cuacos[1], Thomas Lux[4], Axel Himmelbach[1], Reza Haghi[1], Jörg Fuchs[1], Georg Haberer[4], Yi-Tzu Kuo[1], Yu Guo[1], Murukarthick Jayakodi[1,5,6], Helena Toegelová[7], Dörte Harpke[1], Manuela Knauft[1], Anne Fiebig[1], Maren Maruschewski[1], Moshe Ronen[8], Amir Sharon[8,9], Hana Šimková[7], Klaus F. X. Mayer[4,10], Manuel Spannagl[4,11], Jochen Kumlehn[1], Stefan Heckmann[1], Andreas Houben[1], Frank R. Blattner[1 ✉], Nils Stein[1,12 ✉] & Martin Mascher[1,11,13 ✉]

Wild plants can contribute valuable genes to their domesticated relatives[1]. Fertility barriers and a lack of genomic resources have hindered the effective use of crop–wild introgressions. Decades of research into barley's closest wild relative, *Hordeum bulbosum*, a grass native to the Mediterranean basin and Western Asia, have yet to manifest themselves in the release of a cultivar bearing alien genes[2]. Here we construct a pangenome of bulbous barley comprising 10 phased genome sequence assemblies amounting to 32 distinct haplotypes. Autotetraploid cytotypes, among which the donors of resistance-conferring introgressions are found, arose at least twice, and are connected among each other and to diploid forms through gene flow. The differential amplification of transposable elements after barley and *H. bulbosum* diverged from each other is responsible for genome size differences between them. We illustrate the translational value of our resource by mapping non-host resistance to a viral pathogen to a structurally diverse multigene cluster that has been implicated in diverse immune responses in wheat and barley.

Wild relatives of cultivated plants are important sources of genetic diversity for crop improvement[3]. A common means of transferring genes from wild into domesticated plants is introgression lines (ILs) that are derived from crop–wild crosses and bear a small proportion of the wild parent's genes in a cultivated genomic background[4–6]. However, developing 'alien' introgressions into elite varieties is challenging in many crops. A case in point is barley (*Hordeum vulgare*). Barley's 'secondary gene pool'[7]—those wild species that are amenable to gene transfer through meiotic recombination—is narrower than that of wheat, another grain crop and fellow member of the Triticeae tribe of cereal grasses, in which ILs have been more widely deployed[8,9]. There are more than 30 species in the genus *Hordeum*[10], all of which might be considered candidate donors of valuable alleles. Yet only one yields fertile progeny when crossed with barley: bulbous barley (*Hordeum bulbosum*). Although *H. bulbosum* is also barley's closest wild relative with an estimated divergence time of 4.5 million years[10], their genetics are quite different. Bulbous barley is a perennial outcrosser with either a diploid or an autotetraploid genome[11,12]. By contrast, barley breeding has profited from the tractable genetics of a diploid annual inbreeder. Crop–wild introgressions have yet to meet with agronomic success in barley. Wide crosses yielded ILs in which the distal regions of any single chromosome have been replaced with their homeologues in bulbous barley[13,14]. Resistances to numerous diseases are common in this germplasm, but so are yield penalties[15] owing to the co-transfer with beneficial alleles of harmful variants. To eliminate the latter, geneticists and breeders characterize ILs with molecular markers and select rare recombinants to fine-map resistance loci[16,17]. Pangenomes, collections of genome sequences of multiple individuals of a species or higher taxon[18], are taking over the role of single reference genomes as anchors for assessing genetic diversity, genetic mapping and comparative genomics. Complementing the pangenome of domesticated barley and its wild progenitor *H. vulgare* subsp. *spontaneum*[19], here we report a haplotype-resolved pangenome of the barley wild relative *H. bulbosum* and illustrate its applications in evolutionary research and trait mapping.

## A diploid reference genome

We first attempted the sequence assembly of a diploid *H. bulbosum* individual, FB19-011-3, grown from seed collected close to Andirrio in western Greece. The contiguity and completeness of the contigs constructed from PacBio HiFi reads[20] (Supplementary Tables 1 and 2)

[1]Leibniz Institute of Plant Genetics and Crop Plant Research (IPK) Gatersleben, Seeland, Germany. [2]Julius Kühn Institute (JKI) – Federal Research Centre for Cultivated Plants, Institute for Resistance Research and Stress Tolerance, Quedlinburg, Germany. [3]UMR AGAP Institut, Univ. Montpellier, CIRAD, INRAE, Institut Agro, Montpellier, France. [4]PGSB – Plant Genome and Systems Biology, Helmholtz Center Munich – German Research Center for Environmental Health, Neuherberg, Germany. [5]Texas A&M AgriLife Research Center at Dallas, Texas A&M University System, Dallas, TX, USA. [6]Department of Soil & Crop Sciences, Texas A&M University, College Station, TX, USA. [7]Institute of Experimental Botany of the Czech Academy of Sciences, Centre of Plant Structural and Functional Genomics, Olomouc, Czech Republic. [8]The Institute for Cereal Crops Research, Tel Aviv University, Tel Aviv, Israel. [9]School of Plant Sciences and Food Security, Tel Aviv University, Tel Aviv, Israel. [10]School of Life Sciences, Technical University Munich, Freising, Germany. [11]Centre for Crop & Food Innovation, Food Futures Institute, Murdoch University, Murdoch, Western Australia, Australia. [12]Crop Plant Genetics, Institute of Agricultural and Nutritional Sciences, Martin Luther University Halle-Wittenberg, Halle–Saale, Germany. [13]German Centre for Integrative Biodiversity Research (iDiv) Halle-Jena-Leipzig, Leipzig, Germany. ✉e-mail: blattner@ipk-gatersleben.de; stein@ipk-gatersleben.de; mascher@ipk-gatersleben.de

were promising: the assembled contigs amounted to 6.78 gigabases (Gb) of sequence, and the contig N50 was 9.82 Mb. The good separation between the haplotypes is reflected by a preponderance of unique 71-base oligonucleotides (71-mers) over duplicated ones in *k*-mer profiles of the HiFi reads (Supplementary Fig. 1). Chromosome conformation capture sequencing (Hi-C) data[21] were used to assign contigs to haplotypes (Methods and Extended Data Fig. 1a), which were then arranged into chromosomal pseudomolecules with the help of physical linkage information provided by Hi-C data and a Bionano genome map (Supplementary Tables 3–5). The resultant pseudomolecules were longer than their constituent contigs because there was a large identity-by-descent (IBD) tract spanning the centromere of chromosome 4H (Extended Data Fig. 1b). Both haplotypes were highly co-linear to the barley genome (Supplementary Fig. 2). Visual inspection of the Hi-C contact matrices supported the absence of phase switches or other misassemblies (Fig. 1a). To confirm the structural integrity of our assembly, we genotyped single pollen nuclei of FB19-011-3 through Illumina short-read sequencing[22] (Fig. 1b and Supplementary Table 6) and called single nucleotide polymorphisms (SNPs). In the resultant variant matrix, we looked for patterns consistent with phase switches. A total of 60,934 neighbouring SNP pairs (1.01%) had recombination fractions exceeding 50%, indicating that they were potentially located in wrongly phased regions (Extended Data Fig. 1c and Supplementary Figs. 3 and 4). The genetic map constructed from single-pollen-nucleus sequencing data was co-linear with the genome sequence (Supplementary Fig. 5).

Pericentric recombination deserts common to all Triticeae extended further into distal regions in *H. bulbosum* than in barley (Supplementary Fig. 5). This is confirmed by cytological observations: extra-terminal chiasmata (that is, those where homologous chromosomes were distally connected only by a thin thread of chromatin during meiotic metaphase I) were twice as frequent in *H. bulbosum* as in barley (Extended Data Fig. 2). To validate our haplotype phasing, we designed oligonucleotide probes[23] for single-copy regions on the long arm of chromosome 5H that are unique to either haplotype (Extended Data Fig. 3a,b). Fluorescence in situ hybridization (FISH) showed the expected red and green signals marking both 5H homologues without interspersed or mixed signals (Fig. 1c). Hybridization of the probes to meiotic chromosomes of FB19-011-3 and mitotic chromosomes of selfed offspring revealed crossover events (Extended Data Fig. 3c,d). Taken together, co-linearity to barley, Hi-C signals, genetic mapping and FISH supported the accuracy of the haplotype-resolved assembly of FB19-011-3.

## Assembly of diverse *H. bulbosum* genomes

A pangenome of bulbous barley would be incomplete if it did not include tetraploid genotypes. We had phased a diploid sequence assembly essentially by a principal component analysis (PCA) on the Hi-C data and expected that this approach should work for higher ploidies as well. We attempted the assembly of a tetraploid individual, FB19-028-3 (Methods and Extended Data Fig. 1d). The contig-level N50 (7.7 Mb) was lower than in the diploid. IBD haplotypes, but no misassemblies, were discernible in Hi-C contact matrices (Fig. 1d). No phase switches were seen in four-colour FISH with haplotype-specific painting probes on chromosome 5H (Fig. 1e and Extended Data Fig. 3e).

To represent diversity of *H. bulbosum*, we selected another three diploid and five tetraploid accessions for genome sequencing, resulting in a total of 32 haplotypes (Supplementary Figs. 6–13 and Supplementary Table 9). The selection was based on a PCA on genotyping-by-sequencing data for 260 individuals (Extended Data Fig. 4a). The PCA indicated that geographical origin and ploidy are the main drivers of differentiation between genomes of *H. bulbosum* (Extended Data Fig. 4b,c). The selected genotypes span the geographic range of the species from Central Asia to the Western Mediterranean (Extended Data Fig. 4d,e). In addition, we sequenced the genomes of three genotypes—A17, A40 and A42—that have served as donors to ILs. The assembly metrics of

diploid and tetraploid genotypes were comparable: initial unphased assemblies amounted to between 6.9 Gb and 14.2 Gb and had N50 values between 6.4 Mb and 17.0 Mb (Supplementary Table 8). In the reads of all genomes, unique 71-mers dominated *k*-mer profiles, but IBD between haplotypes was more pronounced in some genomes than in others (Supplementary Figs. 1, 14 and 15). Locally high similarity or pairwise identity of haplotypes was accompanied, respectively, by lower assembly contiguity or fewer than otherwise the four haplotypes. Nevertheless, the Hi-C contact matrices enabled us to assign the contigs of the primary assemblies to one or more haplotypes (Extended Data Fig. 5). Diploid assemblies were more comprehensive and contiguous than their tetraploid counterparts. The average size of the diploid assemblies was 7.4 Gb. Between 88.1% and 97.7% of the contigs of the primary assemblies were incorporated into pseudomolecules and partitioned into haplotypes (Supplementary Table 9). In addition, ten complete chloroplast genomes were assembled (Supplementary Table 8).

To assess the gene content in the *H. bulbosum* pangenome, we used a multi-tier gene prediction strategy (Methods). The number of gene models annotated per genotype identified ranges from 107,037 for the diploid PI365428 to 238,246 for the tetraploid FB19-028-3. We benchmarked the quality of the gene predictions against the BUSCO Poales dataset. On average, we found 95.5% as complete copies, 0.4% as fragmented copies and 4.2% as missing genes (Supplementary Table 10).

Gene order in the 32 haplotypes was conserved. Large (>2 Mb) inversions were evident from the whole-genome alignments, but we did not observe any interchromosomal translocations (Fig. 2a). The largest event that we detected was a 243-Mb inversion unique to haplotype 2 of FB19-028-3 (Extended Data Fig. 6a). Distal inversions between the haplotypes of FB19-011-3 corresponded to regions of suppressed recombination revealed by pollen sequencing (Extended Data Fig. 6b,c). We discovered a total of 221,326,255 SNPs and 8,926,931 indels across the 32 haplotypes (Extended Data Fig. 6d). A reference-free assessment of sequence diversity based on *k*-mers revealed higher diversity in *H. bulbosum* than in *H. vulgare* subsp. *spontaneum*, the wild progenitor of domesticated barley (Extended Data Fig. 7a).

## A haplotype-resolved pangenome

To characterize the complexity of the *H. bulbosum* pangenome without relying on a single reference genome, we explored three different approaches. In the first approach, we constructed a single-copy pangenome following the approach of ref. 24. We defined two sets of single-copy sequences: those that are unique at the haplotype level (that is, occurring in possibly up to four times in a tetraploid genome) and those that are unique at the genotype level (thus, occurring in only one haplotype). The haplotype-level single-copy pangenome was more complex than that of barley because the content of unique sequences grew faster as more sequences were added (Fig. 2b). We divided both the haplotype-level and genotype-level single-copy pangenome into core (present in all individuals) and variable (absent in some individuals) compartments (Fig. 2c and Extended Data Fig. 7b,c). Sequences that are variable at the haplotype level but core at the genotype level were more abundant in autotetraploids than in diploids, supporting the notion that autotetraploids tolerate higher levels of presence–absence variation between haplotypes (Fig. 2d).

In the second approach, we constructed whole-genome graphs using the Minigraph-Cactus pipeline (Extended Data Fig. 7d,e and Supplementary Figs. 16–23). Compared to the graph for the single-copy pangenome, the pangenome graph was more complex. The proportion of core sequence identification is much smaller than that of the single-copy pangenome (Extended Data Fig. 7c,e). Although more variants are discovered in the graph genomes, their interpretation remains challenging in complex plant genomes[25]. To address this, we simplified the application by constructing a graph for both haplotypes of the highly heterozygous diploid individual FB19-011-3. The graph incorporated

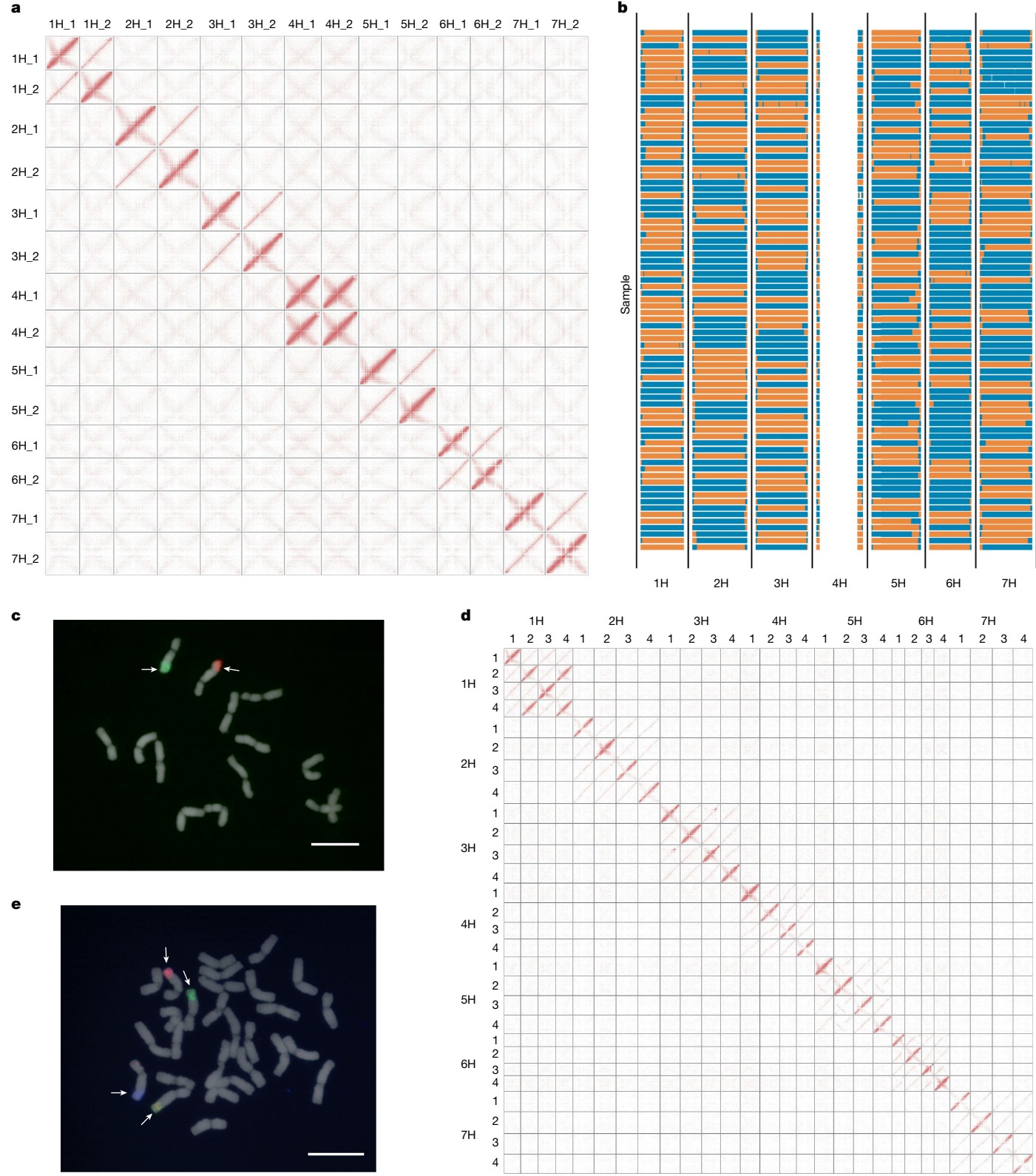

**Fig. 1 | Haplotype-resolved assembly of diploid and tetraploid *H. bulbosum* genomes. a**, Diploid Hi-C contact matrix of the FB19-011-3 genome assembly. **b**, Graphical genotypes constructed from single-nucleus sequence data of FB19-011-3 pollen. Blue colour, haplotype 1; orange colour, haplotype 2. **c**, Haplotype-specific oligonucleotide-based FISH in mitotic chromosomes of FB19-011-3. The red and green probes target haplotypes 1 and 2, respectively, of chromosome 5H. At least two independent experiments were carried out to confirm the reproducibility of the labelling patterns. Scale bar, 10 µm. **d**, Hi-C contact matrix of the sequence assembly of the tetraploid clone FB19-028-3. **e**, Haplotype-specific oligonucleotide-based FISH in mitotic chromosomes of FB19-028-3. The red, green, blue and yellow probes target haplotypes 1, 2, 3 and 4, respectively, of chromosome 5H. At least two independent experiments were carried out to confirm the reproducibility of the labelling patterns. Scale bar, 10 µm.

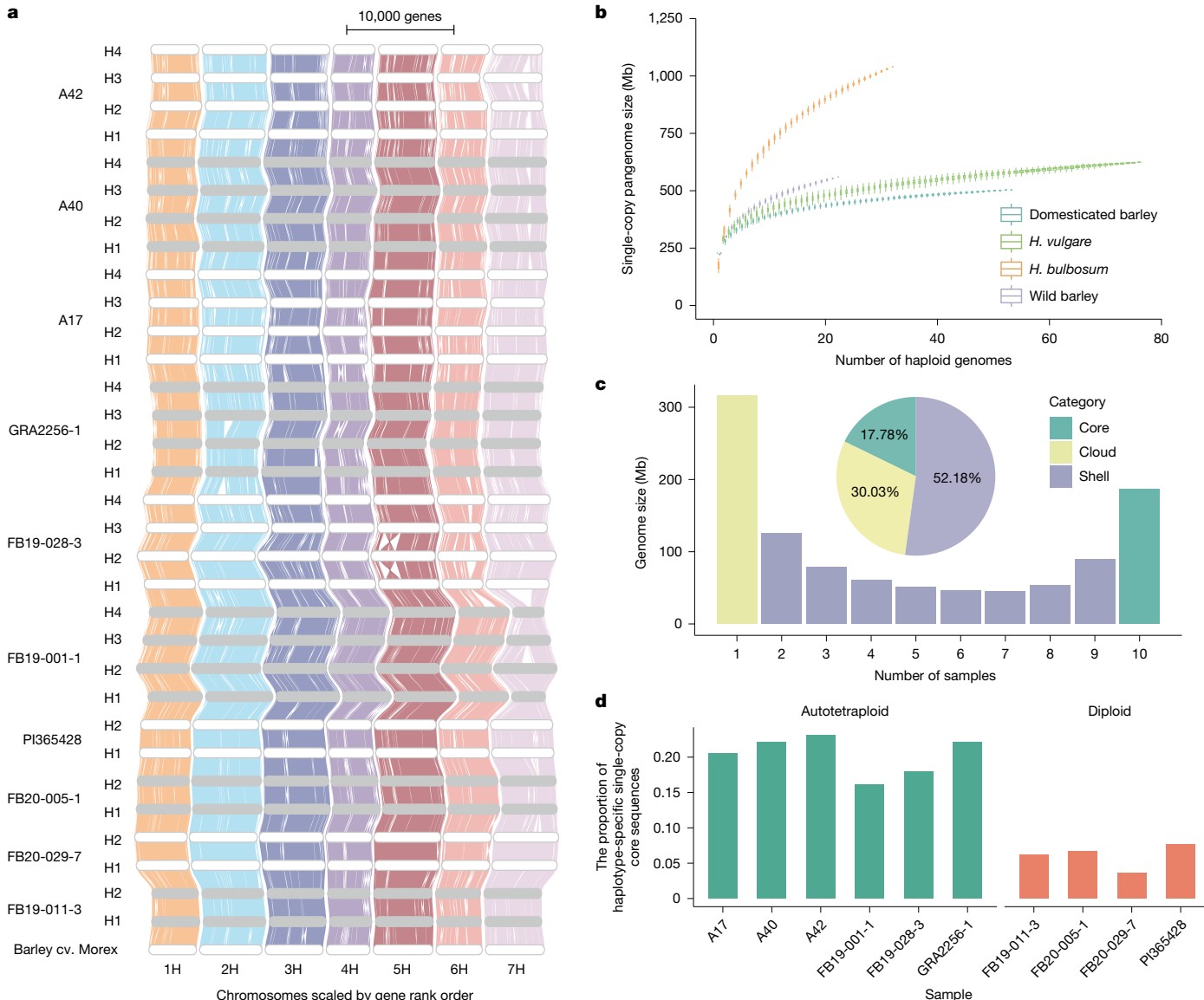

**Fig. 2 | A pangenome of 32 haplotypes from diploid and tetraploid *H. bulbosum* clones. a**, GENESPACE synteny plot between 32 *H. bulbosum* haplotypes. Grey bars represent chromosomes and are scaled according to the number of genes with syntenic alignments to other genomes. H1–H4 denote haplotypes 1–4. **b**, Pangenome complexity estimated by single-copy *k*-mers. The curves trace the growth of non-redundant single-copy sequences as sample size increases. Error bars were derived from 100 ordered permutations each. The central line represents the median; the box spans the interquartile range, from the first quartile to the third quartile and whiskers extend to the most extreme values within 1.5× the interquartile range from the quartiles. **c**, Composition of core, shell and cloud single-copy sequences in accession-level pangenomes. **d**, Summary of haplotype-specific single-copy core sequences for each accession. These core sequences were assigned to chromosomal locations, but not present in all haplotypes within each accession.

haplotype-specific structural variants that we genotyped by mapping our pollen sequencing reads to the graph genome (Supplementary Fig. 3).

Our third approach for pangenome analysis was gene centric. We constructed an orthologous framework for the gene-based *H. bulbosum* pangenome at the genotype level. We identified a total of 112,327 orthologous groups, of which 28,074 (954,739 genes, 55.67%) contained at least one orthologous gene from all compared genotypes (core genome). A further 67,568 orthologous groups (501,602 genes, 29.25%) were found with genes not present in one or several genotypes (shell genome). Finally, 258,525 (15.07%) genes were found in only a single genome (cloud genes; Extended Data Fig. 7f,g).

## Genome size evolution

Studies of structural evolution in the repeat-rich genomes of *H. bulbosum* and *H. vulgare* should look beyond genes and other low-copy sequences. We annotated transposable elements (TEs) and other classes of repetitive elements in both our *H. bulbosum* assemblies and a long-read assembly of the barley genome. TEs make up a large proportion of the barley genome[26], as they do in *H. bulbosum* (Extended Data Fig. 8a and Supplementary Table 11). The genome of *H. bulbosum* is about 80% the size of the barley genome, and much of this difference is attributable to a difference in overall TE content (3.70 Gb in barley cv. Morex, 3.14 Gb in *H. bulbosum* FB19-011-3). The predominant class of TEs in both genomes is long terminal repeat retrotransposons of the *Copia* and *Gypsy* superfamilies (Fig. 3a). However, TE profiles differ locally, as do the relative rates of genome size expansion, as was revealed by a generalized additive model fitted to chromosome-scale alignments (Fig. 3b and Supplementary Fig. 24). Although the overall genome size of *H. bulbosum* is smaller, its distal regions are relatively longer than those of *H. vulgare* (Fig. 3c), primarily owing to more frequent insertions of *Gypsy* elements. The expansion of the *H. vulgare*

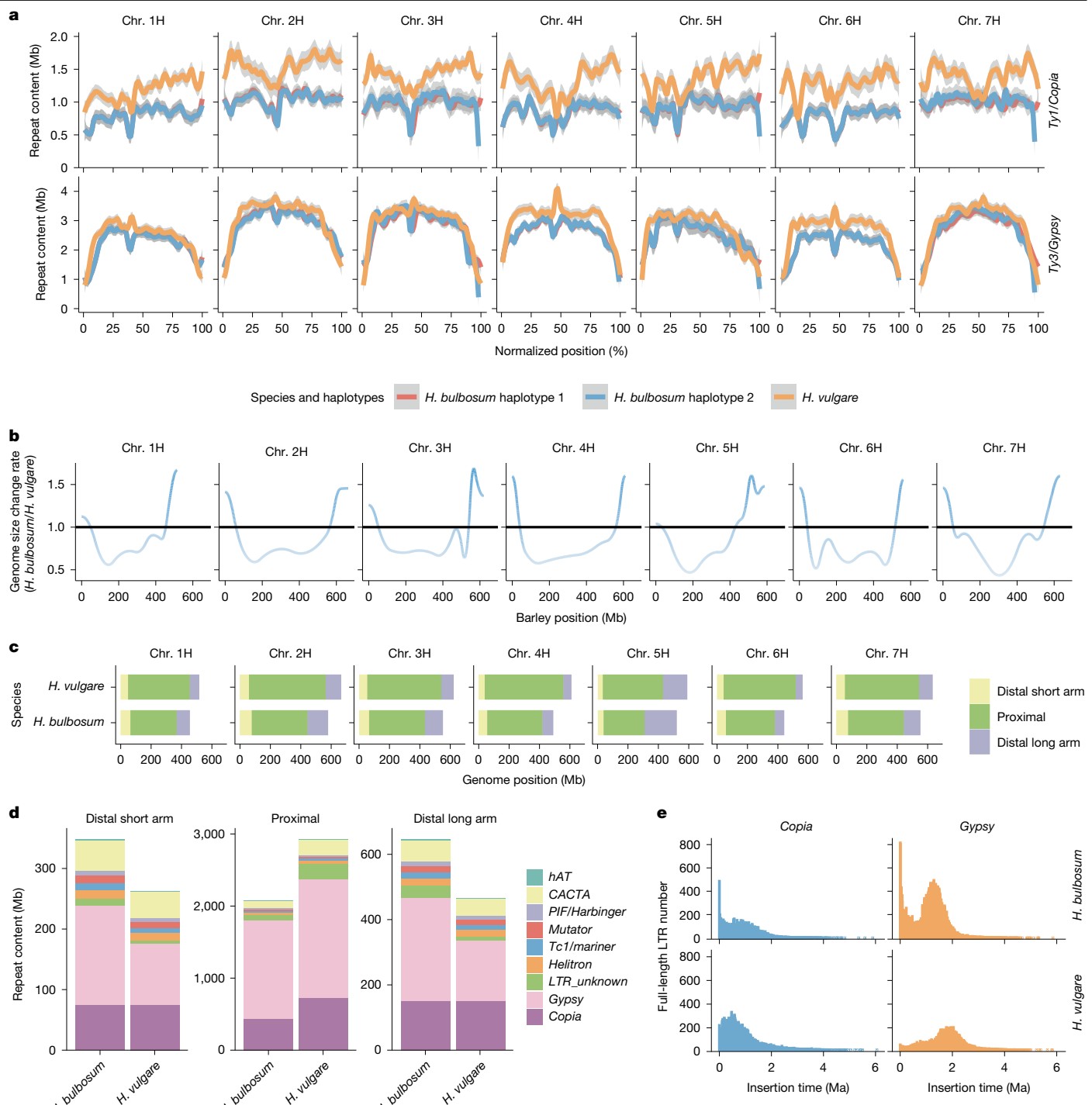

**Fig. 3 | Evolution of TEs and genome size in *H. bulbosum* and *H. vulgare*.**
**a**, Absolute content of *Copia* and *Gypsy* elements along the genomes of both species. Each sequence assembly was divided into 100 bins of equal size. The grey shading indicates the 95% confidence interval. **b**, Local rates of genome size change between *H. vulgare* and *H. bulbosum*. **c**, Schema showing the partitioning into regions where either the *H. bulbosum* or the *H. vulgare* genome is locally expanded. **d**, Repeat content in these partitions. **e**, Distribution of insertion times of full-length *Copia* and *Gypsy* elements.

genome, by contrast, is not attributable to any single class of TE in proximal regions (Fig. 3d). As those TE bursts that are still discernible in today's genomes all occurred after the divergence of barley and *H. bulbosum* (Fig. 3e), sequences conserved between both genomes are limited mainly to genes and their regulatory sequences. The same TE families have colonized the genomes of both species, but which of their members inserted where and when sets the species apart (Extended Data Fig. 8b,c). *BARE-1* elements, a family of *Copia* long terminal repeat retrotransposons[27], make up 493 Mb of the Morex

genome, whereas its closest homologues in FB19-011-3 account for only 302 Mb of sequence (Extended Data Fig. 8c,d). Vice versa, *Sabrina*, a family of *Gypsy* elements, amounts to 501 Mb in *H. bulbosum*, but only 420 Mb in *H. vulgare* (Extended Data Fig. 8c). These differences in overall sequence content are reflected by differences in the timing and intensities of bursts of *BARE-1* and *Sabrina* (Extended Data Fig. 8e,f). Although smaller in size, the genome of *H. bulbosum* has more full-length long terminal repeat elements (61,916) than that of *H. vulgare* (46,727). This observation is in line with a higher number

of TE insertions, mainly of *Gypsy* elements, in proximal regions of the *H. bulbosum* genome in the past 300,000 years (Fig. 3e and Extended Data Fig. 8g). These relative differences between the two genomes should not overshadow the fact that, in both species, TE insertions occurred most frequently in distal regions, where young elements predominate (Extended Data Fig. 8g,h).

## Autotetraploids arose at least twice

*H. bulbosum* is peculiar as this taxon is composed of both diploid and tetraploid members. It is expected that the latter traces back to the former and that both groups are reproductively isolated. Yet, despite different chromosome numbers, genome sizes and clear genetic differentiation (Extended Data Fig. 4b,c and Supplementary Table 8), diploid and tetraploid plants are morphologically indistinguishable[11]. It is pertinent to question how often tetraploids have arisen, how they are related to diploids, and whether there is gene flow between them. To answer these questions, we studied the divergence of our 32 pangenome haplotypes and 263 lineages genotyped by reduced representation sequencing (Extended Data Fig. 4b,c and Supplementary Table 12). We constructed phylogenetic trees from nuclear and chloroplast genes (Fig. 4a and Extended Data Fig. 9a–c), assigned them to four ancestral populations (Fig. 4a,e), inferred past population size trajectories by pairwise comparisons of haplotypes (Fig. 4b–d and Extended Data Fig. 9d) and searched for shared ancestral haplotype groups[28] (AHGs; Fig. 4f and Extended Data Fig. 10).

Population size trajectories also provide information on divergence times: if the ancestral populations to which two haplotypes are assigned differed in size sometime in the past, those two populations must have split before that point in time[29]. PI365428, a diploid accession from Libya, diverged earlier from all other genotypes than these did from each other (Fig. 4a and Extended Data Fig. 9a–e). Its population seems to have undergone a recent collapse (Fig. 4b), probably owing to the desertification of the Sahara starting around 10,000 years ago[30]. PI365428 represents an early diverged diploid population (Extended Data Fig. 9a–f). It may be a relic of an early diploid population in the eastern Mediterranean or North Africa, reaching back as far as 2 million years ago (Ma; Extended Data Fig. 9d). It should be noted that this estimate comes with some uncertainty surrounding mutation rates and generation times in a perennial plant. However, it is consistent with its divergence 4.5 Ma from barley (Supplementary Fig. 25). FB19-001-1 from the Peloponnese in Greece was an exception. Its lineage separated from Greek diploids so recently that their population size trajectories are largely overlapping (Fig. 4c). By contrast, GRA2256-1, a tetraploid from Tajikistan, separated from the Greek diploid lineage around 1.5 Ma (Fig. 4d). Furthermore, Greek tetraploids are genetically closer to neighbouring diploids than to other tetraploid populations (Figs. 2b and 4a and Extended Data Fig. 9a–c), and their genomes resemble those of Greek diploids in their TE profiles (Extended Data Fig. 8b). Taken together, these observations led us to conclude that FB19-001-1 is a member of a tetraploid lineage that might have arisen later than the Asian tetraploids, possibly in Greece. By contrast, GRA2256-1 is an unadmixed representative of an early polyploid lineage. Admixture between both lineages occurs in Greece, where ancestry from the younger lineage is concentrated in the Peloponnese and the western part of the mainland (Fig. 4e and Extended Data Fig. 9h). Our hypothesis of multiple origins of tetraploid *H. bulbosum* was supported by divergence patterns between haplotypes. The SNP-based distance between older tetraploid lineages peaked at 10 variants per kilobase corresponding to 2.23 Ma, which is close to the inferred split time (Fig. 4d and Extended Data Fig. 10a). AHGs[28] were in concordance with population structure. For example, in FB19-028-3, an admixed tetraploid, 20.14% of the genome was assigned to an AHG prevalent in older tetraploids and 66.2% to one common in diploids in the western Mediterranean basin.

By contrast, FB19-001-1 had larger contributions from the latter AHG (84.4%) and only 5.44% from the tetraploid AHG (Fig. 4f and Extended Data Fig. 10b). All of this supports the notion that FB19-001-1 is not a chance finding, but is a representative example of interconnectedness of diploid and tetraploid populations. Whereas present distribution ranges restrict admixture between cytotypes to Greece, where the Pindos mountain range[11] forms the border between diploid and tetraploid cytotypes, tetraploidizations in evolutionarily recent times followed by hybridization with older tetraploids may have occurred often enough to leave signals that can be detected by population genetic analysis[12].

## Characterizing crop–wild introgressions

Investigations of tetraploid bulbous barley have implications for plant breeding. Crosses between diploid or tetraploidized barley and tetraploid *H. bulbosum* are the most commonly used method of producing ILs[15]. Genome sequences help to confirm the identity of ILs and delineate their boundaries. We updated a previous introgression atlas with the positional information obtained from the latest genome assemblies of barley[31] and *H. bulbosum* (Fig. 5a,b, Extended Data Fig. 11a, Supplementary Table 13 and Supplementary Fig. 26). We found that, cumulatively, 1,085 Mb of the *H. bulbosum* genome have been transferred into the barley background at least once, with a cumulative 464 Mb being represented in at least two ILs (Fig. 5a and Supplementary Table 13). Owing to the fact that both recombination and resistance gene homologues occur preferentially in distal regions, the regions of the *H. bulbosum* genome that were transferred at least twice into barley harbour 65% of resistance gene homologues of nucleotide-binding leucine-rich repeat receptors (NLRs; Fig. 5c and Extended Data Fig. 11b). This means that in principle a large fraction of *H. bulbosum* resistance gene diversity can be transferred into barley. The provenance of the donors A17, A40 and A42 has been uncertain. Our genotyping-by-sequencing data showed that these genotypes are genetically associated with Western Asian populations of the older tetraploid lineage (Extended Data Fig. 11c). Long IBD blocks show that A40 and A42 share a recent common ancestor (Extended Data Fig. 10c).

In addition, we constructed contig-level assemblies for ten ILs from PacBio HiFi reads (Supplementary Figs. 27–31 and Supplementary Table 14). We predicted origins of haplotypes for those ILs that were derived from A17, one of our pangenome accessions (Fig. 5b and Supplementary Table 13). In the IL JKI-5215, a backcross line in the background of barley cv. Igri, we found a 32.1-Mb A17-derived segment on the long arm of chromosome 3H (Fig. 5d). This segment was a mosaic of two haplotypes: the proximal 6.6 Mb match haplotype 3, whereas the distal 25.5 Mb match haplotype 4. Inspection of Hi-C matrices ruled out a misassembly in A17 (Fig. 5e,f and Supplementary Fig. 32). A crossover may have occurred in the initial *H. vulgare* × *H. bulbosum* hybrid progeny, or the actual donor plant was related but not identical to the sequenced A17. A similar recombination event was observed on chromosome 2H of IL 88, also derived from A17 (Supplementary Figs. 30 and 33). To further expand the application of our pangenome, we developed universal *H. bulbosum* identification marker regions based on the *H. bulbosum* single-copy pangenome and 76 barley genomes[19], and used our IL assembly to validate these marker regions (Supplementary Fig. 34).

## Genomic dissection of *Ryd4*^*Hb*^

If haplotype diversity is high, genome sequences are indispensable in the search for candidate genes and the discovery of structural variants[32]. We explored genomic variation at two resistance gene loci using local pangenome graphs. We focused on two loci, *Mlo* and *Ryd4*^*Hb*^. *Mlo* is a well-studied locus, recessive alleles of which

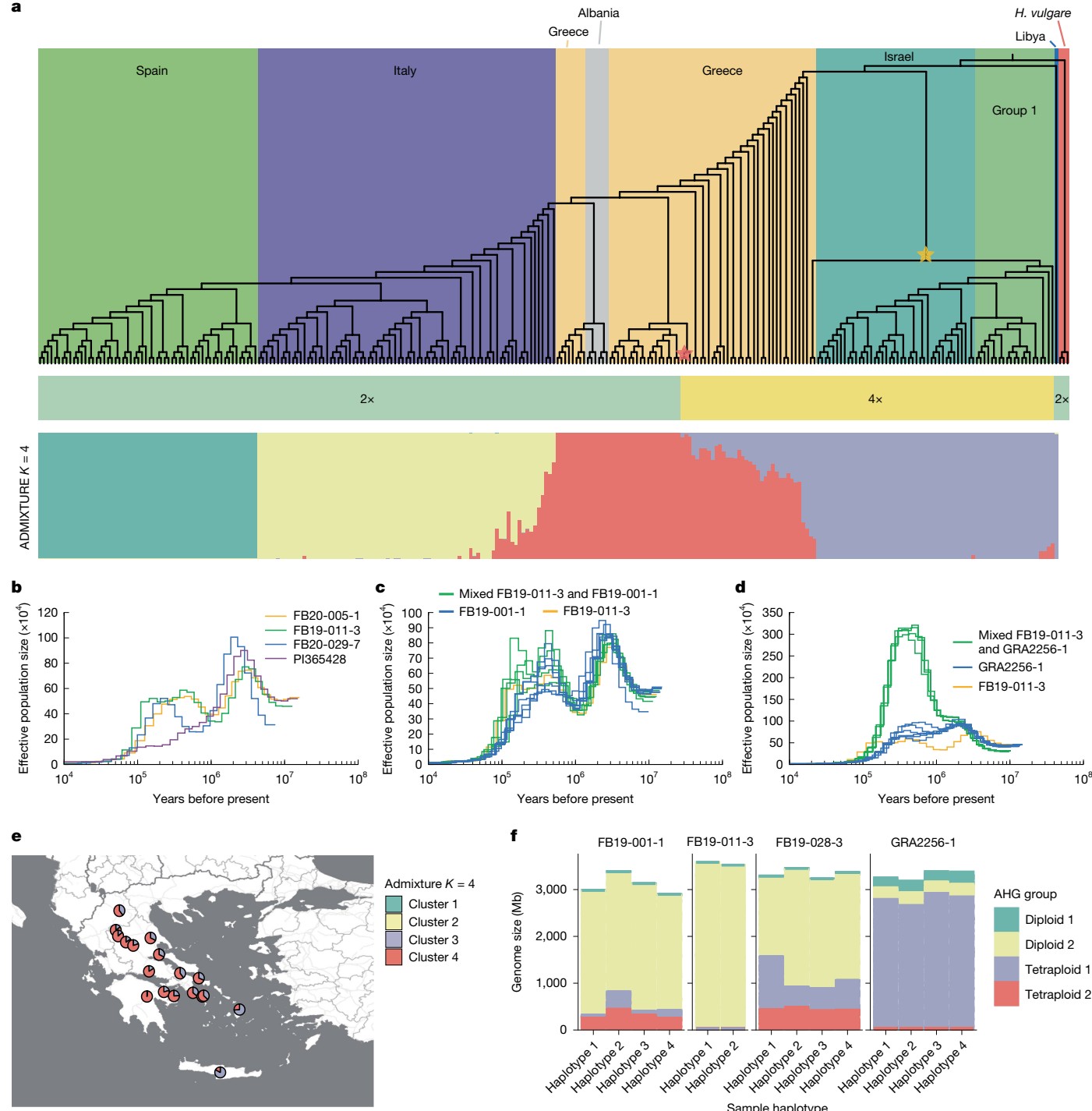

**Fig. 4 | Multiple origins of tetraploid cytotypes. a**, The upper panel shows the neighbour-joining tree of the 270 *H. bulbosum* genotypes. The background color of the tree indicates geographic origin. The colors of the middle bar plot indicate ploidy. Group 1 comprises samples from Armenia, Bulgaria, Tajikistan, Turkey, Ukraine and Uzbekistan and those of unknown provenance. The lower panel is the model-based ancestry estimation with ADMIXTURE. The bar plot shows the ancestry coefficent with *K* = 4 populations. **b**–**d**, Population size trajectories as inferred by pairwise sequentially Markovian coalescent analysis of heterozygous genomes (four diploid genomes (**b**); the Greek diploid FB19-011-3, the Greek tetraploid FB19-001-1 and synthetic diploids between them (**c**); and

FB19-011-3, the tetraploid GRA2256-1 from Tajikistan and synthetic diploids (**d**)). Generation time = 2 years; mutation rate = 0.7 × 10⁻⁸ per site per generation. **e**, Map depicting the collection sites of Greek tetraploids. Pie charts show the ADMIXTURE ancestry coefficients. The base map was created using tiles provided by Stadia Maps, styled by Stamen Design, and incorporating geographic data from OpenStreetMap contributors via OpenMapTiles. Map data © OpenStreetMap contributors, licensed under the Open Database License; visual design © Stamen Design, licensed under CC BY 3.0; map tiles © Stadia Maps, licensed under CC BY 4.0. **f**, AHGs as inferred by IntroBlocker. The bar plots show the proportion of each *H. bulbosum* genome that is assigned to one of four AHGs.

confer resistance to fungal pathogens[33]. Barley ILs carrying the *Ryd4^Hb* locus show complete dominant resistance to Barley yellow dwarf virus (BYDV), an insect-transmitted pathogen of barley[34].

At *Mlo* and *Ryd4^Hb*, haplotype diversity is higher in *H. bulbosum* than in *H. vulgare* (Supplementary Figs. 35 and 36 and Supplementary Table 15).

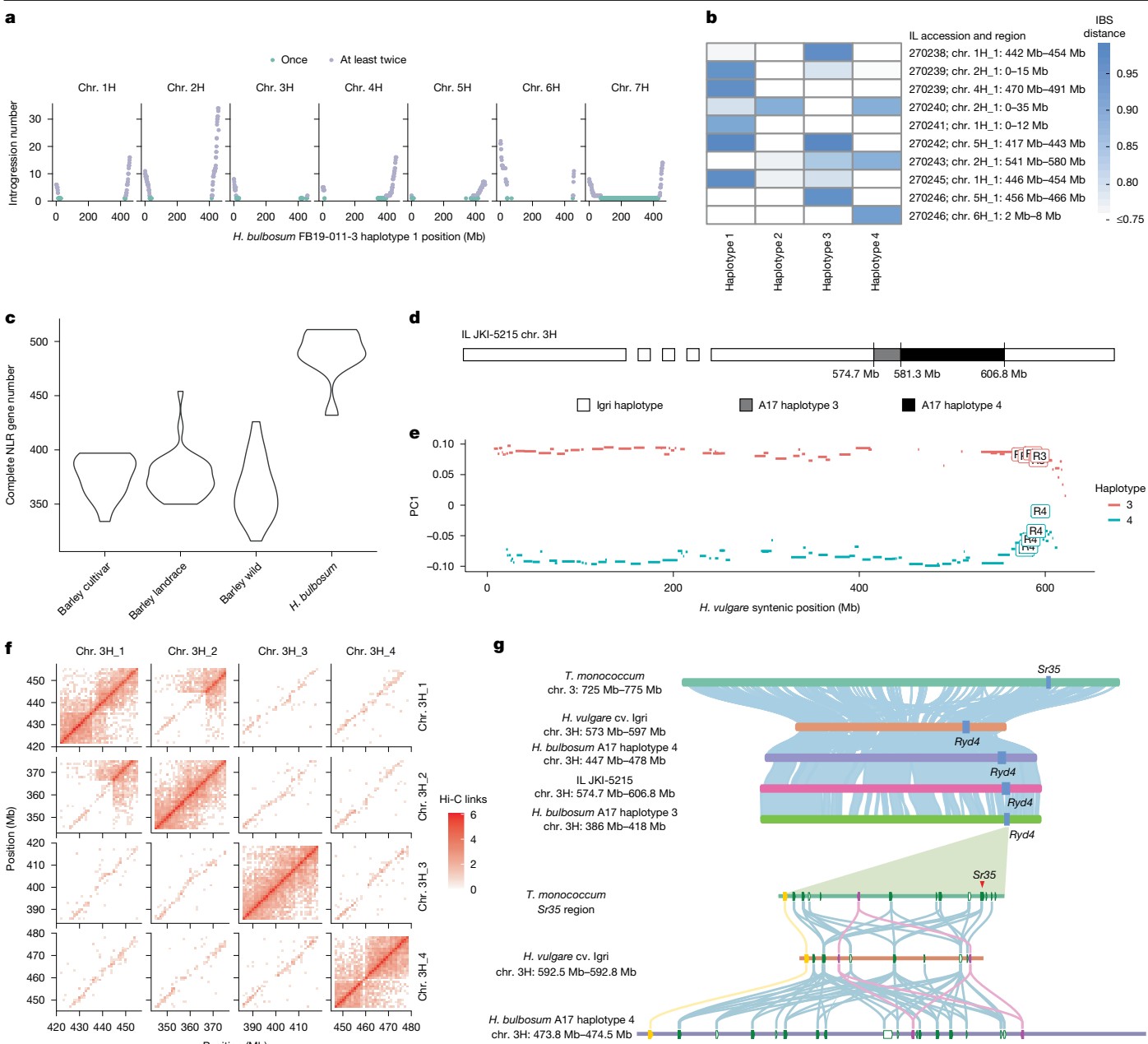

**Fig. 5 | The *H. bulbosum* pangenome supports introgression breeding in barley. a**, Introgressed regions of the *H. bulbosum* genome in barley, shown by the number of lines carrying each 1-Mb segment. **b**, Heat map showing similarity of introgressed segments to four A17 haplotypes from the *H. bulbosum* pangenome. IBS, identity-by-state. **c**, Number of NLR genes annotated in barley and *H. bulbosum* haplotypes. **d**, Schematic of the recombined haplotype in IL JKI-5215 (barley cv. Igri background). **e**, PCA of A17 haplotypes 3 and 4 on chromosome 3H, using Hi-C link matrices to distinguish contigs near introgressed regions. R3 corresponds to contigs that are in the region of A17 haplotype 3, co-linear to the JKI-5215 introgressed region. R4 corresponds to contigs that are in the region of A17 haplotype 4, co-linear to the JKI-5215 introgressed region. PC1, principal component 1. **f**, Hi-C contact matrix of the A17 region co-linear to the JKI-5215 introgressed segment. **g**, Microsynteny comparison of the *Sr35* and *Ryd4Hb* resistance loci. The upper panel shows long-range synteny among Igri, A17, *Triticum monococcum* TA10622 and JKI-5215; the lower panel compares local gene order and content between *H. vulgare* cv. Igri and A17 haplotype 4. Genes are colour-coded: yellow, *S*-formylglutathione hydrolase-like; purple, ankyrin-repeat proteins; green, complete NLRs; white/green outline, partial NLRs. *Sr35* is marked in red. Syntenic links (≥ 80% nucleotide identity) are colour-coded by gene family.

We used our genome sequences to further explore the complexity of *Ryd4Hb*. This locus has been delimited to a 65.5-kb interval in the genome of barley cv. Morex that contains two resistance gene homologues of the NLR class[35]. The closest *Ryd4Hb* flanking markers delimited a 692.6-kb interval in the *Ryd4Hb* IL JKI-5215 that contained 16 NLR genes, 4 of which were possibly functional and are therefore considered as candidates (Supplementary Table 16). The sequences of the proteins encoded by these four genes are between 58.7% and 86.6% identical.

Two, Ryd4_NLR1 and Ryd4_NLR5, are the closest homologues of and share 80% of their amino acid sequences with the protein encoded by the wheat stem rust resistance gene *Sr35* (Fig. 5g). The *Sr35* locus is also homologous and syntenic to the *Rph13* leaf rust resistance locus of barley[36]. Consistent with diversifying selection often seen at plant resistance gene loci[37], haplotype diversity in the *Ryd4Hb* interval was high in *H. bulbosum*: the JKI-5215 sequence was private to haplotype 4 of A17. Apart from IBD of three haplotype pairs in A17, GRA2256-1

and PI365428, all haplotypes were singletons. The locus was structurally diverse, mainly owing to duplications of NLR genes. Among the 32 haplotypes of the *H. bulbosum* pangenome, NLR numbers varied from 2 to 23, of which between 2 and 18 had all 3 typical domains: coiled-coil, nucleotide-binding adaptor shared by APAF1, certain R gene products and CED4 (NB-ARC), and leucine-rich repeats (Extended Data Fig. 11d and Supplementary Fig. 37). This exceeded the level of diversity seen in 76 *H. vulgare* genomes[19], for which NLR numbers at the *Ryd4Hb* locus ranged from 2 to 21; most *H. vulgare* accessions carried fewer than 8 NLRs. Clustering based on structural similarity of local pangenome graphs in *H. bulbosum* revealed that cluster composition is associated with the number of NLR genes. By contrast, clustering in *H. vulgare* seems to be influenced by domestication status (Supplementary Fig. 36).

Deployment of *Ryd4Hb* in barley breeding has been hampered by the presence of a sublethality factor, which has been delimited to a 483-kb region in barley cv. Morex, homologous to a 2.3-Mb interval in A17 haplotype 4. The recombination event in JKI-5215, which removed lethality, narrowed it down further to 656 kb in A17 and 188 kb in the recurrent parent Igri. Inspection of the gene content in this region (Supplementary Table 17) revealed a homologue of *RecA*, a component of the DNA repair machinery, as a plausible candidate gene. A17 haplotype 4 has three copies, all of which differ in their 5′ regions from their barley homologue. In particular, repositioned start codons might affect transcription and translation. These polymorphisms were not found in barley or other *H. bulbosum* haplotypes. The sublethality of *Ryd4Hb* introgressions illustrates that loss of heterozygosity in ILs increases the mutation load in crop–wild ILs. A survey of deleterious variants in the *H. bulbosum* pangenome showed that diploid and younger tetraploids contain fewer potentially harmful mutations (Extended Data Fig. 11e). Here, deleterious mutations have the same trend as haplotype-specific single-copy core sequences (Fig. 2d). Their use as donors might reduce the mutational load of newly generated ILs.

## Discussion

Pangenomes of crops and their relatives will be assembled in the coming years in increasing numbers, as will collections of genome sequences for a given crop and all of its wild relatives in a wider taxonomic group such as a genus[38]. As sequence reads become longer, haplotype phasing in autotetraploid or heterozygous crop genomes will become easier[39,40]. The contiguity and completeness of resultant assemblies will still depend on the differentiation of haplotypes, which is closely linked to the level of diversity in a species. In the case of *H. bulbosum*, sequence diversity between haplotypes was large enough to obtain and arrange these into haplotype-resolved pseudomolecules. The availability of chromosome-scale genome sequences for crop wild relatives will be useful for introgression breeding. Genome sequences help design markers for tracing the inheritance of alien chromatin in a crop background, confirm genotype identity and match ILs to donor haplotypes. We advocate for the construction of genome sequences of all donor genotypes in new introgression breeding programmes to underpin the confirmation and mapping introgression events. Future research should explore the possibility to enrich haplotype diversity in barley by using established ILs as recipient parents. Crossovers between *H. vulgare* and *H. bulbosum* chromosomes might thus be directed to existing segments of *H. bulbosum* chromatin in the barley genome. The successful introgression of new haplotypes might then be ascertained by sequence-based genotyping or full-genome sequencing.

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

## Methods

### Sample collection

*H. bulbosum* populations in the Central and Western Mediterranean were visited between the end of May and early October and about ten ripe spikelets each of three to seven individuals throughout each population were collected, bagged individually per mother plant, and transported to Gatersleben. Most collections took place between 2019 and 2022, with a few populations already visited in the years before. Seeds were germinated in pots, and one seedling per mother plant and three to five seedlings per population were grown further, regularly repotted and kept until the flowering stage. With the onset of flowering, the individuals belonging to the same population were placed together in isolation greenhouses to obtain seeds. The population samples were complemented by accessions from germplasm collections of the gene banks of IPK Gatersleben and the US Department of Agriculture to also represent genotypes from northern Africa and western Asia. Ploidy levels of all individuals were determined by flow cytometry on a Cyflow Space (Sysmex Partec) with propidium iodide as the stain and using *Vicia faba* cv. Tinova (IPK Genebank accession FAB 602; https://doi.org/10.25642/IPK/GBIS/33373; 2C = 26.21 pg) as the size standard, otherwise essentially following the procedure described previously[41]. Collecting and exchange of seeds followed the regulations of the respective countries and Nagoya agreements for material exchange. One herbarium voucher for each population was deposited in the herbarium of IPK Gatersleben (GAT).

### DNA extraction for genome sequencing

DNA extraction for reduced representation libraries and Illumina short-read sequencing was conducted with DNeasy Plant Mini kits (Qiagen) from young leaves dried in silica gel. DNA extraction followed the protocol of the manufacturer. DNA quality was checked on agarose gels, and for quantification, the Qubit dsDNA High Sensitivity assay kit (Thermo Fisher Scientific) was used.

To obtain high molecular weight (HMW) DNA for long-read sequencing 8 g leaf tissue was collected per individual, ground with liquid nitrogen to a fine powder and stored at −80 °C. HMW DNA was purified from the powder, essentially as described previously[42]. Briefly, nuclei were isolated, digested with proteinase K and lysed with SDS. Here a standard watercolour brush with synthetic hair (size 8) was used to resuspend the nuclei for digestion and lysis. HMW DNA was purified using phenol–chloroform extraction and precipitation with ethanol[42]. Subsequently, the HMW DNA was dissolved in 50 ml TE (pH 8.0) and precipitated by the addition of 5 ml 3 M sodium acetate (pH 5.2) and 100 ml ice-cold ethanol. The suspension was mixed by slow circular movements resulting in the formation of a white precipitate (HMW DNA), which was collected using a wide-bore 5-ml pipette tip and transferred for 30 s into a tube containing 5 ml 75% ethanol. The washing was repeated twice. The HMW DNA was transferred into a 2-ml tube using a wide-bore tip, collected with a polystyrene spatula, air-dried in a fresh 2-ml tube and dissolved in 500 μl 10 mM Tris-Cl (pH 8.0). For quantification, the Qubit dsDNA High Sensitivity assay kit (Thermo Fisher Scientific) was used. The DNA size profile was recorded using the Femto Pulse system and the Genomic DNA 165 kb kit (Agilent Technologies). In typical experiments, the peak of the size profile of the HMW DNA for library preparation was around 165 kb.

### HiFi sequencing

HMW DNA was provided to the University of Wisconsin Biotechnology Center DNA Sequencing Facility. The quality of the extracted DNA was measured on a NanoDrop One instrument (Thermo Fisher Scientific). Concentrations, 260/230 ratios and 260/280 ratios were logged. Quantification of the extracted DNA was measured using the Qubit dsDNA High Sensitivity kit (Thermo Fisher Scientific). Samples were diluted before running on the Agilent FemtoPulse System to assess DNA sizing and quality. A Pacific Biosciences HiFi library was prepared according to PN 101-853-100 Version 03 (Pacific Biosciences). Modifications include shearing with Covaris gTUBEs and size selecting with Sage Sciences BluePippin. Library quality was assessed using the Agilent FemtoPulse System. The library was quantified using the Qubit dsDNA High Sensitivity kit. The library was sequenced on a Sequel II using Sequel Polymerase Binding Kit 2.2 at the University of Wisconsin–Madison Biotechnology Center DNA Sequencing Facility.

### Hi-C sequencing

In situ Hi-C libraries were prepared from young seedlings according to the previously published protocol, using DpnII for the digestion of crosslinked chromatin[2]. Sequencing and Hi-C raw data processing was performed as described before[43,44].

### Genotyping-by-sequencing library construction and sequencing

Genotyping-by-sequencing (GBS) library construction using PstI and MspI (NEB) was done essentially as described previously. Samples were provided with unique dual barcodes and sequenced (1 × 107 cycles, single read) involving a custom sequencing primer[45] on the Illumina NovaSeq 6000 device (Illumina).

### Transcriptomics

Clones of individuals of FB19-011-3, GRA2256, A17, A40 and A42 were grown in a greenhouse at the Leibniz Institute of Plant Genetics and Crop Plant Research. Tissue samples of root, bulb, stem, leaf, stalk, flag leaf and young inflorescences (including flowers) were harvested (Supplementary Table 7), frozen in liquid nitrogen and stored at −80 °C. Both leaf tissues were obtained at the end of the night to reduce the number of photosynthesis-related transcripts in the extracted RNA. Total RNA was isolated with the RNeasy Plant Mini kit (Qiagen) following the instructions provided by the manufacturer. Quality was determined by spectrophotometry, and RNA integrity was determined using the RNA ScreenTape Assay on an Agilent 4200 TapeStation System (Agilent Technologies). RNA-sequencing libraries were constructed from 500 ng total RNA using the Illumina Stranded mRNA Prep Ligation Kit (Illumina) according to the manufacturer's protocol. The libraries were sequenced (XP workflow, paired-end, 2 × 109 cycles) on an Illumina NovaSeq 6000 device according to standard manufacturer protocols (Illumina). RNAs of the single-tissue samples were pooled for isoform sequencing (Iso-Seq). An Iso-Seq library was constructed for FB19-011-3 from 500 ng total RNA following the Iso-Seq Express 2.0 workflow as described by the manufacturer (Pacific Biosciences of California). The cDNA was amplified by three additional PCR cycles as recommended by the manufacturer. The Iso-Seq library (average size: 2.5 kb) was sequenced using the PacBio Sequel IIe device (24 h video time, 100 pM loading concentration, 2 h pre-extension time, diffusion loading) following standard manufacturer protocols (Pacific Biosciences of California).

### Diploid genome assembly

HiFi reads were assembled with hifiasm (0.13-r308)[46]. Unitigs were used as input for pseudomolecule construction. The GFA output was converted to FASTA format with gfatools (v0.4-r179, https://github.com/lh3/gfatools). Hi-C reads were aligned to the unitigs using the TRITEX pipeline[47]. High-confidence gene models annotated on the barley Morex V3 assembly[31] were used as a guide map. The gene sequences were aligned to the unitigs with GMAP[48]. Unitigs were assigned to chromosomes using the guide map and Hi-C information. A Hi-C matrix considering only reads mapping within 2 Mb of contig borders was constructed. The Hi-C matrix was normalized with a linear model to remove positional effects arising from the Rabl configuration as described previously[26]. A PCA was run on the normalized Hi-C matrix in a chromosome-wise manner considering only contigs longer than 1 Mb. Separation on the first principal component was used to group

the unitigs into haplotypes. Smaller or otherwise unphased contigs were assigned to haplotypes on the basis of majority rule using Hi-C information. Haplotype-wise pseudomolecule construction was performed with TRITEX[47,49]. Hi-C contact matrices were visually inspected and, if necessary, corrected for positional placement and chromosomal and haplotypical assignment as described previously[47] (Extended Data Fig. 1a).

## Optical map for the diploid clone FB19-011-3

The optical map was constructed on the platform of Bionano Genomics following procedures described previously[31] with several modifications. Briefly, HMW DNA was prepared from 2.5 million nuclei released from young leaves and purified by flow cytometry as described previously[50]. Next, 525 ng DNA was directly labelled at DLE-1 sites and analysed on a Saphyr instrument (Bionano Genomics). The collected 1.54 terabase pairs (Tb) of single-molecule data with N50 of 230 kb provided 146× effective coverage of the *H. bulbosum FB19-011-3* genome. The data were de novo-assembled by Bionano Solve 3.6.1_11162020, applying the configuration file *optArguments_nonhaplotype_noES_noCut_BG_DLE1_saphyr.xml*. A *P*-value threshold of $1 \times 10^{-12}$ was used to build the initial assembly, a *P*-value threshold of $1 \times 10^{-13}$ was used for extension and refinement steps (five rounds), and a *P*-value threshold of $1 \times 10^{-17}$ was used for the final merging of maps. To increase the contiguity of the sequence assembly, an automatic hybrid scaffold pipeline integrated in Bionano Solve 3.6.1_11162020 was run with the de novo optical map assembly. The default 'DLE-1 Hybrid Scaffold' configuration file was used with the 'Resolve conflict' option for conflict resolution. Scaffolding was performed for either of the two FB19-011-3 haplotypes. The validity of each link in the hybrid scaffolds was assessed on a case-by-case manner through inspection of Hi-C contact matrices and the genetic map from pollen sequencing. Valid links resulted in the correction of contig placements and orientations and the incorporation of previously unplaced contigs into the Hi-C map. Manual editing of the Hi-C map was carried out as described previously[47].

## Flow sorting and sequencing of single pollen nuclei

Anthers with mature pollen grains of 20–30 *H. bulbosum* FB19-011-3 florets were collected in a 1.5-ml Eppendorf tube, and the corresponding nuclei were isolated by applying the filter bursting method[51] using nuclei isolation buffer according to ref. 52 and CellTrics disposable filters of 100 μm and 20 μm (SYSMEX). The resulting nucleus suspension was stained with propidium iodide (50 μg ml$^{-1}$) and analysed on a BD Influx (BD Biosciences) by blotting the propidium iodide fluorescence against the side scatter signal. The nucleus population was selected and a corresponding sorting gate was defined in a histogram displaying the fluorescence intensity. Single haploid pollen nuclei were sorted into individual wells of a 96-well plate containing 2 μl distilled water using the '1.0 Drop Single' sort mode of the BD FACS Software (BD Biosciences).

Illumina NGS libraries were prepared from 96 single pollen nuclei using the PicoPLEX Gold Single Cell DNA-Seq kit (R300670, Takara Bio USA) essentially as described before[22]. For amplification and addition of unique dual barcodes, the libraries were denatured (95 °C for 3 min), amplified for 4 cycles (95 °C for 30 s, 63 °C for 25 s, 68 °C for 1 min) followed by 14 cycles (95 °C for 30 s, 68 °C for 1 min) and stored at 4 °C. Pooling (2-μl aliquots of each individual library) and size-fractionation using a SYBR-Gold-stained 2% agarose gel were as described previously[22]. Here, a narrow size range of the pooled library (approximately 300–500 bp) appropriate for sequencing using a patterned flow cell was excised and purified[22]. After the addition of 1.5% PhiX DNA as a control, the final pooled library was sequenced (SP flow cell, 2 × 151 cycles paired-end, dual-indexing with 8 cycles per index) using the Illumina NovaSeq 6000 instrument according to protocols provided by the manufacturer (Illumina).

## Genetic map construction

Primary data analysis including read mapping, SNP calling and filtration was carried out as described previously[22] with some modifications. Haplotypes 1 and 2 of FB19-011-3 were considered as parental genomes. Reads were aligned to haplotype 1 of that genotype. Simulated short reads were included in the analysis to determine the positions of segregating variants. Reads were simulated with wgsim (v0.3.1-r13, https://github.com/lh3/wgsim). Genotype calls were aggregated in 200-kb bins. Binned genotypes were used for genetic map construction with ASMAP (v1.0-2)[53]. We determined the phasing error rate as the proportion of SNP loci that do not support the assembly haplotype among all SNPs. We defined the main haplotypes in each pollen sample and determined whether there are genotype calls in the sample that do not match the majority haplotype. The majority haplotype depends on the genotype calls of all SNPs in a given sample and chromosome. We considered a SNP to be in wrongly phased regions if fewer than 50% of samples supported the majority haplotype (Extended Data Fig. 1c). To further verify the phasing of our genome assembly, we aligned the pollen sequencing reads to the concatenated genome of FB19-011-3 haplotypes 1 and 2. This confirmed that the genome phasing matched the SNP-based phasing results. Additionally, we used the Minigraph-Cactus pangenome pipeline (v2.6.5)[54] to construct a graph genome from the assemblies of both parental haplotypes, filtering out structural variations smaller than 50 bp. The graph genome index was built using vg (v1.49.0)[55] autoindex, and pollen sequencing reads were aligned with vg giraffe[56]. Finally, pollen sequencing was genotyped using vg call[57], and variation statistics for the PASS results were screened. The outcome was consistent with SNP-based phasing results (Supplementary Fig. 3).

## Autotetraploid genome assembly

Primary unitig assembly and haplotype-wise pseudomolecule construction were carried out as for diploids. To assign a haplotype to one or more of the four possible haplotypes, we first constructed a Hi-C contact matrix considering only reads mapping within 2 Mb of contig borders. Only single-copy unitigs (that is, those with an average coverage consistent with their presence in only a single haplotype) were considered. PCA was run on the Hi-C matrices in a chromosome-wise manner. Clusters were defined by the *k*-means method on principal components 1 and 2. The cumulative lengths of the defined clusters were inspected to decide whether they were present in one or more haplotypes. *k*-means clusters composed of single-copy unitigs from multiple types were subjected to another round of PCA. Once four haplotypes were obtained, pairwise PCA including only unitigs from any group of two haplotypes was conducted. If no clear separation was evident, unitigs belonging to inseparable clusters and their Hi-C links were used as input, and the clustering was repeated recursively until complete separation in pairwise PCAs (Extended Data Fig. 5c) was achieved. Multi-copy unitigs (that is, those with coverage levels consistent with their presence in more than one haplotype) were assigned to haplotypes. For example, if a unitig had double coverage, we determined the two haplotypes that are closest in the PCA to that unitig among all four haplotype centres (Supplementary Fig. 38). Hi-C links and a majority rule were used to map smaller or other otherwise unphased unitigs to haplotypes. Multi-copy unitigs were included in the pseudomolecule construction of each haplotype they were assigned to (Extended Data Fig. 1d).

## Chloroplast genome assembly

HiFi reads of *H. bulbosum* genomes were aligned to the chloroplast genome of barley cv. Morex (GenBank accession code EF115541) using minimap2 (version 2.24-r1122)[58] with the parameters -ax map-hifi --secondary=no --sam-hit-only. Alignments with quality below 30 were discarded. Aligned reads were converted to FASTQ format with SAMtools (version 1.16.1)[59] and assembled with Canu (version 2.1.1)[60] in the

PacBio HiFi mode with expected genome sizes ranging from 120 kb to 400 kb in 10-kb increments. All circular sequences were extracted from the assemblies, and BEDTools (version 2.30.0)[61] was used to identify the sequences' circular starting points. Genes and other features on the circular sequences were annotated with GeSeq[62]. The assemblies with complete annotations were selected and aligned to the Morex chloroplast sequence with Mummer4 (version 4.0.0beta2)[63]. The starting positions and directions of the small single-copy region match the Morex reference with seqkit (version 0.9.1)[64]. Multiple sequence alignment was carried out with MAFFT (version 7.490). A phylogenetic tree was constructed from the alignments with IQ-TREE (version 2.2.0-beta)[65] using the parameters -m MFP -bb 1000 -bnni -redo -o wheat.

### Mitotic and male meiotic chromosome preparations
Mitotic root-tip cells were prepared as described previously[66], and male meiotic chromosomes were prepared according to ref. 67 and either stained with acetocarmine (Morphisto) for scoring chiasmata or used for FISH.

### Preparation of haplotype-specific oligonucleotide-based FISH probes and in situ hybridization
For the diploid *H. bulbosum* accession FB19-011-3, BBDuk (BBMap_37.93)[68] was used to mask regions in the genome where 31-mers appear more than once. BEDtools (2.30.0)[61] was utilized to identify the unmasked regions in the genome. All unmasked regions that were longer than 45 bp in length were selected as candidate regions for probe design. For the tetraploid *H. bulbosum* accession FB19-028-3, we increased the $k$-mer size from 31 to 34 $k$-mers to increase the probe density. This higher $k$-mer size allows for identifying more distinct regions in the genome for probe design. Non-overlapping target-specific oligonucleotides were selected for the final probe sets and synthesized as myTAGs Labeled Libraries (Daicel Arbor Bioscience). In the diploid probe, oligonucleotides were labelled with either Atto 594 (red) or Alexa 488 (green). In the tetraploid probe, oligonucleotides were labelled with Atto594 (red, haplotype 1), Alexa488 (green, haplotype 2), Atto 647N (far-red, haplotype 3) and digoxygenin (haplotype 4). The digoxygenin-labelled oligonucleotides were detected by applying simultaneously anti-digoxygenin-rhodamine (https://www.sigmaaldrich.com/DE/de/product/roche/11207750910) and anti-digoxigenin-fluorescein (https://www.sigmaaldrich.com/DE/de/product/roche/11207741910). Chromosomes were counterstained with 4′,6-diamidino-2-phenylindole (DAPI) in Vectashield antifade solution (Vector Laboratories). 5S (pCT4.2; ref. 69) and 45S (pTa71; ref. 70) rDNA, HvT01 (ref. 71) and *Arabidopsis*-type telomer FISH probes were labelled by nick translation with Texas Red and Atto488 (NT labelling kits, Jena Biosciences). FISH with repeat-specific probes and chromosome painting probes was carried out according to refs. 67 and 72, respectively.

### Microscopy
Wide-field fluorescence images were captured using a microscope BX61 (Olympus Europa SE) equipped with an Orca ER CCD camera (Hamamatsu) or a Nikon Eclipse Ni-E fluorescence microscope equipped with a Nikon DS-Qi2 camera and NIS-Elements-AR version 4.60 software (Nikon). The final contrast of the images and pseudo colouring was processed using Adobe PHOTOSHOP software or GIMP 2.10 (https://www.gimp.org).

### Population genetic analysis based on short-read data
GBS data for 263 *H. bulbosum* plants (Supplementary Table 12) and 10 genome assemblies were used. A total of 350 million simulated short reads were generated per assembly with wgsim (version 1.14, https://github.com/lh3/wgsim). Reads were trimmed with cutadapt (version 1.15) and aligned to the haplotype 1 sequence assembly of FB19-011-3 with BWA-MEM (v0.7.17-r1188)[73]. Alignment records were converted to Binary/Alignment Map format with SAMtools[59] and sorted with Novosort (v3.09.01, http://www.novocraft.com/products/novosort/). BCFtools (v1.9)[59] was used to call SNPs. The resultant VCF files were filtered with a custom AWK script (https://bitbucket.org/ipk_dg_public/vcf_filtering) using the options '--minmaf 0 --dphom 4 --dphet 8 --mindp 50 --minhomn 1 --minhomp 0.05 --tol 0.1 --minpresent 0.4 --minqual 40' and imported into R using seqArray[74]. Only SNPs were used in downstream analyses. PCAs were carried out with the snpgdsPCA() function of the package SNPrelate[75] on SNPs with less than 20% missing data and a minor allele frequency of ≥5%. A distance matrix was calculated with PLINK (v1.90b6.9)[76] using the parameters '--distance 1-ibs flat-missing square' and transformed into a neighbour-joining tree with FASTME (version 2.1.5)[77]. ADMIXTURE (version 1.3.0)[78] was run with $k$ ranging from 1 to 9 on a linkage-disequilibrium-pruned SNP matrix with less than 10% missing data. Linkage disequilibrium pruning was carried out with PLINK using the option '-indep-pairwise 50 10 0.1'. The optimal number of populations ($k$ = 4) was determined by selecting the $k$ with the lowest cross-validation error.

### Variant discovery in the haplotype-resolved pangenome
SNPs, indels and structural variants (SVs) were discovered in a panel of 33 sequence assemblies comprising the 32 assembled *H. bulbosum* haplotypes and the inbred genome of barley cv. Morex. Calling of SNP and indels was carried out with simulated long-read data using the FB19-011-3 haplotype 1 assembly as the reference. For each haplotype or genome, 30× coverage and 20,000–40,000 length reads were simulated with Badread (v0.4.1)[79]. Reads were aligned to the haplotype 1 sequence assembly of FB19-011-3 with pbmm2 (https://github.com/PacificBiosciences/pbmm2). DeepVariant (v1.6.0)[80] with the pretrained PacBio mode (--model_type PACBIO) was then used for variant calling of each accession, and all individual variants were merged using glnexus_cli (v1.4.1)[81] with the DeepVariant config file. SNPs and indels were filtered with the parameters '--minmaf 0 --mindp 30 --minhomn 1 --dphom 3 --dphet 20 --minhomp 0 --threads 10 --tol 0.5 --minpresent 0 --minqual 20' using an AWK script (https://bitbucket.org/ipk_dg_public/vcf_filtering). After filtering for SNPs with less than 20% SNP missing data, a neighbour-joining tree was constructed as for the GBS data. To call 'small' SVs in the size range 50 bp to 20 kb, the variants were called with Svim (v1.4.2)[82]. To call 'large' (>20 kb) SVs, genome sequences were aligned to the common FB19-011-3 reference with wfmash (v0.8.2, https://github.com/waveygang/wfmash) with the parameters '-p 90 -l 0', and SVs were called with SyRI (version 1.6)[83].

### Single-copy pangenome
A single-copy pangenome was constructed as described previously[24] (https://bitbucket.org/ipk_dg_public/barley_pangenome/) with one modification. MMSeq2[84] was used with the option '--cluster-mode' instead of BLAST for all-versus-all alignment. A minimum sequence identity of 95% was required to accept matches. To estimate pangenome complexity, the lengths of the largest sequences in each cluster were summed up. To construct a single-copy pangenome based on ten samples, regions where the masked genome $k$-mer appears at least three times are considered for the diploid genome, and regions where the $k$-mer appears five times or more are used for the tetraploid genome. To identify haplotype-specific core genome sequences, we excluded core genome regions that were not scaffolded to pseudomolecule chromosomes and checked their presence in each haploid. Any region absent in a particular haploid was defined as a haplotype-specific core genome sequence.

### Pangenome graph
The 32 assembled *H. bulbosum* haplotypes were divided by chromosome, and then the Minigraph-Cactus pangenome pipeline (v2.6.5)[54] was used to construct the chromosome- and graph-based pangenome with the following parameters: --filter 4 --giraffe clip filter --viz --odgi --chrom-vg clip filter --chrom-og --gbz clip filter full --gfa clip

full --vcf. We used ODGI (v0.8.6-2)[85] similarity to calculate the average distance of each haplotype genome in the chromosome- and graph-based pangenome, and the distance matrix was transformed into a neighbour-joining tree with FASTME (version 2.1.5)[77]. In the FB19-011-3 haplotype 1 genome, a 100-kb region upstream and downstream of *mlo* was selected, and the sequence was aligned to 32 *H. bulbosum* and 76 barley genomes using minimap2 to obtain the local sequences. The local graph-based pangenome was then constructed using PGGB (v0.5.3)[86] with the parameters -p 95 -s 1000. *Ryd4* sequences in *H. bulbosum* pangenomes and Morex were obtained by aligning flanking markers[35] using BLAST+[87]. *Ryd4* sequences in barley pangenomes were obtained through alignment to the Morex *Ryd4* sequence using minimap2. The local graph-based pangenome was then constructed using PGGB with the parameters -p 90 -s 1000.

## Annotation and analysis of TEs

Structurally intact and fragmented TEs in each genome were annotated using the Extensive de novo TE Annotator (EDTA, v1.9.0)[88]. TREP[89] (https://trep-db.uzh.ch/) was used as a curated library in addition to the manual selection of TE categories according to the classification from EDTA (https://github.com/oushujun/EDTA/blob/master/util/TE_Sequence_Ontology.txt). The barley Morex V3 high-confidence gene models[31] were provided with the --cds option to exclude genic sequences from the TE annotation. The length, classification, insertion time and divergence information of each intact long terminal repeat retrotransposon was determined from the TE annotation output files of each genome (intact.gff3 and pass.list). TEsorter[90] was used to identify individual domains in BARE-1 families. Alignments of full-domain of long terminal repeat retrotransposons were generated with TEsorter's concatenate_domains.py script. Approximately maximum-likelihood phylogenetic trees were computed from these alignments with FastTree (version 2.1.11)[91]. Trees were visualized with iTOL[92].

## De novo gene annotation

To generate a uniform and comparable gene annotation for all *H. bulbosum* assemblies, and identify potential genotype-specific gene structures at the same time, we developed a hybrid gene prediction strategy. Structural gene annotation of all *H. bulbosum* assemblies was conducted by integrating ab initio gene calling and homology-based approaches with protein datasets and RNA-sequencing and Iso-Seq transcriptome data, for cases in which these were available. Transcriptome data were generated for A17, A40, A42, GRA2256_1 and FB19_011_3 in this study (see the section above entitled Transcriptomics) or obtained from public repositories (National Center for Biotechnology Information BioProject accession code PRJNA810915 (ref. 93)). All expression data were mapped using STAR (version 2.7.8a)[94] and assembled into transcripts with StringTie (version 2.1.5, parameters -m 150 -t -f 0.3)[95]. Triticeae reference protein sequences (UniProt, https://www.uniprot.org, 5 October 2016) were aligned using miniprot[96] (0.11-r234), and the Isoseq dataset of FB19_011_3 was aligned with minimap2 (2.26-r1175)[97]. Assembled transcripts and aligned protein sequences were combined using Mikado (v2.3.4)[98].

Ab initio annotation was performed using Augustus (version 3.3.3)[99] and was performed on all assemblies. To avoid potential over-prediction, guiding hints were generated from RNA-sequencing, protein and Iso-Seq datasets and TE annotation as described previously[99]. A *H. bulbosum*-specific prediction model was built by generating a set of gene models with full support from RNA sequencing and Iso-Seq, and Augustus was trained and optimized using the steps detailed previously[99]. All annotations were joined using EVidenceModeller[100] (version 2.1.0) with adjusted weights: ab initio (Augustus: 10), homology-based (8), protein-based (4). Two rounds of PASA (version 2.5.3)[101] identified untranslated regions and isoforms. Gene models were classified as high or low confidence on the basis of BLASTP

coverage against reference database hits (Uniprot Magnoliophyta, reviewed/Swissprot, downloaded on 3 August 2016; https://www.uniprot.org; PTREP (Release 19; http://botserv2.uzh.ch/kelldata/trep-db/index.html)).

Best hits were selected for each predicted protein from each of the three databases. Only hits with an *e*-value below $10 \times 10^{-10}$ were considered. This differentiated candidates into complete and valid genes, non-coding transcripts, pseudogenes and TEs. Hits with subject coverage (for protein references) or query coverage (transposon database) above 75% were considered significant, and protein sequences were classified as high-confidence using the following criteria: protein sequence was complete and had a subject and query coverage above the threshold in the UniMag database or no BLAST hit in UniMag but in UniPoa and not PTREP; a low-confidence protein sequence was incomplete and had a hit in the UniMag or UniPoa database but not in PTREP. Alternatively, it had no hit in UniMag, UniPoa or PTREP, but the protein sequence was complete.

## Gene projection

To consolidate the above annotation for gene models potentially missed in one or several *H. bulbosum* genotypes, we combined the high-confidence annotations of all genotypes to a non-redundant collection of (source) genes and projected them to the genome sequences, similar to an approach previously described in the second version of the barley pangenome[19]; overview scheme provided at https://github.com/GeorgHaberer/gene_projection/tree/main.

In brief, we clustered the entire set of the high-confidence proteins applying CD-hit[102] (-i 1.0 -S 4 -d 80, and otherwise default parameters) to remove nearly identical models and to generate a total of 479,530 non-redundant source models. Source proteins were mapped to genome sequences of all genotypes by miniprot[96] (-G 70k, --outs 0.7, --outc 0.7, and gff outformat). Source transcripts were aligned to the genome sequences using both minimap2 (ref. 97) with options '-ax splice:hq' and '-uf' and BLAT[103] v34 with options '−fine' and '−noTrimA' and a maximal intron size of 70 kb. Subsequently, quality tags were assigned to the source models by normalized scores, orthology and protein domains, and non-overlapping projected gene models were derived by stepwise insertions of mapped models proceeding from higher to lower qualities. Definitions, preprocessing and parameters of the quality criteria as well as a description of the projection pipeline and custom python scripts used for *H. bulbosum* annotations are listed in detail in the repository https://github.com/GeorgHaberer/gene_projection/tree/main/panhordeum.

For the final consolidation step, we integrated all of the above consensus projections with the evidence-based annotations if they did not overlap their coding regions and had a hit to a pfam domain with an *e*-value of $\leq 1 \times 10^{-10}$. The script for this padding step is also provided in the above-mentioned GitHub repository.

## Construction of the gene-based pangenome

Orthologous groups based on the primary protein sequences from *H. bulbosum* genotypes were calculated using Orthofinder[104] version 2.5.527 with standard parameters. For assemblies with phased chromosomes or polyploids, we included the gene predictions on both phases or subgenomes but treated them as one genotype in Orthofinder. The scripts for calculating core, shell and cloud genes have been deposited in the repository https://github.com/PGSB-HMGU/hbulbosum/tree/main.

Core orthologous groups contain at least one gene model from all compared *H. bulbosum* genotypes. Shell orthologous groups contain gene models from at least two *H. bulbosum* genotypes and at most nine *H. bulbosum* genotypes. Genes not included in any orthologous group ('singletons'), or clustered with genes only from the same genotype, were defined as cloud genes. Gene-based co-linearity plots were drawn with GENESPACE[105].

## Genome size change rates

The sequence assembly of FB19-011-3 haplotype 1 was aligned to the barley Morex V3 assembly[31] with minimap2. Alignments were filtered to retain those with alignment quality surpassing the 98th percentile and alignment lengths exceeding the 95th percentile. A generalized additive model was used with the R package mgcv[106] to combine and smooth the filtered alignments (Supplementary Fig. 24). The generalized additive model was utilized to infer the genome size of *H. bulbosum* in 1-Mb intervals of *H. vulgare* along the whole genome and quantify variations in genome size between the two species.

## Inference of demographic history

The pairwise sequentially Markovian coalescent software[107] was used to infer population size trajectories in *H. bulbosum*. We used a synonymous mutation rate per base per generation of $6.5 \times 10^{-9}$ as per ref. 108 and a generation time of 2 years. Simulated long reads were generated with splitfa (https://github.com/ekg/splitfa) using a window size of 24 kb and a step size of 800 bp. Reads of different haplotypes were combined to reconstitute diploid genomes or create synthetic heterozygotes aligned to the common reference FB19-011-3 haplotype 1 with minimap2 (ref. 58). Synthetic heterozygotes were created for: autotetraploids (six pairwise combinations per genotype); and between haplotypes of different genotypes to estimate divergence times between populations by looking for increases of population sizes not seen between haplotypes of the same genotype as described previously[107].

## Haplotype analysis

Haplotypes were defined on the basis of the SNP matrix whose generation is described in the section entitled Variant discovery in the haplotype-resolved pangenome. Genetic distances based on variant number were computed in 5-Mb genomic windows of the FB19-011-3 reference haplotypes using IntroBlocker[28]. The distribution of genetics distance revealed two major peaks (Extended Data Fig. 10a), one at 9,500 variants per megabase, and the other at 14,500 variants per megabase. A threshold of 11,000 was used to differentiate between these two peaks, and a genomic variant density of 2,500 SNPs per megabase was used in the Bayesian smoothing process of IntroBlocker. Haplotypes were defined with IntroBlocker in a semi-supervised manner using the priority order diploid 2 > diploid 1 > tetraploid 1 > tetraploid 2. Diploid 1 refers to diploid populations within Libya, diploid 2 refers to diploid populations outside Libya, tetraploid 1 refers to the first-formed tetraploid populations, including those from Central Asia, and tetraploid 2 refers to the tetraploid populations from Greece. A CNV block was defined as the area where the normalized read depth is less than 0.6 or greater than 1.4. Those regions were excluded from the analysis. The divergence times ($T$) were calculated using $T/g = D/2\mu$. The mean genetic distances ($D$) were derived from the peaks identified by Introblocker. The generation time ($g$) and mutation rate ($\mu$) were set as in the section entitled Inference of demographic history.

## Analysis of ILs

GBS data for ILs from ref. 15 were aligned to a synthetic diploid genome combining barley Morex V3 and FB19-011-3 haplotype 1. Alignment of GBS data was conducted with BWA-MEM using a hybrid *H. bulbosum* (FB19-011-3 haplotype 1) and *H. vulgare* (Morex V3) genome. Average read depths were calculated in 1-Mb windows along the genome with SAMtools, excluding secondary and non-unique alignments with the command 'samtools view -q20 -F3332'. Then we normalized the depth of samples: Normalized 1 Mb read depth = $\frac{\text{Aligned read number within 1 Mb}}{\text{Total aligned read number}} \times 10^6$. Windows of the *H. bulbosum* haplotype with more than 200 reads (normalized 1-Mb read depth) were considered as introgressed regions (Supplementary Table 13). The analysis repeated with each of the A17 haplotypes as possible *H. bulbosum* donors. Identity-by-state distances were calculated in the introgressed regions with PLINK to find the closest haplotype(s). We selected nine ILs (Supplementary Table 14) from GBS data and performed PacBio HiFi sequencing. The PacBio HiFi reads were assembled using hifiasm (0.13-r308). Alignment of HiFi reads was conducted with minimap2 using a hybrid *H. bulbosum* (FB19-011-3 haplotype 1) and *H. vulgare* (Morex V3) reference. Average read depths were calculated in 1-Mb windows across the genome using SAMtools. Genomic fragments from *H. bulbosum* (FB19-011-3 haplotype 1) and *H. vulgare* (Morex V3) were identified on the basis of coverage, and then pseudo-chimeric genomes were constructed using RagTag (v2.1.0)[109] and Morex V3 genome. The final IL pseudomolecule construction was completed with RagTag (v2.1.0) and the pseudo-chimeric genomes.

## NLR gene annotation and deleterious variants

Complete NLR genes in the sequence assemblies of 32 *H. bulbosum* haplotypes and 76 *H.* vulgare genomes were identified with NLR-Annotator[110]. To predict deleterious variants, SNP calling was carried out with simulated short-read data using the Morex V3 assembly as the reference. For each haplotype and genome, 270 million pairs of 150-bp reads were simulated with wgsim. Reads were aligned to the haplotype 1 sequence assembly of FB19-011-3 with BWA-MEM[73]. Alignment records were converted to Binary/Alignment Map format with SAMtools[59] and sorted with Novosort (http://www.novocraft.com/products/novosort/). BCFtools[59] was used to call SNPs. The resultant VCF files were filtered with a custom AWK script (https://bitbucket.org/ipk_dg_public/vcf_filtering) using the options --minmaf 0 --mindp 30 --minhomn 1 --minhomp 0 --tol 0.5 --minpresent 0 --minqual 4. Amino acid substitutions and their effects on protein function were predicted with SIFT[111]. A substitution was considered deleterious if its score was ≤0.05 and tolerated if the score was >0.05. Variants that occurred in the heterozygous state at least once in one sample were considered deleterious.

## Analysis of the ryd4 locus

The JKI-5215 line was used to identify the A17-derived *Ryd4^Hb* interval. It is derived from a BC$_2$F$_7$ plant from the population described previously[112]. *Ryd4^Hb* flanking markers[35] were aligned with BLAST+[87] against the assemblies of the pangenomes, and the resulting loci were extracted with SAMtools[59] faidx. The NLRs in those intervals were annotated with NLR-Annotator[110], and non-NLR genes in the interval in barley cv. Morex were aligned to identify their homologues. Intervals were compared using LASTZ alignments[113] plotted as dot plots in R[114]. Genes and protein similarities were obtained through MUSCLE alignments[115]. The *T. monococcum* genome[116] was downloaded from Graingenes[117] (https://avena.pw.usda.gov/genomes/mono/pan_home.php). The microsynteny comparison between the *Sr35* interval and the *Ryd4^Hb* interval was carried out with NGenomeSyn[118] and minimap2[97]. The *Ryd4^Hb* interval in the *H. vulgare* cv. Igri was aligned to the sequence assembly of the *Sr35* interval published in ref. 119 and the *Ryd4^Hb* interval in *H. bulbosum* A17 haplotype 4 with BLASTN[87]. We graphically represented the homology between annotated genes with more than 80% nucleotide identity in Fig. 5e.

## Reporting summary

Further information on research design is available in the Nature Portfolio Reporting Summary linked to this article.

## Data availability

All of the sequence data collected in this study have been deposited at the European Nucleotide Archive under the BioProject accession codes PRJEB65276 (genome assemblies, transcriptome Illumina data and GBS data) and PRJEB65918 (pollen nucleus single-cell sequencing data). Accession codes for individual genotypes are listed in Supplementary

Tables 1, 2 and 7 (pangenome assemblies and associated raw data), Supplementary Table 6 (pollen single-nucleus sequencing data) and Supplementary Table 12 (GBS). The assemblies and annotations are available for download from https://galaxy-web.ipk-gatersleben.de/libraries/folders/Fb701da857886499b. The variant matrix and pangenome graphs are available via the Plant Genomics and Phenomics Research Data Repository[120] at https://doi.org/10.5447/ipk/2025/4.

## Code availability

Code for genome assembly and data analysis is available via GitHub at https://github.com/jia-wu-feng/Pan_Bulbosum. Code for gene projection is available via GitHub at https://github.com/GeorgHaberer/gene_projection/tree/main/panhordeum. Code for analysis of orthologous groups is available via GitHub at https://github.com/PGSB-HMGU/hbulbosum/tree/main.

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

**Acknowledgements** We thank C. Koch and K. Kumke for technical assistance; the gene banks of IPK Gatersleben and the U.S. Department of Agriculture Agricultural Research Service for providing seeds for this study, and the countries of origin for permitting collections; and J.-M. Rouillard for support in the design of FISH probes. We utilized the University of Wisconsin–Madison Biotechnology Center's DNA Sequencing Facility (Research Resource Identifier RRID:SCR_017759) to generate and sequence PacBio HiFi libraries. This research was supported by IPK core funding. Genomic inquiries into barley wild relatives by M. Mascher. are supported by IPK core funding. Genomic inquiries into barley wild relatives by M. Mascher. are supported by the European Commission (ERC Starting Grant TRANSFER 949873). M.S. and K.F.X.M. acknowledge support by the German Federal Ministry of Education and Research (BMBF)

under project number 031B0190 (de.NBI). S.H. acknowledges support by the European Commission (ERC Starting Grant MEIOBARMIX 949618). M.C. acknowledges funding from the German federal state of Saxony-Anhalt (grant ZB I 180).

**Author contributions** M. Mascher, A. Houben, N.S. and F.R.B. designed research. F.R.B. collected *H. bulbosum* materials. F.R.B. and J.K. grew plant material. M.R. and A.S. contributed plant material. A. Himmelbach performed Illumina and PacBio sequencing. M. Maruschewski and A.F. handled sequence submission. R.H. conducted transcriptome studies. J.-W.F., M.J. and M. Mascher assembled genome sequences. J.-W.F. annotated TEs. D.H. contributed GBS data. J.-W.F. and Y.G. analysed diversity and evolution. H.T. and H.Š. carried out Bionano genome mapping. T.L., G.H., M.S. and K.F.X.M. annotated the genome assemblies. H.P., J.-W.F. and N.S. analysed the *Ryd4^{Hb}* locus. Y.G. contributed analysis methods. M.C., J.F., Y.-T.K., S.H. and A. Houben performed and interpreted cytological experiments. J.-W.F., H.P. and M. Mascher wrote the manuscript with input from all co-authors.

**Funding** Open access funding provided by Leibniz-Institut für Pflanzengenetik und Kulturpflanzenforschung (IPK).

**Competing interests** The authors declare no competing interests.

**Additional information**
**Correspondence and requests for materials** should be addressed to Frank R. Blattner, Nils Stein or Martin Mascher.

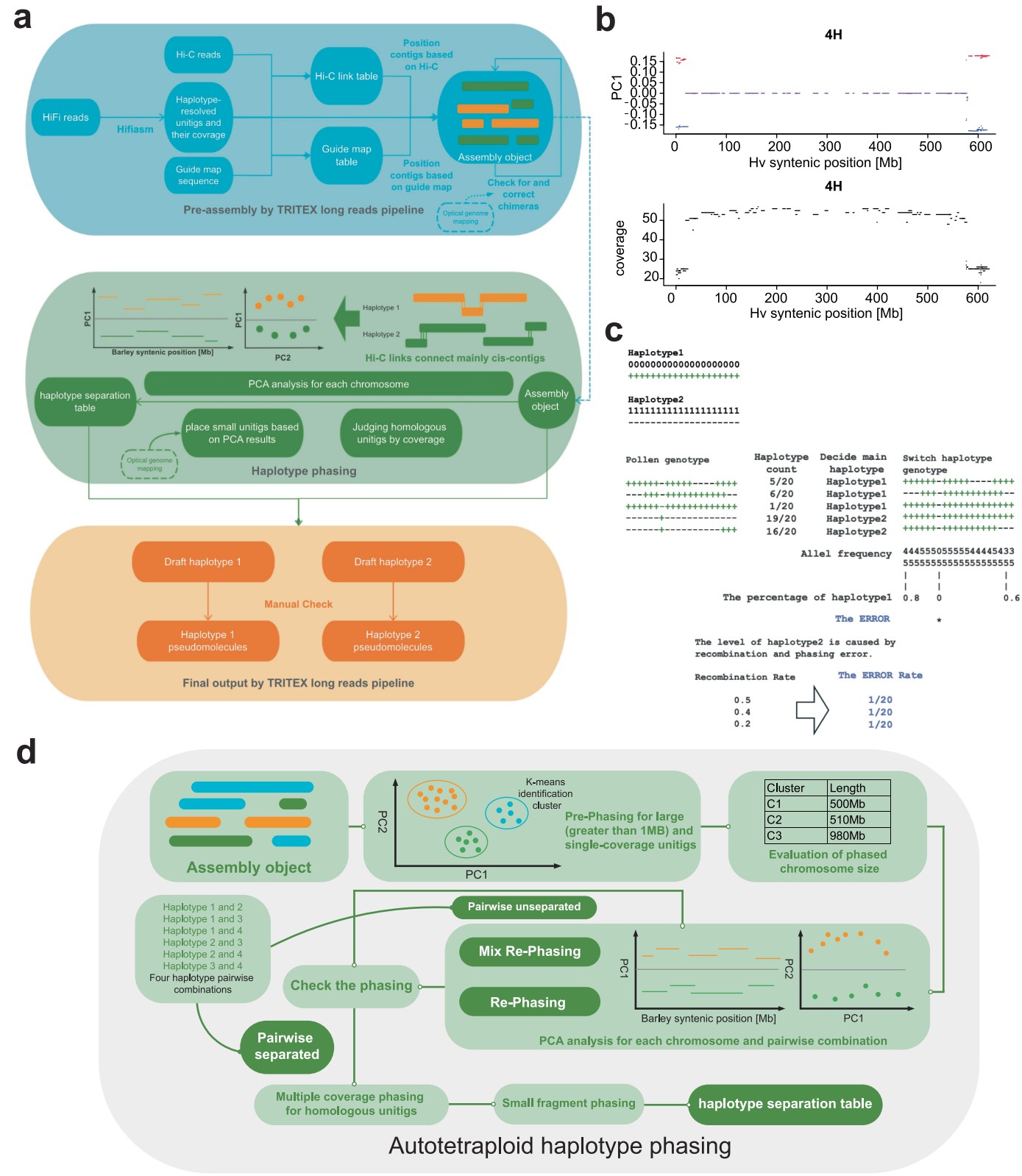

**Extended Data Fig. 1 | Pipeline for haplotype-resolved genome assembly.** (**a**) Flowchart of the diploid phasing pipeline. (**b**) Example of haplotype separation by PCA in diploid samples. An identity-by-descent region has double coverage in the unphased contig-level assembly. (**c**) Schematic overview of the pipeline for estimating phasing error rates from single pollen nuclei sequencing data. (**d**) Flowchart of the tetraploid phasing pipeline.

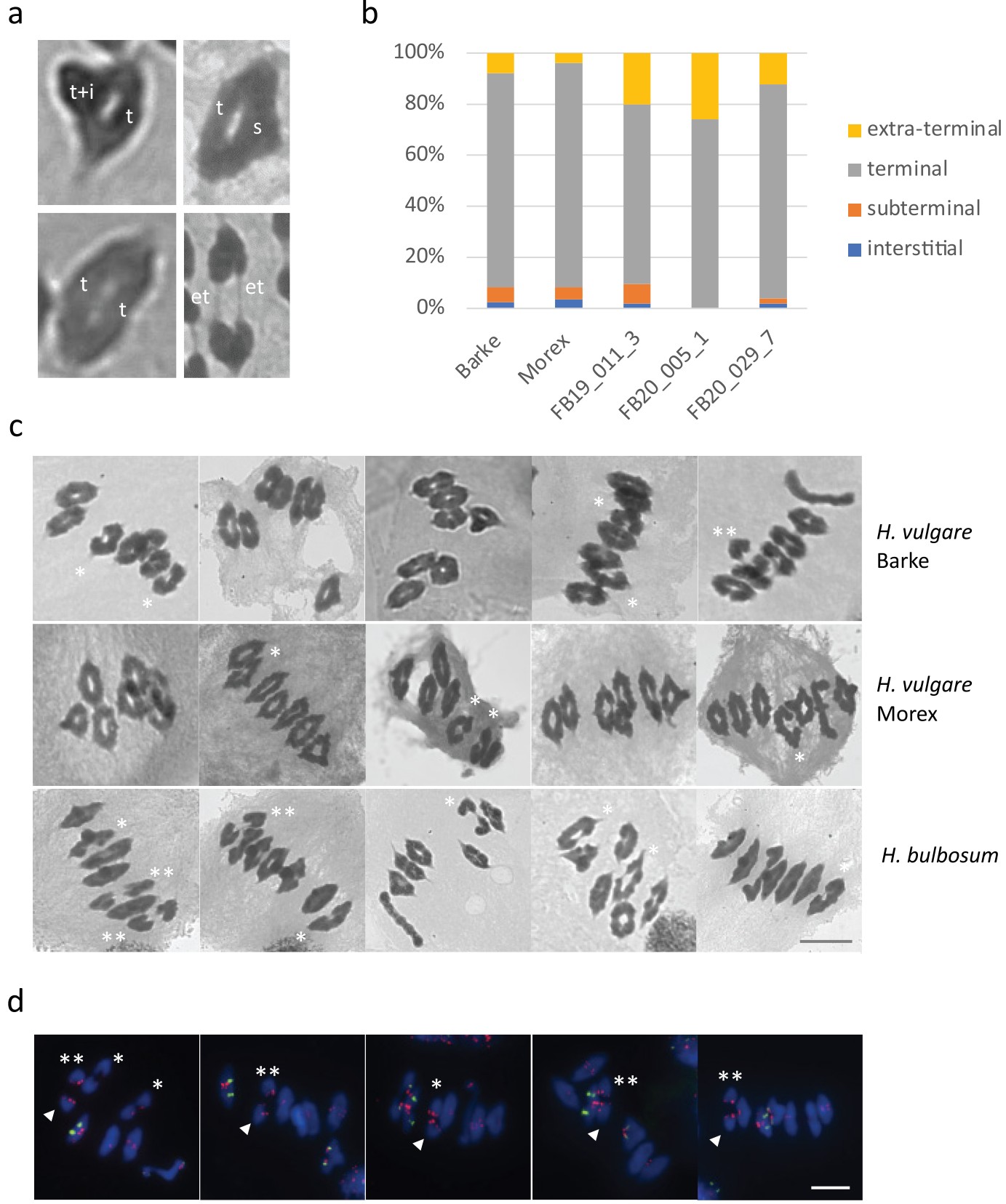

**Extended Data Fig. 2 | Extra-terminal crossovers occur in *H. bulbosum* more often than in *H. vulgare*.** (a) Representative examples of ring bivalents with interstitial (i), subterminal (s), terminal (t) and extra-terminal (et) chiasmata as calculated in (**b**). (**b**) Proportion of each type of chiasma based on the classification shown in (**a**). Number of cells scored from three independent plants: Barke: 110; Morex: 140; FB19_011_3:120; FB20_005_1:112; FB20_029_7:104. (**c**) Representative examples of meiotic cells at metaphase I stage of *H. vulgare* cultivars "Barke" and "Morex" and *H. bulbosum* genotype FB19-011-3 showing different bivalent configurations and chiasma positions as calculated in b. (**d**) FISH with 5S and 45S rDNA (green) and HvT01 (subtelomeric; red) on metaphase I cells. 26 cells analyzed from two independent plants. One chromosome pair (arrowhead) showed extra-terminal chiasmata in 19 out of 26 cells analysed. Asterisks: extra-terminal chiasmata. Size bar = 10 μm.

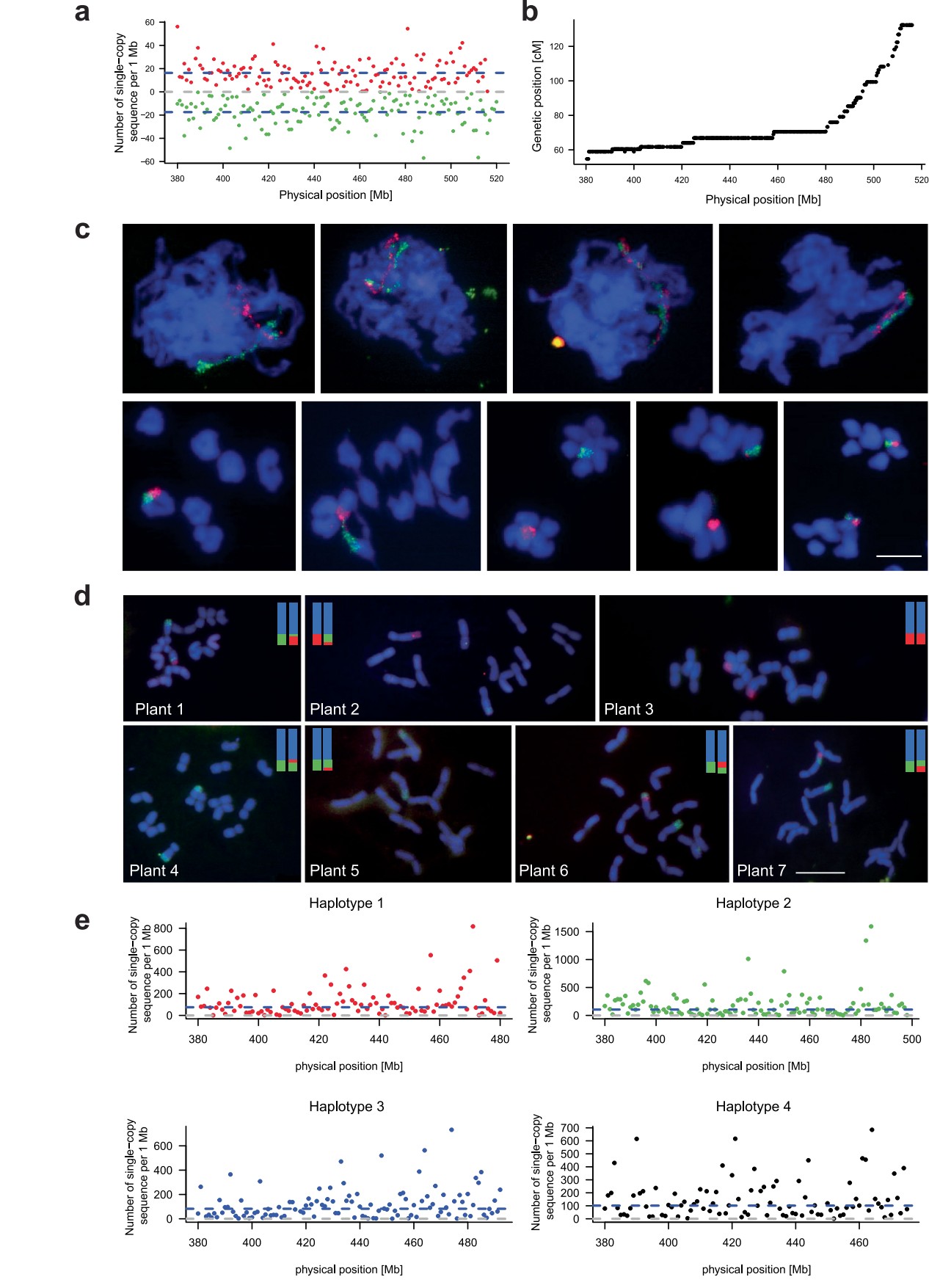

**Extended Data Fig. 3 |** See next page for caption.

**Extended Data Fig. 3 | Design of oligo-FISH probes.** (**a**) Haplotype-specific single-copy sequence in 1 Mb bins at the distal end of the long arm of chromosome 5H in FB19-011-3. (**b**) Relationship between the genetic and physical maps of the long arm of chromosome 5H. (**c**) Meiotic dynamics of chromosome 5H homologs. From left to right, top: leptotene, zygotene, and pachytene, diplotene. Haplotype 1 (green) and haplotype 2 (red) initially unpaired come in close apposition until becoming physically connected in synapsed chromosomes; bottom: metaphase I, early anaphase I, and three examples of late anaphase I showing the presence or absence of recombination between haplotypes 1 and 2. At least two independent experiments were carried out to confirm the reproducibility of the labeling patterns. Size bar = 10 μm. (**d**) Meiotic crossovers revealed by chromosome painting. Mitotic chromosomes of seven selfed offspring of the diploid *H. bulbosum* genotype FB19-011-3 after FISH with haplotype-specific probes. Explanatory graphical genotypes are shown in the top-right corners of each subpanel. At least two independent experiments were carried out to confirm the reproducibility of the labeling patterns. Size bar = 10 μm. (**e**) Haplotype-specific single-copy sequence in 1 Mb bins at the distal end of the long arm of chromosome 5H in FB19-028-3.

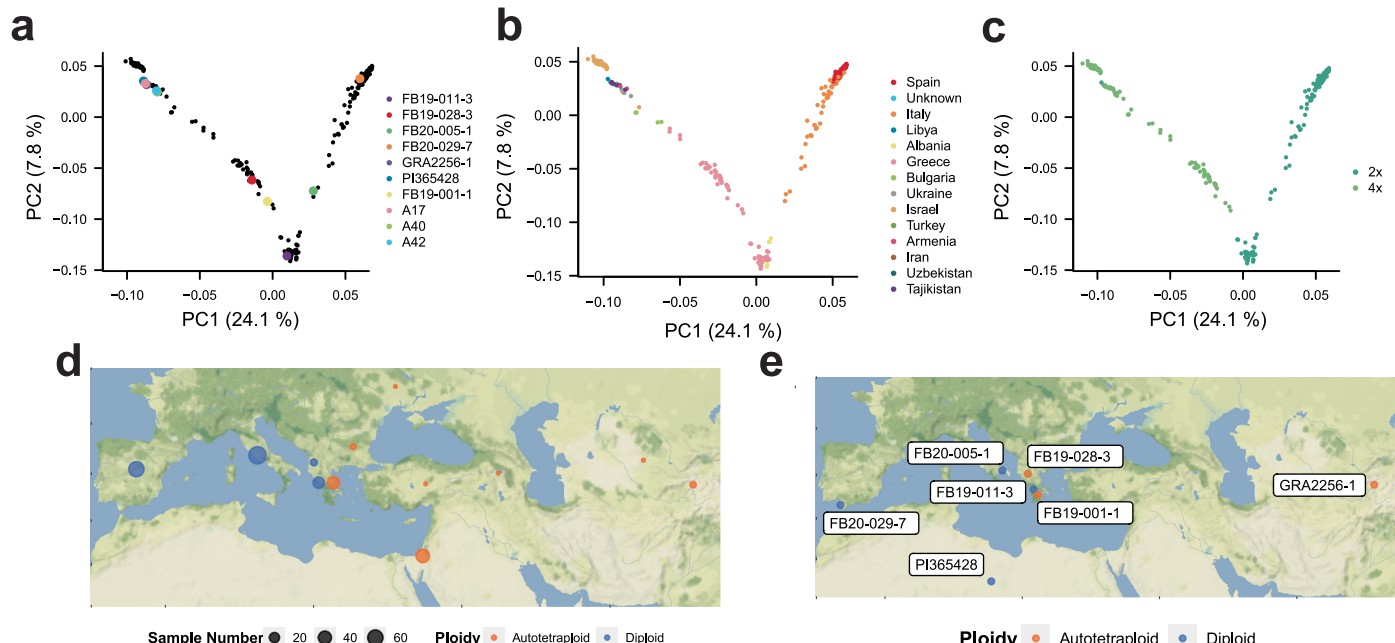

**Extended Data Fig. 4 | A pangenome selection in global *H. bulbosum* clones.**
(**a, b, c**) Principal component analysis (PCA) of 270 *H. bulbosum* genotypes.
(**a**) Positions of the ten selected pangenome accessions in the PCA diversity
space. The samples are colored according to country of origin (**b**) or ploidy (**c**).
(**d**) Geographic map showing the origins of 260 individuals subjected to
genotyping-by-sequencing. Information was aggregated at the country level.
(**e**) Map showing the geographic origins of seven *H. bulbosum* clones that are
part of the pangenome. The collection sites of another three accessions are
unknown. Colors indicate ploidy. For **d**,**e**, the base map was created using
tiles provided by Stadia Maps, styled by Stamen Design, and incorporating
geographic data from OpenStreetMap contributors via OpenMapTiles. Map
data © OpenStreetMap contributors, licensed under the Open Database
License; visual design © Stamen Design, licensed under CC BY 3.0; map tiles ©
Stadia Maps, licensed under CC BY 4.0.

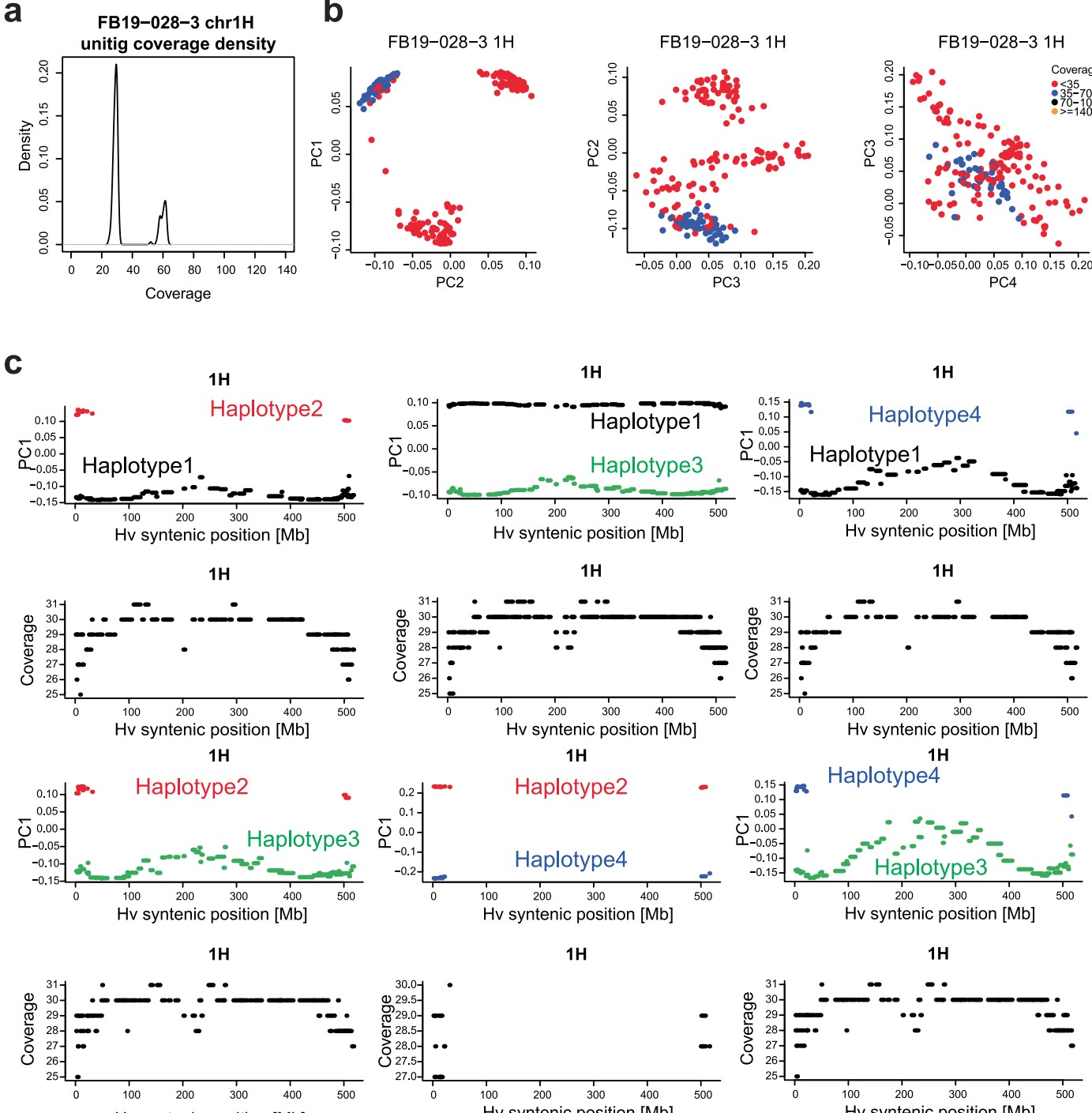

**Extended Data Fig. 5 | Haplotype phasing of autotetraploid *H. bulbosum* genomes.** (**a**) Distribution of sequence coverage of contigs in the unphased assembly. Contigs that are shared between two haplotypes have double coverage (right-hand peak). (**b**) PCA clustering of the matrix of Hi-C links connecting the ends of contigs, i.e. those mapped to within 2 Mb of either contig end. Chromosome 1H of the tetraploid clone FB19-028-3 is shown.

Colors correspond to coverage brackets. The contigs with double coverage (shown in blue) correspond to IBD regions shared between two haplotypes. (**c**) PCA separation and contig-level coverage on chromosome 1H of FB19-028-3 after pseudomolecule construction and manual curation. Pairwise differences between the four haplotypes are shown.

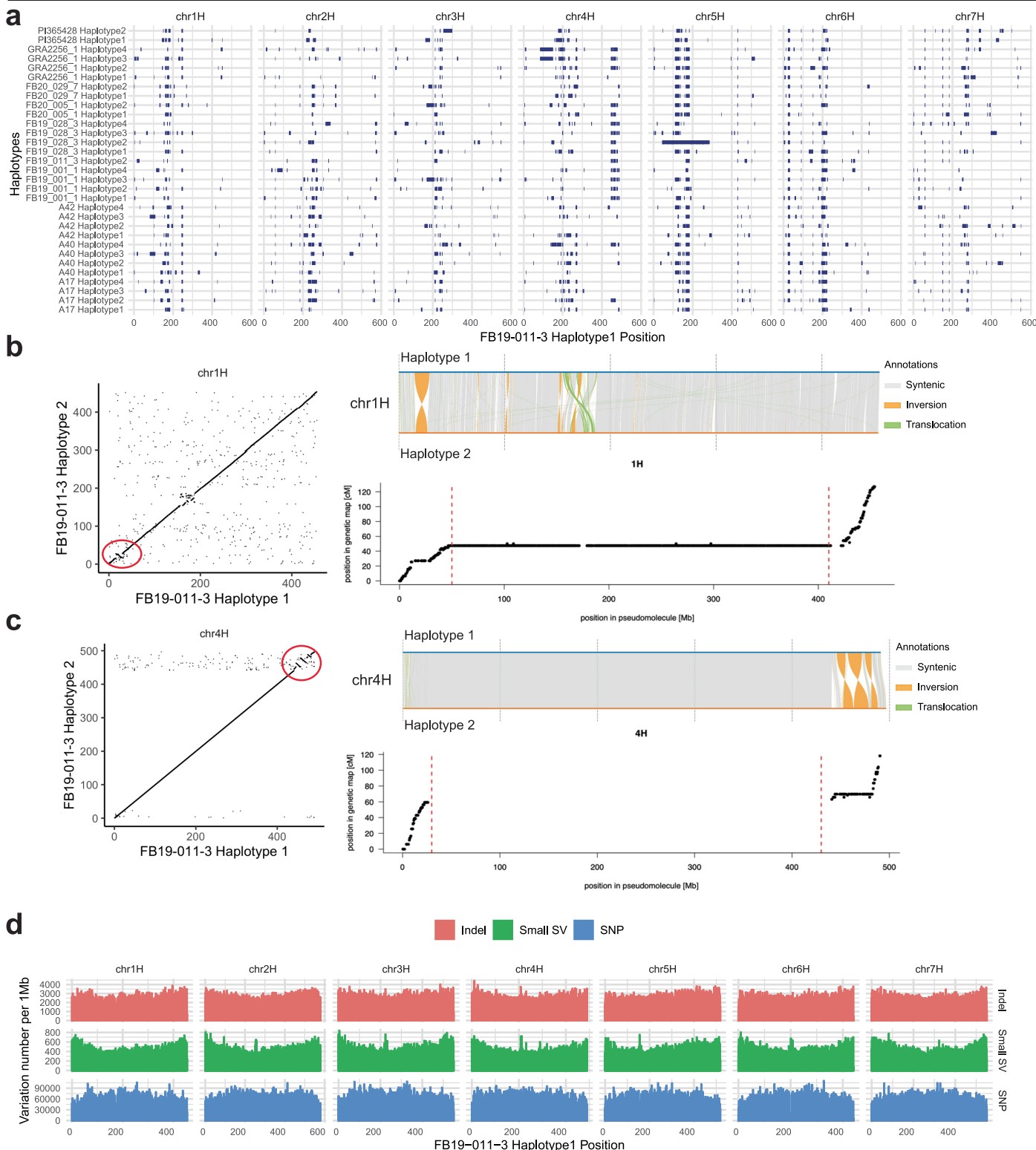

**Extended Data Fig. 6 | Structural variation in *H. bulbosum*. (a)** Genome-wide map of polymorphic inversions longer than 2 Mb. (**b**, **c**) Genetic validation of inversions between the haplotypes FB19-011-3 on chromosomes 1H (**b**) and 4H (**c**). The left-hand panels show chromosome-level alignments between both haplotypes. Red circles mark the inversions in question. The inversions were identified by SYRI in wfmash whole-genome alignments (top-right subpanels). The bottom-right subpanels show the correspondence between the physical and genetic maps. (**d**) Distribution of indels, SNPs and small (≥40 bp and ≤20 kb) SVs along the *H. bulbosum* genome.

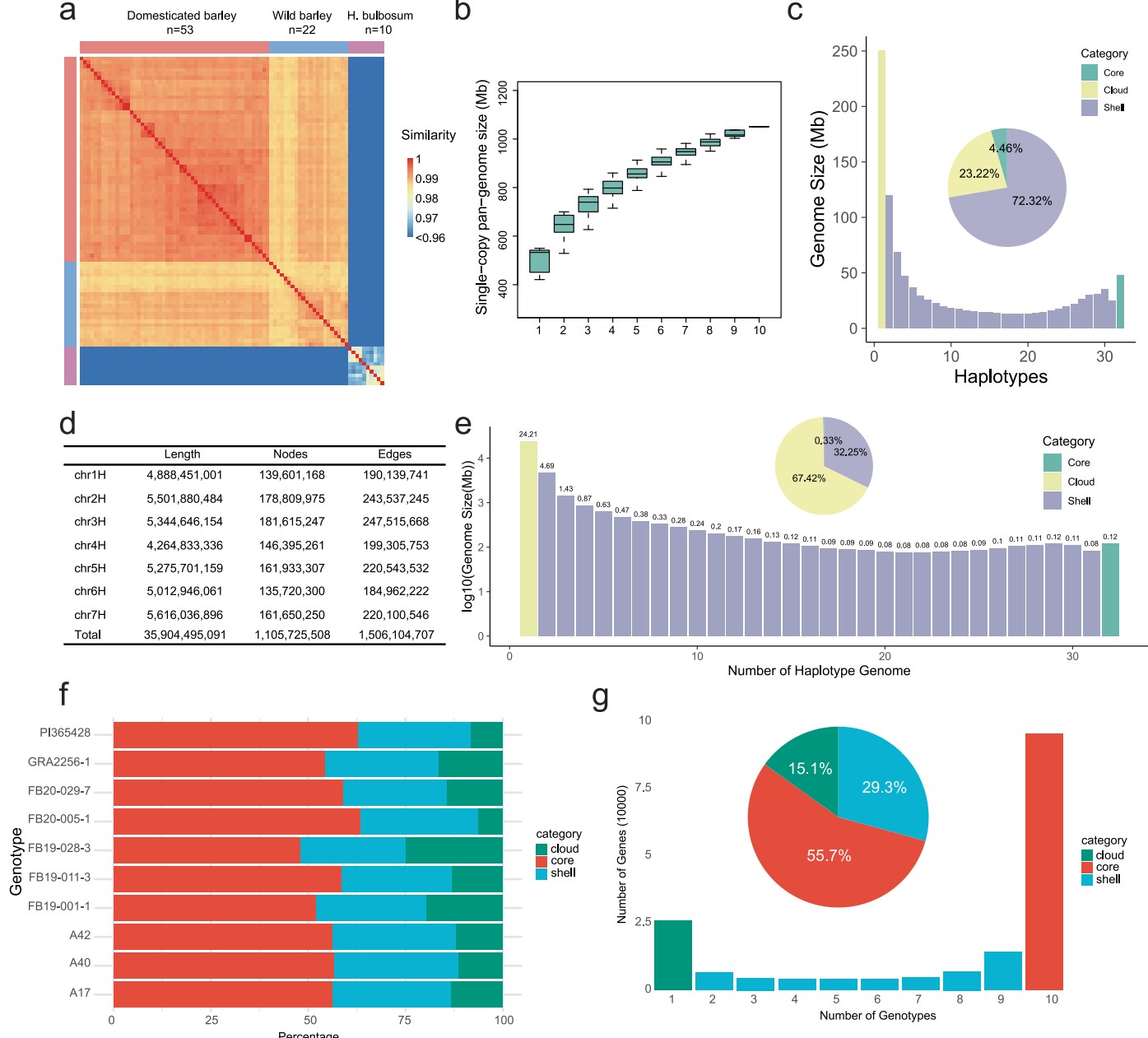

**Extended Data Fig. 7 | Pangenome analysis of *H. bulbosum*.** (**a**) Heatmap showing the similarity of k-mer hashes of *H. vulgare* and *H. bulbosum* genome sequences. (**b**) The cumulative size of single-copy regions in genome assemblies of 10 *H. bulbosum* genomes. The curves trace the growth of non-redundant single-copy sequences as sample size increases. Error bars derived from 100 ordered permutations each. The central line represents the median; the box spans the interquartile range (IQR), from the first quartile to the third quartile. Whiskers extend to the most extreme values within 1.5 × IQR from the quartiles. (**c**) Composition of core, shell, and cloud single-copy sequences in the haplotype-level pangenomes. (**d**) Summary of the *H. bulbosum* graph pangenome constructed by the Minigraph-Cactus pangenome pipeline. (**e**) Composition of core, shell, and cloud sequences in the haplotype-level graph-based pangenome. (**f**) Bar chart illustrating the proportion of genes contained in core, shell and cloud OGs (see Methods for details) by genotype. (**g**) Bar chart illustrating the number of *H. bulbosum* genotypes represented in the individual OGs. The x-axis gives the number of genotypes included in an OG. The pie chart provides ratios of conserved and variable genes for all 10 genotypes.

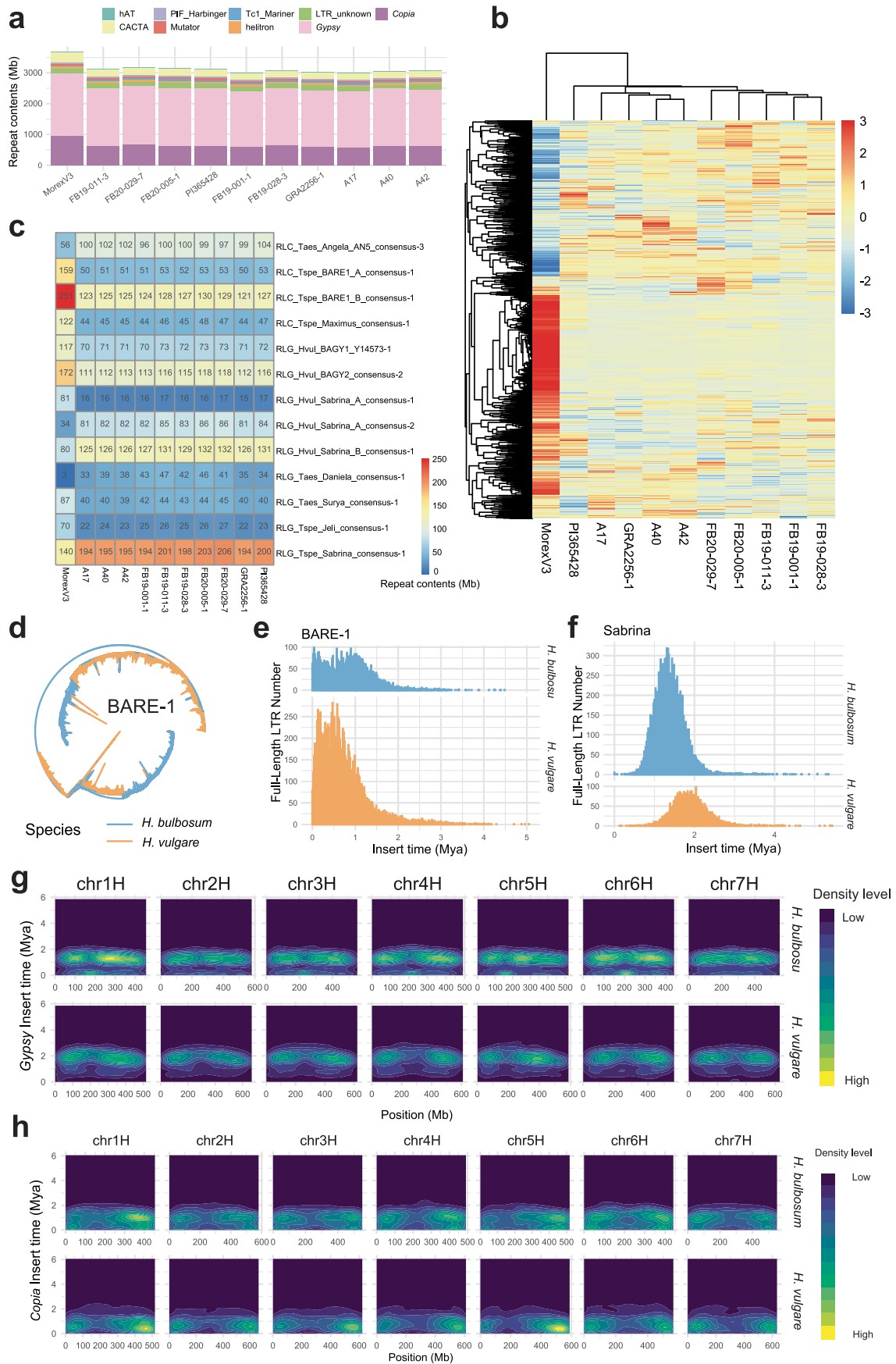

**Extended Data Fig. 8** | See next page for caption.

**Extended Data Fig. 8 | Evolution of TEs in *H. bulbosum* and *H. vulgare*.**
(**a**) Size and composition of the repetitive portion of the *H. bulbosum* and *H. vulgare* genomes. Haplotype 1 of each *H. bulbosum* genome is shown. (**b**) Heatmap constructed from a matrix tabulating the abundance of 989 TEs representatives from the TREP database. (**c**) Subsets of abundance matrix for the 13 most variable TE representatives. Numbers in the boxes and the color code refer to the cumulative lengths of TE sequences assigned to each representative per haplotype. (**d**) Approximately maximum-likelihood phylogenetic trees for all full-length BARE-1 elements in *H. vulgare* and *H. bulbosum*. (**e**) Distribution of the insertion times of all full-length BARE-1 elements in *H. vulgare* and *H. bulbosum*. (**f**) Distribution of the insertion times of all full-length Sabrina elements in *H. vulgare* and *H. bulbosum*. (**g**) Two-dimensional density plot showing the relationship between insertion time and genomic position of *Gypsy* elements. (**h**) Two-dimensional density plot showing the relationship between insertion time and genomic position of *Copia* elements.

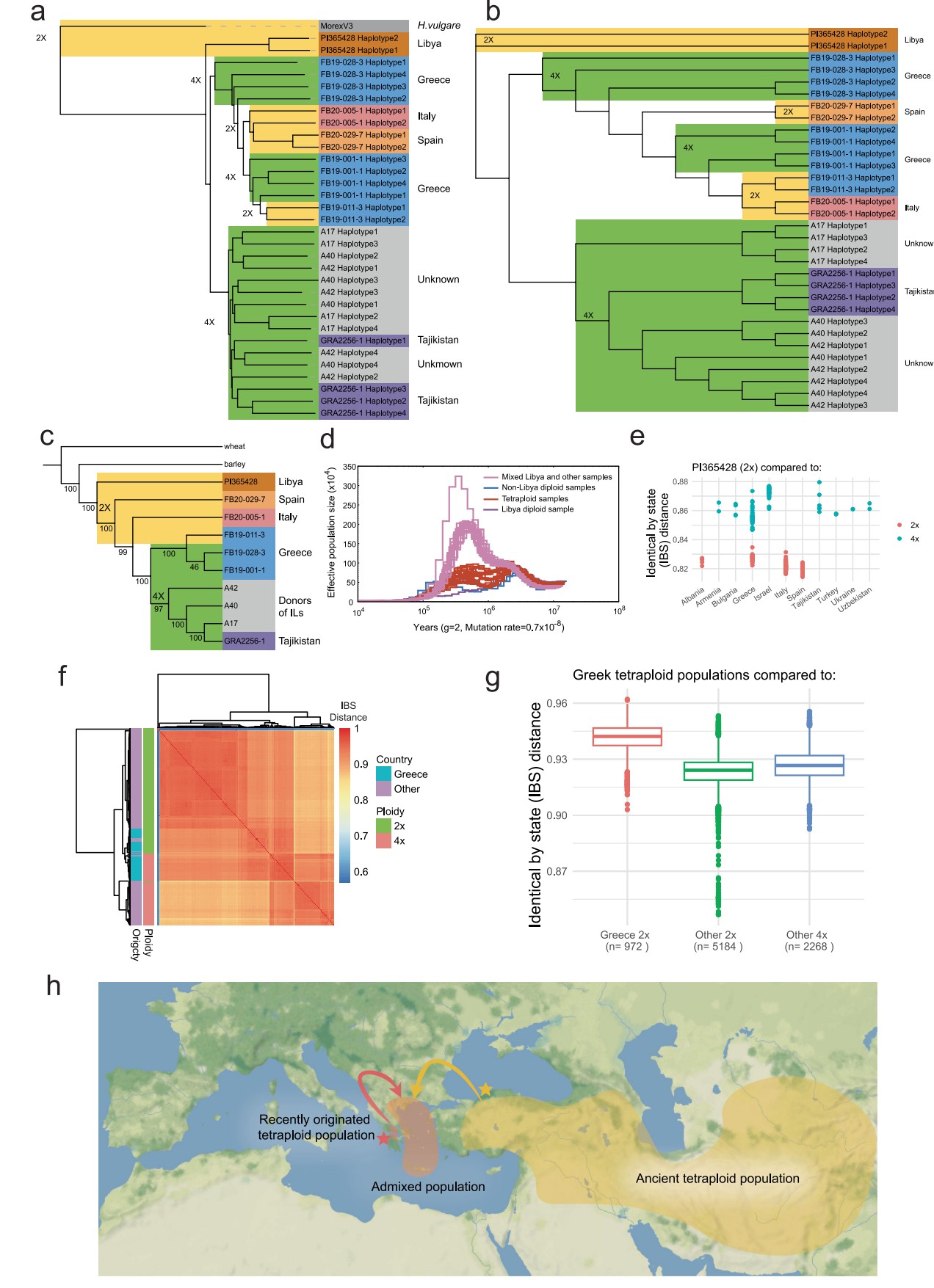

**Extended Data Fig. 9 |** See next page for caption.

**Extended Data Fig. 9 | Relationships within *H. bulbosum*. (a)** Neighbor-joining tree constructed from a genome-wide SNP matrix (variant calling by deepvariant long reads pipeline) of 32 *H. bulbosum* haplotypes and one *H. vulgare* genome. Geographic origins and ploidy are color-coded. **(b)** Neighbor-joining tree constructed from graph pangenome similarity from ODGI **(c)** Maximum likelihood (ML) tree constructed from the whole chloroplast genomes of 32 *H. bulbosum* individuals and one *H. vulgare* individual. The wheat chloroplast genome was included as an outgroup. **(d)** Population trajectory as inferred by PSMC of the Libyan diploid PI365428, diploids from other countries, autotetraploids, and synthetic heterozygotes between the Libyan diploid and other diploid and tetraploid haplotypes. **(e)** Genetic distance of PI365428 to other populations. **(f)** Heatmap showing the identity-by-state (IBS) distance matrix between 270 *H. bulbosum* and 3 *H. vulgare* genotypes. **(g)** Genetic distance of the Greek tetraploid populations. The central line represents the median; the lower and upper edges of the box correspond to the first (Q1) and third quartiles (Q3), respectively. The interquartile range (IQR) is defined as Q3 − Q1. Whiskers extend to the most extreme data points within 1.5 × IQR from the quartiles. Data points beyond this range are plotted individually as outliers. **(h)** The diagram of the hypothesis of at least two origins. The base map was created using tiles provided by Stadia Maps, styled by Stamen Design, and incorporating geographic data from OpenStreetMap contributors via OpenMapTiles. Map data © OpenStreetMap contributors, licensed under the Open Database License; visual design © Stamen Design, licensed under CC BY 3.0; map tiles © Stadia Maps, licensed under CC BY 4.0.

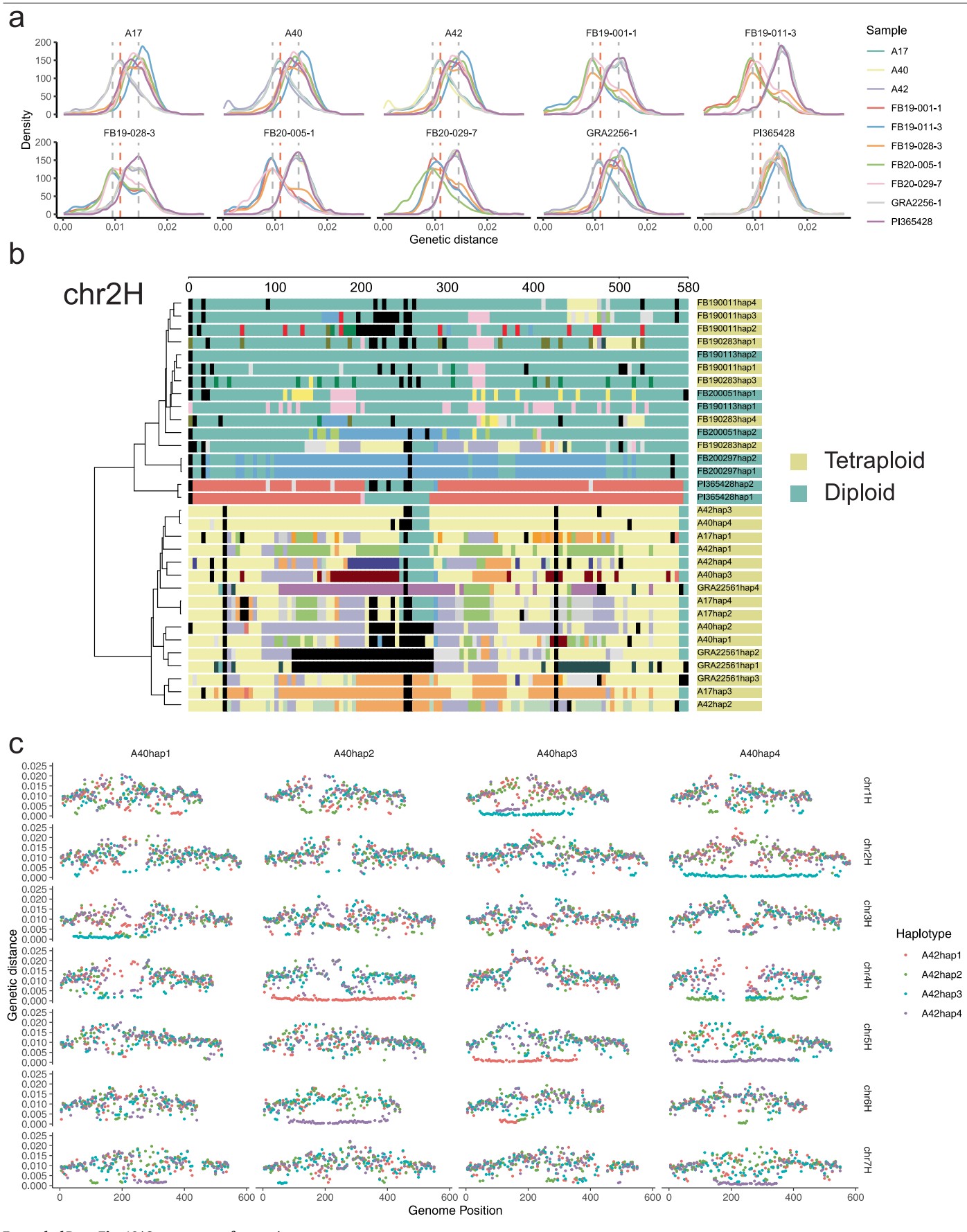

**Extended Data Fig. 10** | See next page for caption.

**Extended Data Fig. 10 | Haplotype definition with IntroBlocker.**
(**a**) Distribution of pairwise genetic distances in 5 Mb bins. Each subpanel shows the pairwise comparison of the haplotypes of one genome to the haplotypes of all other genomes. The distribution of pairwise distance revealed two major peaks (gray dashed line), one at 0.0095 variants per bp, and the other at 0.0145 variants per bp. A threshold of 0.011 was used to differentiate between these two peaks in IntroBlocker (red dashed line). (**b**) Graphical genotypes on chromosomes 2H and 4H. Regions of identical colors are assigned to the same haplotype at the chosen identity threshold (0.011). (**c**) Genetic distances between haplotypes of A40 and A42. Pairwise genetic distance between the haplotypes of A40 and A42 were plotted along the genome in 5 Mb windows.

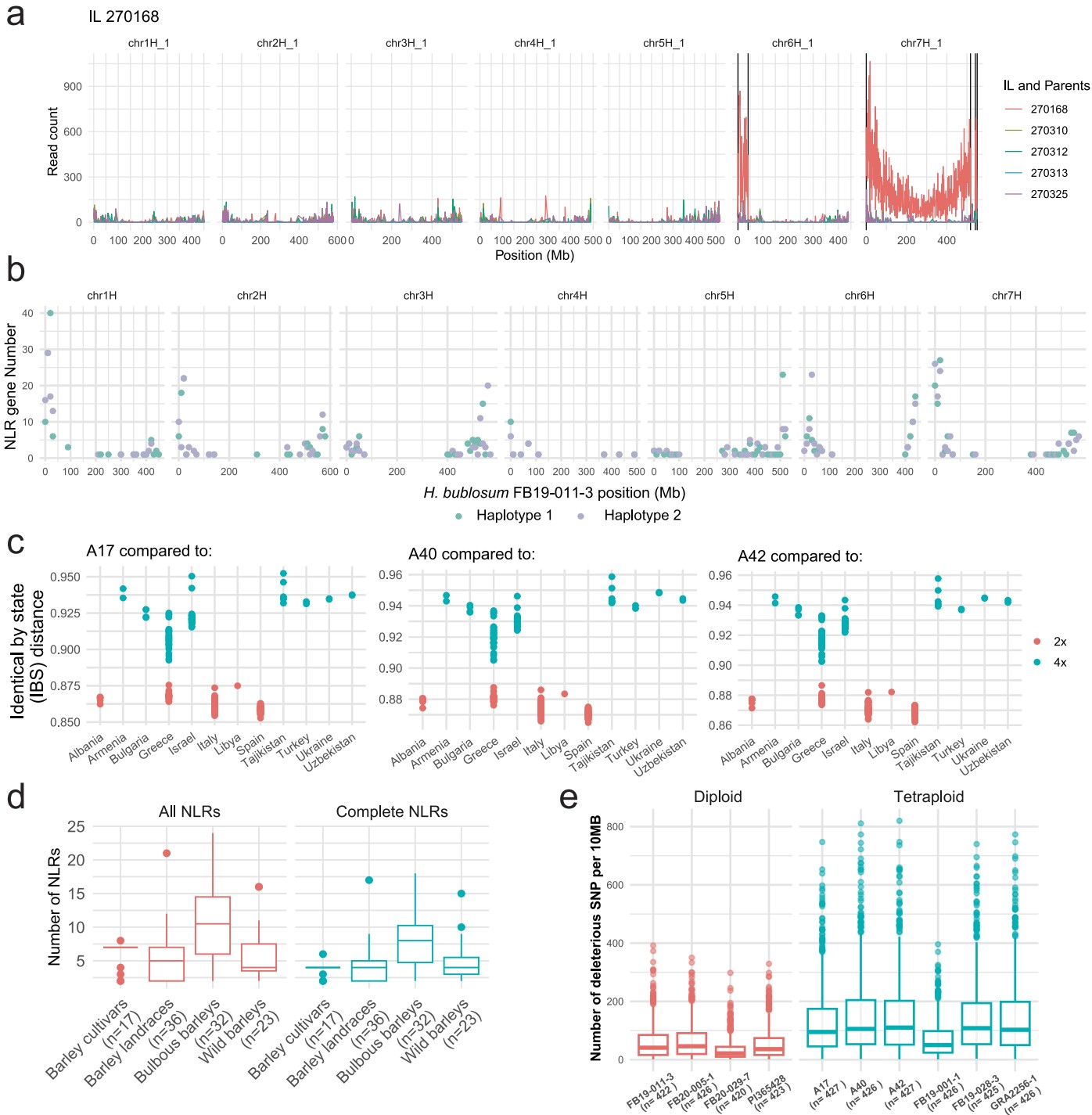

**Extended Data Fig. 11 | Genomic characterization of *H. vulgare-H. bulbosum* introgression lines and their *H. bulbosum* parents.** (**a**) Identification and mapping of alien chromatin in the introgression line sample 270147. (**b**) Genome-wide distribution of resistance genes homolog of the NLR family in FB19-011-3. NLR numbers were tabulated in 1 Mb windows along the genome. (**c**) Identity-by-descent between the three donor genotypes A17, A40 and A42 and other populations point to a central Asian origin. (**d**) NLR numbers in the Ryd4 interval in *H. bulbosum* haplotypes and selected *H. vulgare* genomes. (**e**) Distribution of putative deleterious SNPs in *H. bulbosum* genomes. In d and e, the central line represents the median; the lower and upper edges of the box correspond to the first (Q1) and third quartiles (Q3), respectively. The interquartile range (IQR) is defined as Q3 − Q1. Whiskers extend to the most extreme data points within 1.5 × IQR from the quartiles. Data points beyond this range are plotted individually as outliers.

# Reporting Summary

Please do not complete any field with "not applicable" or n/a.  Refer to the help text for what text to use if an item is not relevant to your study.
For final submission: please carefully check your responses for accuracy; you will not be able to make changes later.

## Statistics

For all statistical analyses, confirm that the following items are present in the figure legend, table legend, main text, or Methods section.

| n/a | Confirmed | |
|---|---|---|
| ☐ | ☒ | The exact sample size (*n*) for each experimental group/condition, given as a discrete number and unit of measurement |
| ☐ | ☒ | A statement on whether measurements were taken from distinct samples or whether the same sample was measured repeatedly |
| ☐ | ☒ | The statistical test(s) used AND whether they are one- or two-sided<br>*Only common tests should be described solely by name; describe more complex techniques in the Methods section.* |
| ☒ | ☐ | A description of all covariates tested |
| ☐ | ☒ | A description of any assumptions or corrections, such as tests of normality and adjustment for multiple comparisons |
| ☐ | ☒ | A full description of the statistical parameters including central tendency (e.g. means) or other basic estimates (e.g. regression coefficient) AND variation (e.g. standard deviation) or associated estimates of uncertainty (e.g. confidence intervals) |
| ☒ | ☐ | For null hypothesis testing, the test statistic (e.g. $F$, $t$, $r$) with confidence intervals, effect sizes, degrees of freedom and $P$ value noted<br>*Give P values as exact values whenever suitable.* |
| ☒ | ☐ | For Bayesian analysis, information on the choice of priors and Markov chain Monte Carlo settings |
| ☒ | ☐ | For hierarchical and complex designs, identification of the appropriate level for tests and full reporting of outcomes |
| ☒ | ☐ | Estimates of effect sizes (e.g. Cohen's *d*, Pearson's *r*), indicating how they were calculated |

*Our web collection on statistics for biologists contains articles on many of the points above.*

## Software and code

Policy information about availability of computer code

| Data collection | Data collection used proprietary software of PacBio and Illumina for raw data processing. |
|---|---|
| Data analysis | Multiple published software packages were used in the analysis including: ADMIXTURE (v1.3.0), ASMAP(v1.0-2), Augustus (v3.3.3), Badread (v0.4.1), BBMap (v37.93), BCFtools (v1.9), BEDTools (v2.30.0), BLAST (v2.13.0), bwa (v0.7.17-r1188), Canu (v2.1.1), CD-HIT, cutadapt (v1.15), DeepVariant (v1.6.0), EVidenceModeller (v2.1.0), Extensive de-novo TE Annotator (EDTA) (v1.9.0), FASTME (v2.1.5), FastTree (v2.1.11), GENESPACE (v1.2.3), GeSeq (https://chlorobox.mpimp-golm.mpg.de/geseq.html), gfatools (v0.4-r179), glnexus_cli (v1.4.1), GMAP (v2018-07-04), hifiasm (v0.13-r308), IntroBlocker (v1.0.0), IQ-TREE (v2.2.0-beta), iTOL (https://itol.embl.de/) , LASTZ (v1.04.03), MAFFT (v7.490), Merqury (v1.3), mgcv (v1.8-42), Mikado (v2.3.4), Minigraph-Cactus pangenome pipeline (v2.6.5), minimap2 (v2.24-r1122 or 2.26-r1175), miniprot (0.11-r234), MMSeq2 (v2), Mummer4 (v4.0.0beta2), MUSCLE, NLR annotator (v2), Novosort (v3.09.01), ODGI (v0.8.6-2), PASA (v2.5.3), pbmm2 (v1.13.1), PGGB (v0.5.3), PLINK (v1.90b6.9), PSMC (V0.6.5-r67), RagTag (v2.1.0), SAMtools (v1.16.1), seqkit (v0.9.1), SIFT, SNPrelate(v1.16.0) splitfa, STAR (v2.7.8a), StringTie (v2.1.5), svim (v1.4.2), SyRI (v1.6), TRITEX pipeline (https://tritexassembly.bitbucket.io/), vg (v1.49.0), wfmash (v0.8.2), wgsim (v0.3.1-r13). Genome assembly and data analysis (https://github.com/jia-wu-feng/Pan_Bulbosum), Gene projection (https://github.com/GeorgHaberer/gene_projection/tree/main/panhordeum), Analyis of orthologous groups (https://github.com/PGSB-HMGU/hbulbosum/tree/main) |

For manuscripts utilizing custom algorithms or software that are central to the research but not yet described in published literature, software must be made available to editors and reviewers. We strongly encourage code deposition in a community repository (e.g. GitHub). See the Nature Portfolio guidelines for submitting code & software for further information.

## Data

Policy information about availability of data

All manuscripts must include a data availability statement. This statement should provide the following information, where applicable:

- Accession codes, unique identifiers, or web links for publicly available datasets
- A description of any restrictions on data availability
- For clinical datasets or third party data, please ensure that the statement adheres to our policy

All the sequence data collected in this study have been deposited at the European Nucleotide Archive (ENA) under BioProjects PRJEB65276 (Genome assemblies, transcriptome Illumina data, GBS data) and PRJEB65918 (pollen nuclei single-cell sequencing data). Accession codes for individual genotypes are listed in supplementary tables: Supplementary Table 1, 2 and 7 (pangenome assemblies and associated raw data), Supplementary Table 6 (pollen single-nuclei sequencing data), Supplementary Table 12 (GBS). The assemblies and annotations are available for download from https://galaxy-web.ipk-gatersleben.de/libraries/folders/Fb701da857886499b. The variant matrix and pangenome graphs were deposited under a DOI (http://dx.doi.org/10.5447/ipk/2025/4) at the Plant Genomics and Phenomics Research Data Repository. TREP database (https://trep-db.uzh.ch/)

## Research involving human participants, their data, or biological material

Policy information about studies with human participants or human data. See also policy information about sex, gender (identity/presentation), and sexual orientation and race, ethnicity and racism.

| | |
|---|---|
| Reporting on sex and gender | not applicable |
| Reporting on race, ethnicity, or other socially relevant groupings | not applicable |
| Population characteristics | not applicable |
| Recruitment | not applicable |
| Ethics oversight | *Identify the organization(s) that approved the study protocol.* |

Note that full information on the approval of the study protocol must also be provided in the manuscript.

# Field-specific reporting

Please select the one below that is the best fit for your research. If you are not sure, read the appropriate sections before making your selection.

☒ Life sciences          ☐ Behavioural & social sciences          ☐ Ecological, evolutionary & environmental sciences

For a reference copy of the document with all sections, see nature.com/documents/nr-reporting-summary-flat.pdf

# Life sciences study design

All studies must disclose on these points even when the disclosure is negative.

| | |
|---|---|
| Sample size | Population structure of a diversity panel was analyzed. Representative accessions from different populations, geographic origins and ploidy levels were chosen. |
| Data exclusions | No data were excluded. |
| Replication | DNA/RNA was extracted from verified clones of the same genotypes. |
| Randomization | Genome assembly and analysis were conducted on a individual H. bulbosum plant, thus randomization is not necessary. |
| Blinding | Genome assembly and analysis were conducted on a individual H. bulbosum plant, thus randomization is not necessary. |

# Reporting for specific materials, systems and methods

We require information from authors about some types of materials, experimental systems and methods used in many studies. Here, indicate whether each material, system or method listed is relevant to your study. If you are not sure if a list item applies to your research, read the appropriate section before selecting a response.

## Materials & experimental systems

| n/a | Involved in the study |
|-----|----------------------|
| ☒ | Antibodies |
| ☒ | Eukaryotic cell lines |
| ☒ | Palaeontology and archaeology |
| ☒ | Animals and other organisms |
| ☒ | Clinical data |
| ☒ | Dual use research of concern |
| ☐ ☒ | Plants |

## Methods

| n/a | Involved in the study |
|-----|----------------------|
| ☒ | ChIP-seq |
| ☒ | Flow cytometry |
| ☒ | MRI-based neuroimaging |

# Dual use research of concern

Policy information about dual use research of concern

## Hazards

Could the accidental, deliberate or reckless misuse of agents or technologies generated in the work, or the application of information presented in the manuscript, pose a threat to:

| No | Yes | |
|----|-----|---|
| ☒ | ☐ | Public health |
| ☒ | ☐ | National security |
| ☒ | ☐ | Crops and/or livestock |
| ☒ | ☐ | Ecosystems |
| ☒ | ☐ | Any other significant area |

## Experiments of concern

Does the work involve any of these experiments of concern:

| No | Yes | |
|----|-----|---|
| ☒ | ☐ | Demonstrate how to render a vaccine ineffective |
| ☒ | ☐ | Confer resistance to therapeutically useful antibiotics or antiviral agents |
| ☒ | ☐ | Enhance the virulence of a pathogen or render a nonpathogen virulent |
| ☒ | ☐ | Increase transmissibility of a pathogen |
| ☒ | ☐ | Alter the host range of a pathogen |
| ☒ | ☐ | Enable evasion of diagnostic/detection modalities |
| ☒ | ☐ | Enable the weaponization of a biological agent or toxin |
| ☒ | ☐ | Any other potentially harmful combination of experiments and agents |

# Plants

| | |
|---|---|
| Seed stocks | The genebanks of IPK Gatersleben and USDA-ARS for provided seeds for this study. Collecting and exchange of seeds followed the regulations of the respective countries and Nagoya agreements for material exchange. One herbarium voucher for each population was deposited in the herbarium of the IPK Gatersleben (GAT). |
| Novel plant genotypes | No novel seed stock were generated. |
| Authentication | NA |

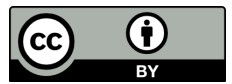

