## [Peer Review File · Nature]

A haplotype-resolved pangenome of the barley wild relative *Hordeum bulbosum*

Corresponding Author: Dr Martin Mascher

Version 0:

Reviewer comments:

Referee #1

(Remarks to the Author)

The manuscript by Feng et al claims to present a pangenome in a barley wild relative. However, the data is just a set of 10 genome assemblies and there is no description of a pangenome. As such the title of the manuscript is misleading, as well as the abstract and section on pan genome development. There is little biology presented, just standard information on TEs and some link to virus resistance for which the pangenome is not required.

The introduction presents much on the use of wild relatives as a source of diversity and novel alleles for crop development, but is very focussed on barley and wheat, with no investigation into other crops. It is also not clear from this how the pangenome can help with the characterisation of the introgressions and how it will assist in overcoming the current barriers that there are with using introgressions from wild relatives.

In the description of the first assembly of the diploid *H. bulbosum* the authors do not clearly state why this line was selected. N50 needs to be clear in the main text, as N50, for readers to relate to and needs to be compared to recent and relevant high quality assemblies. The papers that were selected for comparison are not appropriate – one is from the authors and the others is a low impact factor – how does it compare to current genome assemblies? It is not clear whether the shorter than expected assembly is due to errors in assembly or the cytometric estimates? Why were the single pollen nuclei mapped to only one haplotype – they should be mapped to both.

The “a haplotype resolved pangenome of *H. bulbosum*” does not represent a panegnome, it just represents a series of genome assemblies. The earth biogenome project has a low requirement for assemblies compared to what is published in Nature and as such is not a good benchmark. The selection of the lines for sequencing is not clear – how representative are they of the global diversity and is 10 really enough for a pan genome if it was constructed? The main issue is that this is not a pan genome. The annotation models were projectd from the initial assembly which does not provide information about the novel genes. Standard for pan genomes now is graph based. There is no pan genome here, there is not even comparison about what is novel in the different lines and linking to biology.

The population genetic analysis is concerning. The analysis is based on single read 100 bp short read data which is not useful for mapping to diverse polyploid genomes. It is mapped to a single reference (what is the point of a pan genome if only mapping to a single reference as all the novel information in the other genomes will not be captured) and it is mapped using BWA which tends to force sequences into positions, what was the unique mapping rate? There is no information as to what was done with reads that map to >1 position (highly likely given the data) and nothing seems to have been captured from reads that don't map. It is a very poor analysis and why break up the long read assemblies? This will lose highly valuable information. The fact that the Greek tetraploids are closer to sympatric diploids is likely just due to crossing. All inferences are based on very poor data which may not have mapped correctly and only SNPs, which are limited and question why a pan genome is claimed to have been produced when it is not used here.

The research on the introgression lines does not need a pan genome. They have the assembly of the line used for the intorgression and so if there is increased resistance then the gene conferring this will be in that assembly and so the pan genome is of no value here.

The discussion again states the pangenome is of value, but there is no pangenome presented here, so the discussion needs to be relevant to the data presented.

Line 141 of methods – essentially is not a good enough measure – is it exactly the same as before, if not state what is different

Variant discovery in haplotype resolved pangenomes – why not do whole haplotype comparison?

Single copy pangenome – if this is developed it needs to be described in detail for validation and it hasn't been – there is no pan genome described here, and if there is one developed then it needs to be graph based as is now the standard.

(Remarks on code availability)

I am not an expert and so would not be able to know if accurate

Referee #2

(Remarks to the Author)

This study presents a pangenome of bulbous barley comprising ten haplotype-resolved genome sequence assemblies of diploid and autotetraploid accessions. They show that the differential amplification of transposable elements after barley and *H. bulbosum* diverged from each other is responsible for genome size differences between them and that autotetraploids arose twice.

The availability of chromosome scale genome sequences for crop wild relatives is proving to be very useful for introgression breeding and this useful resource will go a long way for barley improvement.

The work is novel and hugely significant paving the way for pangenomes of other crop wild relatives such as wheat.

The methodology used is appropriate and the quality of the data and presentation is excellent.

Study uses statistics appropriately where required and presentation of the data and its analysis across, figures, tables, extended figures and supplementary items is well thought-out and comprehensive.

In my view this study is robust and reliable. It has been summarised well in the abstract, has an excellent introduction and the main article describes all the results clearly with appropriate citations through the main article and the methodology.

Please find some suggestions to improve the manuscript below:

1. Supplementary Figure 1. The x and y axis labels seem to be gibberish. Please correct.
2. Extended Figure 1a. Does Bionano optical mapping need to be added into the flowchart somewhere since it mentions it on line 81?
3. Sentence starting on line 88 is a little bit confusing. Have the metrics for the pseudomolecule assembly been introduced at this point? I don't think so. So not sure what 7.37 Gb is referring to. Is that the final assembly size and larger than the contig assembly (6.78 Gb)? And is 10.36 Gb the flow-cytometric value of *H. vulgare*? Because 80% of that is higher than 7.37 Gb.
4. Supplementary Tables might need to be reformatted so all columns are aligned with each other. Looks like they have been converted from an excel sheet into word but it was hard to follow those with multiple columns and rows where the columns are split across pages, eg. Supp Table 12.
5. Line 97: Is that 50 SNPs or 50% SNPs?
6. Line 147. You mean the diploid assemblies had an average size of 7.4Gb not the pseudomolecules, correct?
7. Does Figure 3f indicate a relatively higher density of more recent insertion of Gypsy elements in the proximal region (perhaps centromeric) in *H. bulbosum* compared to *H. vulgare*?
8. Figure 5d. Might want to mention somewhere in the figure or figure legend that the schema is of Chr 3H.

(Remarks on code availability)

The code seems to be reproducible and a usable resource for the community. There is a README file but I have not tried to install and run the code.

Referee #3

(Remarks to the Author)

In the manuscript entitled "A haplotype-resolved pangenome of the barley wild relative *Hordeum bulbosum*", Feng and co-authors describe the assembly and analysis of the bulbous barley pangenome, showcasing its utility as a bank of wild alleles that could be introgressed into domesticated barley. The authors assembled, phased and annotated two haplotypes of a diploid genotype and further constructed genomic resources for other diploid and tetraploid species. I commend their sampling and assembly across a wide geographic range and ploidy levels to capture haplotypes present in the population, but I believe there are two major shortcomings in the pan-genome analysis that I discuss below.

First, unfortunately I don't believe this represents a fully realized pan-genome resource of bulbous barley, given that gene models are lifted over from one phased, annotated diploid assembly. The authors argue that there is an untapped reservoir of allelic variation present in bulbous barley that domesticated barley could benefit from; but if that is the case, then only genes present in that diploid genome can be explored and characterized in other genomes. Analysis was supplemented with genes from barley, but gene comparative genomics and presence-absence variation will be undersampled and limited to coding sequences in the diploid, and a large amount of sub/neofunctionalization present in the tetraploids will be overlooked. Furthermore, the authors use long-read sequencing and HiC to construct reference quality (based on LAI metrics) assemblies for multiple genomes, but break them up again using simulated reads to compare samples to a single reference. While pan-genome analysis techniques are relatively new, there are more nuanced approaches (minigraph-cactus for example) than this that leverage multiple reference genomes.

Secondly (and this supports the argument above), when reading about the amount of diversity present across the pan-genome, I was surprised how variants were reported. While the overall SNP rate is not provided among references, inspection of extended data figure 6 shows that in the distal regions of some chromosomes, there are approximately 150,000 to 180,000 SNPs per Mb, which I assume is the sum total of SNPs across all 32 haplotypes. Do these represent unique, repeat masked regions of the genome? Are these extracted from the merged vcf, or counted from each individual alignment/SNP calls? Is this mainly driven by tetraploid samples? It is difficult to evaluate what 180,000 SNPs per Mb means in relation to overall diversity. Assuming these are high-quality SNPs, counted from a merged vcf where repeat regions are masked (to avoid inflated counts), then there is a considerable amount of diversity present in the population (and this is ignoring small and large structural variants), which necessitates needing multiple de novo annotated references to capture it. Differences in ploidy across the population compounds the problem.

Additional comments:

-What is the evidence supporting autopolyploidy in *H. bulbosum*? Inspection of the kmer frequency histograms in supplemental figure 1 did not suggest the genotypes collected are autopolyploids (although evaluating this was a challenge considering the every subplot's title, axes and legends were in wingdings font and were completely illegible) Kmer histogram plots for autotetraploids typically show kmer peaks at 3-4X coverage because, as autopolyploids are typically derived from intra-chromosomal duplication, all chromosomes are very similar to each other, and thus share haplotypes. The kmer frequency subplots shown in figure 1 show two main peaks, which I would interpret as: heterozygous diploids or allotetraploids. While I can see small shoulder peaks in some of the panels (indicative of tetraploidy), the bulk of the kmers are observed within the 1X (unique haplotype) range. This seems at odds with autopolyploidy where high similarity among chromosomes enables random meiotic chromosome pairs to occur.

-I found Figure 4 difficult to interpret and not necessarily supportive of claims made in the manuscript. First (and this is a straightforward change), but the colors used for Spain and Greece in panel B are so similar, it makes distinguishing between them difficult. Second, the authors should showcase how Figure 4C supports the hypothesis that autopolyploids arose twice, because it appears from the phylogenetic tree that polyploidy arose once and is shared between Group 1, Israel and Greece genotypes. Furthermore, it is also difficult to compare between panels B and C because a different number of labels/countries are used.

Minor comments

L74- Giving the assembly size here lacks context. Unless the reader is already familiar with bulbous barley, they would not understand if this represents a substantial capture of the total genome size.

L75-76- If comparing to other genomes, give the stats so the reader does not have to look them up.

Chromosome 4H- IBD section that spans basically the entire chromosome- how did you map short-read HiC to a region that is completely duplicated?

L97- 50% SNPs? This needs to be better described because it is confusing until you read the methods later in the manuscript.

1.1% - what is the total number of SNPs? Are these clustered at a particular position? or are they interspersed across the genome? Interspersed is fine, but clustered could represent a mis-assembly. Would be nice to see these SNPs paired with chromosome coverage along each chromosome.

L112- report this data per haplotype as well.

L119- These values are not not similar. An LAI value > 20 is considered high-quality while 14.7 is simply adequate for a reference genome.

L192-196: this should be scaled in terms of repeats per Mb rather than totals.

In the methods section, coverages of HiC and HiFi and pollen libraries should be reported, they are directly relevant for determining the quality of the assembly.

tetraploid genome assembly- how exactly did the multi-count contigs get placed?

GENESPACE software used to compare gene synteny among references does not appear to be properly referenced

(Remarks on code availability)

Version 1:

Reviewer comments:

Referee #1

(Remarks to the Author)

The authors have made some small attempts to create a pan genome, they did run minigraph-cactus but then said that interpretation is complex. That is true, as these analyses are complex, however but it looks like they didn't try to optimise. At present none of the analysis in the manuscript relates to the pangenome so the title and abstract still remain misleading. Making and analysing graph pangenomes is challenging and the authors should not be claiming they have done so when it looks like they just tried putting the data through the software without optimisation/cleaning, then claimed it was too complex and did other analysis of the haplotypes. This is still not pan genome analysis and trying to change the definition to fit their data is wrong.

The number of genomes assembled is still low even when looking at assembled haplotypes and especially when these are not fully analysed as pan genomes. This is not addressed by stating they have resolved multiple haplotypes as the number is still low.

There is still very little biology presented, this is coming across very much as a resource of some genomes rather than having any biological relevance. The authors do not argue well the case for the pan genome for the virus resistance, there is very little information provided and this is the only real biology presented, which is very poor for a pan genome paper, however, as already stated this is not a pan genome paper. The authors have just moved previous analysis round in the manuscript and the interval for the BYDV is the same, showing no new analysis has been done because there was no real pan genome construction.

The authors have selectively referred to genomes for the comparison which are still not appropriate - they can clearly find things much more related to bulbosa than sugarcane and potato than barley. They are clearly trying to find assemblies more close to theirs in stats rather than what is realistic.

The main issue is that this is still not a pan genome, the authors have made a poor attempt with no optimisation of producing a pan genome and it seems to be a more resource than having a significant biological story

(Remarks on code availability)

Referee #2

(Remarks to the Author)

This study presents a pangenome of bulbous barley comprising ten annotated, haplotype-resolved genome sequence assemblies of diploid and autotetraploid accessions. They show that the differential amplification of transposable elements after barley and *H. bulbosum* diverged from each other is responsible for genome size differences between them and that autotetraploids arose twice. Additionally, the authors demonstrate the utility of the pangenome in introgression breeding of barley with this wild relative species.

Having reviewed all the reviewer comments and the responses of the authors to those, I am satisfied that the authors have made significant additions to the study's methodology, analysis and reported outcomes, including improvements to the manuscript's text and figures.

The work reported is now much clearer and robust and as such I recommend its acceptance for publication.

(Remarks on code availability)

Referee #3

(Remarks to the Author)

First, I want to commend the authors for their revisions that have greatly improved the manuscript. Their improved annotation strategy and analysis has elevated this work to a proper 'pan-genome' analysis of *H. bulbosum*. The analyses are well-thoughtout and considerate. I find little to critique in terms of technical analyses, with good care to show that the diploid genome assembly was properly phased.

My only gripe in terms of technical analysis and interpretation relates to the characterization of crop-wild introgressions. Authors completed a competitive mapping analysis of GBS data to a synthetic diploid of MorexV3 and *H. bulbosum* (genotype FB19-011-3). Regions where read-mapping skewed toward wild *bulbosum* were inferred to have been introgressed. While that is certainly possible, there is not enough information given (map quality/ CIGAR string parsing/etc..) to assess the validity of that claim (Also, Line 1122 refers to Supp Figure 13, but I think the authors mean Supp table 13).

While introgression of bulbosum sequence into the genome could be found using this competitive mapping strategy, additional information is needed to understand how close the sequences in the GBS data match wild barley (kmer masking/read mapping metrics, etc.). As written, the analysis simply shows these regions map better to wild barley than morex, but could still be quite diverged and derived from another donor.

When considering this manuscript, I think it is important to contrast this work with the other wild barley pan-genome that was recently published in Nature (Jayakodi et al. 2024) by a number of the same authors. While this work has generated new genome assemblies and a pan-genome for a different wild species, there is some overlap between the two papers, with a number of similar analyses being completed. This is unavoidable considering the nature of pan-genome analysis, but the title: "Structural variation in the pangenome of wild and domesticated barley" lends itself to stricter scrutiny. I found the previous manuscript that constructs a pan-genome using the wild progenitor of barley (*H. vulgare* subsp. *spontaneum*), quite comprehensive and thus wonder if there are enough impactful findings that the readers of Nature would be interested in another wild barley pan-genome months later, particularly with such a heavy focus on the genome assembly itself (phasing/TE content, etc..)

(Remarks on code availability)

I did not attempt to run the code but I did look over it and appeared to be in good order.

Version 2:

Reviewer comments:

Referee #3

(Remarks to the Author)

With regards to the authors' response to referee #1:

I agree largely with the authors, a pangenome graph is not the sole benchmark for a pangenome study. In my opinion, a 'pan-genome' is intended to encompass the majority of genomic variation of a species, therefore a pangenome study is one that leverages multiple references to answer a biological question. Graphs are merely a way to organize that information succinctly, and are not always useful. The PCA plot showing where references are in relation of total diversity shows the full suite of variation is captured here. The only time I would argue against multiple references being a pangenome is when only liftover annotations are involved, as that involves projecting the same linear reference over and over onto new coordinates. Further, stating that 32 haplotypes isn't 'enough' is a false premise.. haplotype diversity is a species feature of population genomics. In another population, 32 could easily achieve full saturation.

Although, the main issue is that once you state that you have generated a graph using minigraph-cactus for example, the reader is going to expect it to be used. The pollen mapping does show that haplotype specific variants could be called, but usage of the full-graph is limited (my only takeaway from supp figures 16-22 is generally where there are gaps in the graph). In my opinion, the more interesting pangenome analysis are the local graphs related to the disease resistance loci. However the graphs in supplemental figures 35-36 are difficult to digest, but it is understood that bulbosum shows more diversity. I think it would go a long way to show how much variation has been captured in the pangenome at these loci if the heatmaps were fully annotated (colors or bars to differentiate diploids/tetraploids/origin/etc..). Additionally, another opportunity to use a localized, interpretable graph (or sequenceTubeMap) would be to show the RecA locus, how hap4 in A17 differs from others and that having multiple references helped narrow down the candidate list/locus.

Response to referee #3's comments

I am mostly satisfied with the response the authors provide with regards to the introgression. Although I would push back against the statement that: "if the introgression originated from another species, then reads would map equally well to both genomes". That is not necessarily the case, divergence between the reference genomes and the unknown donor could still easily bias the read mapping. But given that bulbosum is the only known species that vulgare can hybridize with, it seems likely that bulbosum is the donor. I also appreciate the text regarding how the mapping was conducted.

Lastly, I am satisfied with the contribution of this work to the field.

Line 92- misassemblies misspelled.

(Remarks on code availability)

Version 3:

Reviewer comments:

Referee #3

(Remarks to the Author)

I am satisfied with the edits made to the manuscript.

(Remarks on code availability)

Editor

While the referees find your work of some interest, they raise concerns about the strength of the novel conclusions that can be drawn at this stage and the appropriateness of the technical approach. We feel that these reservations are sufficiently important as to preclude publication of the present study in Nature.

Having said that, should you be able to address the reviewers' concerns about the technical quality of the pangenome comprehensively (comments from reviewers #1 and 3) and to provide further demonstration of the value of this resource (reviewer #1), we would be willing to look at a revised manuscript in the form of an appeal. I should stress, however, that we would be reluctant to trouble our referees again unless we thought that their comments and any editorial issues had been addressed in full, and we would understand if you prefer instead to now pursue publication of the work elsewhere.

Answer: Reviewers 1 and 3 voiced concerns about the lack of de novo gene annotations and pangenome graphs. We have addressed these concerns with additional analyses and additional newly generated data. Using these, we have re-annotated our genome sequence assemblies. We also used pangenome graphs as references for our population genetics analyses. The details are described below in the responses to the referees' comments.

To demonstrate the usefulness of our resource, we have generated and analyzed genome assemblies of nine additional introgression lines and studied the diversity of resistance gene loci using graph-based approaches. We have also revised the discussion to better describe the applicability of our resource in barley introgression breeding.

We have uploaded along the revised manuscript a companion paper by Blattner et al. on the phylogeography of *H. bulbosum*.

Referee #1

The manuscript by Feng et al claims to present a pangenome in a barley wild relative. However, the data is just a set of 10 genome assemblies and there is no description of a pangenome. As such the title of the manuscript is misleading, as well as the abstract and section on pan genome development.

Answer: To address the referee's concern about the lack of an integrative analysis of genome assemblies, we constructed genome-wide pangenome graphs using three complementary approaches: the single-copy pangenome, gene-based partitioning of the pangenome into core, shell and cloud compartments, and whole-genome pangenome graphs. The results are shown in Figure 2, Extended Data Fig. 6, Supplementary Data. Fig. 16-23, 35 and 36 .

To explore local variations in greater detail, we constructed graph pangenomes for *H. vulgare* and *H. bulbosum* in the *Ryd4* and *Mlo* regions. Both visual inspection of the graph structures (Supplementary Fig. 35 and 36) and summary statistics (Supplementary Table 15) revealed that bulbous barley exhibits higher complexity.

The reviewer expressed concern about the small number of assembled genomes. We note that while the number of genome sequences we assembled (10) may seem low at first glance, they amount to 32 haplotypes. We have amended the abstract to highlight this fact. Our analysis of single-copy pangenome complexity (Fig. 2g) showed that these haplotypes are more diverse than a similar number of inbred genomes of either wild or domesticated barley.

There is little biology presented, just standard information on TEs and some link to virus resistance for which the pangenome is not required.

Answer: We agree that our methods for TE annotation and the information they provide are “standard”. Indeed, the toolchain we employed, EDTA, first published in 2020, is now used by many plant genomic researchers. But the enquiries we conducted – into genome size evolution and its relationship to TE expansion – would not have been possible without haplotype-resolved long-read assemblies. Indeed, even in inbred barley, the TE space can be meaningfully analyzed at the whole-genome scale only with contiguous long-read assemblies.

Below, we discuss in greater detail the usefulness of our pangenome assemblies for understanding virus resistance and allelic variation at resistance gene loci. We have sequenced more introgression lines to make our resource immediately valuable for barley (pre-)breeders and developed markers to identify introgressions based on our single-copy pangenome (Supplementary Fig. 34).

The introduction presents much on the use of wild relatives as a source of diversity and novel alleles for crop development, but is very focussed on barley and wheat, with no investigation into other crops.

Answer: We have added this sentence to the introduction:

Crop-wild introgressions have been deployed successfully in tomato, potato and banana.

It is also not clear from this how the pangenome can help with the characterisation of the introgressions and how it will assist in overcoming the current barriers that there are with using introgressions from wild relatives.

Answer: We consider the following four results our most important findings with respect to introgression breeding:

1. We determined the number of deleterious variants and haplotype-specific single-copy core sequences in diploid and tetraploid *H. bulbosum* genomes and found candidate genes for a lethality factor that was co-transferred with a resistance gene. This illustrates the challenges associated with the use of introgression lines.
2. We identified the tetraploid genotype FB19-001-1 as a promising donor because it contains fewer deleterious variants and haplotype-specific single-copy core sequences than other tetraploids. Similarly, *H. bulbosum* genotypes harboring long IBD tracts may be promising donors because deleterious variants are expected to have purged in these regions. On the other hand, donors with large inversions (which suppress recombination) should be avoided.

3. We found that two introgression lines derived from *H. bulbosum* clone A17 contain chimeric haplotypes. This result could not have been obtained without genome sequence assemblies. It also suggests that we can introduce more haplotypes through direct hybridization of existing ILs, and the identification of these new haplotypes will rely on haplotype-resolved genomes.
4. Our analysis of NLR genes in *H. bulbosum* showed that mostly reside in highly recombinogenic regions that are accessible to introgression breeding.
- 5.

For these reasons, we are convinced that our pangenome resource will advance introgression breeding in barley and motivate further genomic studies of introgression lines and potential donor genotypes.

In the description of the first assembly of the diploid *H. bulbosum* the authors do not clearly state why this line was selected.

Answer: Our initial working hypothesis was that *H. bulbosum* may have originated in Greece as both diploid and tetraploid cytotypes occur in that country. We therefore chose a Greek diploid for our first genome assembly in the hope of selecting an “ancestral” genotypes. Our subsequent phylogenetic analyses based on sequence assemblies and genotyping-by-sequencing data painted a different picture of the phylogeography of the species (a Libyan genotype turned out to occupy a basal position in a phylogenetic tree).

N50 needs to be clear in the main text, as N50, for readers to relate to and needs to be compared to recent and relevant high quality assemblies.

Answer: We mention the N50 in the main text of the revised manuscript:

The contiguity and completeness of the contigs constructed from PacBio HiFi reads were promising: the assembled contigs amounted to 6.78 Gb of sequence and their N50 was 9.82 Mb.

Contig-level contiguity (N50: 7.7 Mb) was lower than in the diploid.

The papers that were selected for comparison are not appropriate – one is from the authors and the others is a low impact factor – how does it compare to current genome assemblies?

Answer: We have replaced the reference to inbred wheat with references to sugar cane and potato. Although the work was done by us, we still believe that the reference to barley is meaningful as this is the species most closely related to *H. bulbosum*.

It is not clear whether the shorter than expected assembly is due to errors in assembly or the cytometric estimates?

Answer: We have published a detailed inquiry (Navratilova et al. 2022, doi: 10.1111/pbi.13816) on the discrepancy between sequence-based and flow-cytometric genome size estimates in barley (*Hordeum vulgare*). This paper is referenced in the sentence

We have attributed this discrepancy in barley to unresolved repeat arrays in ribosomal loci and centromeric and interspersed satellites.

Why were the single pollen nuclei mapped to only one haplotype – they should be mapped to both.

Answer: We have explored two alternative mapping approaches: mapping to a diploid reference and mapping to a graph-based pangenome constructed from both parental haplotypes (FB19-011-3 haplotypes 1 and 2) using Minigraph-Cactus. The results were consistent to those based on read mappings to a single haplotype (Supplementary Fig. 6).

The “a haplotype resolved pangenome of *H. bulbosum*” does not represent a pangenome, it just represents a series of genome assemblies.

Answer: As described above, we have addressed this concern by constructing pangenome graphs using different approaches.

We note that the human pangenome consortium has also adopted a definition of the pangenome as a collection of genome sequence assemblies: “The pangenome contains 47 phased, diploid assemblies from a cohort of genetically diverse individual” (from Liao et al. 2023, doi: 10.1038/s41586-023-05896-x). We have also advocated for the adoption of this definition in a recent review article (Schreiber et al., 2024, doi: 10.1038/s41576-024-00691-4).

The earth biogenome project has a low requirement for assemblies compared to what is published in Nature and as such is not a good benchmark.

Answer: We have removed the reference to the Earth Biogenome project.

Recently, Nature published papers on sugar cane and potato, two crops with complex genomes: Tang et al. 2022, doi:10.1038/s41586-022-04822-x and Healey et al. 2024, doi: 0.1038/s41586-024-07231-4. The N50 of the sugar cane primary assembly (which covered only half the genome) was 12.6 Mb. The N50 of the potato assemblies of Tang et al. before collapsing heterozygous regions was 9.1 Mb. We refer to these papers in the sentence:

This level of contiguity is comparable to those achieved in barley (16.8 Mb), sugar cane (12.6 Mb) and potato (2.1 Mb and 9.1 Mb).

The assembly metrics of these species exceed indeed the requirements of the Earth Biogenome Project (megabase N50 contig continuity and chromosomal-scale scaffolding), but are similar to the ones we achieved for *H. bulbosum*.

The selection of the lines for sequencing is not clear – how representative are they of the global diversity and is 10 really enough for a pan genome if it was constructed?

Answer: We collected genotyping-by-sequencing data for more than 200 individuals from the entire distribution range of the species and conducted phylogeographic analyses with those

data. Based on these analyses, we selected diploid individuals from Greece, Spain, Italy and Libya and tetraploids from Greece and Western Asia.

We have uploaded along with our revised manuscript a companion paper on the phylogeography of *H. bulbosum*. This paper reports on several phylogenetic analyses that we cannot include in the present paper owing to length restrictions.

We have added PCA plots and a geographic map to Figure 2 to illustrate our approach for selecting diverse accessions.

The main issue is that this is not a pan genome. The annotation models were projected from the initial assembly which does not provide information about the novel genes.

Answer: We agree with the reviewer that a limitation of our initial gene projection approach was that we were not able to detect completely novel genes and gene structures in some of the lines. Nevertheless, our gene projections are capable to call and resolve copy number variations (such as tandem copied genes) as well as diverged gene models due to the mapping strategy applied (different rounds of mapping including multiple and low stringency mappings considered).

That said, to resolve the referee's concern about the lack of novel genes, we re-annotated our assemblies from scratch. We developed and applied an integrated gene annotation approach which performs de novo gene calling for all genome assemblies (including individual genomes in phased assemblies and subgenomes in tetraploid accessions) in this study. This gene prediction strategy made use of (i) newly generated RNAseq data for diploid and tetraploid accessions from multiple tissues, (ii) publicly available expression data for tetraploid accessions (Fuerst et al. 2023, doi: 10.1093/gbe/evac168), and (iii) trained ab initio gene finders capable of calling novel genes in individual genomes. A final gene model consolidation step (cross-species all-against-all mappings) was added to the pipeline to minimize false positive CNV and PAV callings in the pangenome analyses.

Based on the updated gene predictions, we constructed an orthologous framework (using OrthoFinder) across the annotations of all individual genotypes and thereby defined the core, shell and cloud genomes of *H. bulbosum* at different ploidy levels. The orthologous framework can be considered as a reference-agnostic, gene-centric pangenome.

Standard for pan genomes now is graph based.

Answer: We constructed pangenome graphs at different levels (genome-wide, gene-based, and at two resistance gene loci). We used the genome graph for diversity analysis, comparisons between diploid and autotetraploid genomes. A genome graph constructed from the two haplotypes of diploid FB19-011-3 was used to analyze single-nucleus pollen sequencing data of that genotype.

There is no pan genome here, there is not even comparison about what is novel in the different lines and linking to biology.

We consider the following findings to be the most important novel insights gleaned from our assemblies:

- Multiple origins of tetraploid cytotypes. Haplotype-resolved assemblies were crucial for genotype calling in a autotetraploid genome with four distinct haplotypes.
- Genomic dissection of a resistance gene locus using the assembly of the *H. bulbosum* donor and the introgression line, including the discovery of a sublethality factor through the comparison of multiple *H. bulbosum* genomes.
- Our comparison of genetic diversity and structural variation in *H. bulbosum* and *H. vulgare* – especially in resistance genes – supports the notion that introgression breeding is a promising approach in barley.

The population genetic analysis is concerning. The analysis is based on single read 100 bp short read data which is not useful for mapping to diverse polyploid genomes. It is mapped to a single reference (what is the point of a pan genome if only mapping to a single reference as all the novel information in the other genomes will not be captured) and it is mapped using BWA which tends to force sequences into positions, what was the unique mapping rate? There is no information as to what was done with reads that map to >1 position (highly likely given the data) and nothing seems to have been captured from reads that don't map. It is a very poor analysis and why break up the long read assemblies? This will lose highly valuable information. The fact that the Greek tetraploids are closer to sympatric diploids is likely just due to crossing. All inferences are based on very poor data which may not have mapped correctly and only SNPs, which are limited and question why a pan genome is claimed to have been produced when it is not used here.

Answer: We repeated our variant calling using simulated long reads (20 – 40 kb) and used the appropriate tools for variant calling. The quality information of read alignment is shown in the Supplementary Table 18. This is the relevant paragraph from the Methods section:

SNP, indels, and SVs were discovered in a panel of 33 sequence assemblies comprising the 32 assembled H. bulbosum haplotypes and the inbred genome of barley cv. Morex. Calling of SNP and indels was done with simulated long-read data using the FB19-011-3 haplotype 1 assembly as reference. For each haplotype/genome, 30x coverage and 20,000-40,000 length reads were simulated with Badread (v0.4.1). Read aligned to the haplotype 1 sequence assembly of FB19-011-3 with pbmm2 (<https://github.com/PacificBiosciences/pbmm2>). DeepVariant (v1.6.0) with the pretrained PacBio mode (--model_type PACBIO) was then used for variant calling of each accession, and all individual variants were merged using glxenus_cli (v1.4.1) with DeepVariant config file.

Note that we could not use the HiFi reads of *H. bulbosum* genotypes for variant calling without prior assembly because they did not come with phasing information. So it was necessary to assemble them into contigs first and then assign these to haplotypes with the help of Hi-C data.

We used the variant matrix derived from the simulated long reads to repeat the definition of haplotype blocks (with Introblocker, Figure 4f and Extended Data Fig. 9) and the inference of

population size trajectories (PSMC, Figure 4b-d and Extended Data Fig. 8d). We also constructed phylogenetic trees directly from pangenome graphs (Extended Data Fig. 8b).

Most of the results remain largely unchanged. A notable change is that the older tetraploid lineage is now estimated to have originated 2.23 million years ago.

We still find a close association between the Greek tetraploid FB19-001-1 and Greek diploids. Crossing or hybridizations fully explain the patterns we observed, as the other Greek tetraploids do not show higher affinity to sympatric diploids, so they must be some barriers to hybridization across ploidy levels.

We uploaded along with this revised manuscript a companion paper by Blattner et al. that shows that in the population of the newly evolved Greek tetraploid FB19-001-1 gene flow with the older stock of tetraploids occurred so that individuals show different levels of introgression.

The research on the introgression lines does not need a pan genome. They have the assembly of the line used for the introgression and so if there is increased resistance then the gene conferring this will be in that assembly and so the pan genome is of no value here.

Answer: To better connect the pangenome with the research on barley introgressions lines, we have generated assemblies for nine additional introgression lines and reorganized the section of the manuscript that describes the analysis of ILs and the *Ryd4* locus.

We believe that the pangenome has value in the study of introgression in general and the *Ryd4* locus in particular for the following reasons:

1. The pangenome revealed the large extent of structural variation at the *Ryd4* locus in both barley and *H. vulgare*. This pattern was evident in local pangenome graphs and was refined by in-depth, gene-base comparison of multiple genome sequences
2. Discovery of cross over events in two introgression lines that recombined haplotypes of the donor line A17. These cross-overs must have happened early in the development of the introgression lines. It could not have been discovered without the comparison of haplotype-resolved genome sequences.
3. Discovery of a candidate gene for a sublethality factor through the comparison of multiple barley and *H. bulbosum* genomes.
4. We have updated the genomic map of *H. bulbosum* introgression lines in barley. *H. bulbosum* genotype A17 is the donor of several independent ILs, which carry *H. bulbosum* chromatin in a barley background on multiple chromosomes. The availability of a chromosome-scale haplotype-resolved assembly of the donor genotype will be helpful for researchers working with these ILs.

The discussion again states the pangenome is of value, but there is no pangenome presented here, so the discussion needs to be relevant to the data presented.

Answer: We have revised the discussion to make clear how genome assemblies will help introgression breeding:

The availability of chromosome-scale genome sequences for crop wild relatives will be useful for introgression breeding. Genome sequence help design markers for tracing the inheritance of alien chromatin in a crop background, confirm genotype identity and match introgression lines to donor haplotypes. They are also indispensable in the search for candidate genes. We advocate for the construction of genome sequence of all donor genotypes in new introgression breeding programs.

Line 141 of methods – essentially is not a good enough measure – is it exactly the same as before, if not state what is different

Answer: We have revised this sentence:

Illumina NGS libraries were prepared from 96 single pollen nuclei using the ‘PicoPLEX Gold Single Cell DNA-Seq Kit’ and the ‘DNA Unique Dual Index Kit’ as described by the manufacturer (Takara Bio USA, Inc.).

Variant discovery in haplotype resolved pangenomes – why not do whole haplotype comparison?

Answer: We aligned entire haplotypes with GENESPACE (Fig. 2f). We also consider our Introblocker analysis (ancestral haplotype groups) and the differential labelling of physical chromosomes with FISH probes as whole-haplotype comparisons.

We have modified our pipeline for variant discovery to use longer simulated reads as haplotype-resolved alignment queries. We used the results of these alignment as the basis for the construction of the phylogenetic tree shown in Extended Data Fig. 8b.

Single copy pangenome – if this is developed it needs to be described in detail for validation and it hasn’t been – there is no pan genome described here, and if there is one developed then it needs to be graph based as is now the standard.

Answer: We introduced the single-copy pangenome in Jayakodi et al. 2020, doi: 10.1038/s41586-020-2947-8. We have expanded the analysis of the single-copy pangenome to illustrate to compare the extent of presence absence variation in autotetraploids (Fig. 2i).

Referee #1 (Remarks on code availability):

I am not an expert and so would not be able to know if accurate

Referee #2 (Remarks to the Author):

This study presents a pangenome of bulbous barley comprising ten haplotype-resolved genome sequence assemblies of diploid and autotetraploid accessions. They show that the differential amplification of transposable elements after barley and *H. bulbosum* diverged from each other is responsible for genome size differences between them and that autotetraploids arose twice.

The availability of chromosome scale genome sequences for crop wild relatives is proving to be very useful for introgression breeding and this useful resource will go a long way for barley improvement.

The work is novel and hugely significant paving the way for pangenomes of other crop wild relatives such as wheat.

The methodology used is appropriate and the quality of the data and presentation is excellent.

Study uses statistics appropriately where required and presentation of the data and its analysis across, figures, tables, extended figures and supplementary items is well thought-out and comprehensive.

In my view this study is robust and reliable. It has been summarised well in the abstract, has an excellent introduction and the main article describes all the results clearly with appropriate citations through the main article and the methodology.

Answer: We thank the reviewer for their encouraging remarks.

Please find some suggestions to improve the manuscript below:

1. Supplementary Figure 1. The x and y axis labels seem to be gibberish. Please correct.

Answer: This has been corrected.

2. Extended Figure 1a. Does Bionano optical mapping need to be added into the flowchart somewhere since it mentions it on line 81?

Answer: We have revised Extended Data Fig. 1 to mention the Bionano map at the appropriate steps.

3. Sentence starting on line 88 is a little bit confusing. Have the metrics for the pseudomolecule assembly been introduced at this point? I don't think so. So not sure what 7.37 Gb is referring to. Is that the final assembly size and larger than the contig assembly (6.78 Gb)? And is 10.36 Gb the flow-cytometric value of *H. vulgare*? Because 80% of that is higher than 7.37 Gb.

Answer: We have simplified this sentence:

The genome assembly was shorter than expected from flow-cytometric estimates (7.37 Gb vs. 8.68 Gb). In barley, we have attributed such discrepancies to unresolved repeat arrays in ribosomal loci and centromeric and interspersed satellites.

4. Supplementary Tables might need to be reformatted so all columns are aligned with each other. Looks like they have been converted from an excel sheet into word but it was hard to

follow those with multiple columns and rows where the columns are split across pages, eg. Supp Table 12.

Answer: Unfortunately, large supplementary are not properly displayed when converted into PDF format by Nature's manuscript submission system. The original XLSX files are available for download and correctly formatted. To access them, please click on "Source files".

5. Line 97: Is that 50 SNPs or 50% SNPs?

Answer: We have changed this sentence to

A total of 60,934 neighbouring SNP pairs (1.01%) had recombination fractions exceeding 50 %, indicating that they were potentially located in wrongly phased regions.

6. Line 147. You mean the diploid assemblies had an average size of 7.4Gb not the pseudomolecules, correct

Answer: Yes, we have corrected this. The revised sentence reads:

The average size of the diploid assemblies was 7.4 Gb.

7. Does Figure 3f indicate a relatively higher density of more recent insertion of Gypsy elements in the proximal region (perhaps centromeric) in *H. bulbosum* compared to *H. vulgare*?

Answer: Yes, this interpretation is correct. We have rephrased the corresponding sentence in the main text:

*Although smaller in size, the genome of *H. bulbosum* has more full-length LTR elements (61,916) than that of *H. vulgare* (46,727). This observation is in line with a higher number of TE insertions, mainly of Gypsy elements, in proximal regions of the *H. bulbosum* genome in the last 300,000 years.*

8. Figure 5d. Might want to mention somewhere in the figure or figure legend that the schema is of Chr 3H.

Answer: We have added "chr3H" to the figure title.

Referee #2 (Remarks on code availability):

The code seems to be reproducible and a usable resource for the community. There is a README file but I have not tried to install and run the code.

Referee #3

In the manuscript entitled "A haplotype-resolved pangenome of the barley wild relative *Hordeum bulbosum*", Feng and co-authors describe the assembly and analysis of the bulbous

barley pangenome, showcasing its utility as a bank of wild alleles that could be introgressed into domesticated barley. The authors assembled, phased and annotated two haplotypes of a diploid genotype and further constructed genomic resources for other diploid and tetraploid species. I commend their sampling and assembly across a wide geographic range and ploidies to capture haplotypes present in the population, but I believe there are two major shortcomings in the pan-genome analysis that I discuss below.

Answer: We thank the referee for their encouraging remarks.

First, unfortunately I don't believe this represents a fully realized pan-genome resource of bulbous barley, given that gene models are lifted over from one phased, annotated diploid assembly. The authors argue that there is an untapped reservoir of allelic variation present in bulbous barley that domesticated barley could benefit from; but if that is the case, then only genes present in that diploid genome can be explored and characterized in other genomes.

Answer: Reviewer #1 voiced a very similar concern about the lack of de novo gene annotations. We have outlined our strategy how to address this concern above and repeat our answer here for the convenience of Reviewer #3:

We agree with the reviewer that a limitation of our initial gene projection approach was that we were not able to detect completely novel genes and gene structures in some of the lines. Nevertheless, our gene projections are capable to call and resolve copy number variations (such as tandem copied genes) as well as diverged gene models due to the mapping strategy applied (different rounds of mapping including multiple and low stringency mappings considered).

That said, to resolve the referee's concern about the lack of novel genes, we re-annotated our assemblies from scratch. We developed and applied an integrated gene annotation approach which performs de novo gene calling for all genome assemblies (including individual genomes in phased assemblies and subgenomes in tetraploid accessions) in this study. This gene prediction strategy made use of (i) newly generated RNAseq data for diploid and tetraploid accessions from multiple tissues, (ii) publicly available expression data for tetraploid accessions (Fuerst et al. 2023, doi: 10.1093/gbe/evac168), and (iii) trained ab initio gene finders capable of calling novel genes in individual genomes. A final gene model consolidation step (cross-species all-against-all mappings) was added to the pipeline to minimize false positive CNV and PAV callings in the pangenome analyses.

Based on the updated gene predictions, we constructed an orthologous framework (using OrthoFinder) across the annotations of all individual genotypes and thereby defined the core, shell and cloud genomes of *H. bulbosum* at different ploidy levels. The orthologous framework can be considered as a reference-agnostic, gene-centric pangenome.

Analysis was supplemented with genes from barley, but gene comparative genomics and presence-absence variation will be undersampled and limited to coding sequences in the diploid, and a large amount of sub/neofunctionalization present in the tetraploids will be overlooked.

Answer: We agree with the reviewer that de novo gene annotations increase the chance of finding evidence for sub- and neofunctionalization of genes. At the same time, we have good reasons to doubt whether sub-/neofunctionalization has been pervasive in tetraploid *H. bulbosum*. Our analysis has shown that diploid and tetraploid lineages split in the last 1-2 million years and in the case of Greek tetraploids much more recently. Furthermore, our analysis supports the idea that there has been geneflow between ploidy levels. This would homogenize the genomes of diploid and tetraploid lineages and may make *H. bulbosum* a poor model to study the differential evolution of diploid and tetraploid genotypes.

What is more, the haplotypes of *H. bulbosum* recombine freely with each other thanks to polysomic inheritance and thus do not sort into subgenomes. In this scenario, Ward et al. 2004 (doi: 10.1016/j.tpb.2004.03.004) noted that the speed of subfunctionalization is low at high effective population sizes (which we observed in *H. bulbosum*).

To compare diploid and tetraploid genomes, we used the single-copy pangenomes to compare levels of presence-absence variation between diploid and autotetraploid genomes (Fig. 2i). We also found that tetraploid genomes have accumulated more deleterious variants than diploid ones (Extended Data Fig. 10e).

Furthermore, the authors use long-read sequencing and HiC to construct reference quality (based on LAI metrics) assemblies for multiple genomes, but break them up again using simulated reads to compare samples to a single reference.

Answer: The reviewer's concern about breaking up assemblies shared by reviewer #1. We have explained our rationale and steps to address the referees' concern in our response to reviewer #1. For the convenience of reviewer #3, we repeat our answer here:

We repeated our variant calling using simulated long reads (20 – 40 kb) and used the appropriate tools for variant calling. The quality information of read alignment is shown in the Supplementary Table 18. This is the relevant paragraph from the Methods section:

SNP, indels, and SVs were discovered in a panel of 33 sequence assemblies comprising the 32 assembled H. bulbosum haplotypes and the inbred genome of barley cv. Morex. Calling of SNP and indels was done with simulated long-read data using the FB19-011-3 haplotype 1 assembly as reference. For each haplotype/genome, 30x coverage and 20,000-40,000 length reads were simulated with Badread (v0.4.1). Read aligned to the haplotype 1 sequence assembly of FB19-011-3 with pbmm2 (<https://github.com/PacificBiosciences/pbmm2>). DeepVariant (v1.6.0) with the pretrained PacBio mode (--model_type PACBIO) was then used for variant calling of each accession, and all individual variants were merged using glxexus_cli (v1.4.1) with DeepVariant config file.

Note that we could not use the HiFi reads of *H. bulbosum* genotypes for variant calling without prior assembly because they did not come with phasing information. So it was necessary to assemble them into contigs first and then assign these to haplotypes with the help of Hi-C data.

While pan-genome analysis techniques are relatively new, there are more nuanced approaches (minigraph-cactus for example) than this that leverage multiple reference genomes.

Answer: We have included the results of graph-based approaches (including Minigraph-Cactus) in the revised manuscript. The results are shown in Extended Data Figures 6d-e and Supplementary Figures 16-23, and described in a new paragraph of the main text entitled “A haplotype-resolved pangenome of *H. bulbosum*”.

Secondly (and this supports the argument above), when reading about the amount of diversity present across the pan-genome, I was surprised how variants were reported. While the overall SNP rate is not provided among references, inspection of extended data figure 6 shows that in the distal regions of some chromosomes, there are approximately 150,000 to 180,000 SNPs per Mb, which I assume is the sum total of SNPs across all 32 haplotypes.

Answer: Also to address a concern of reviewer 1, we repeat the variant calling using long reads. We used Deepvariant to call the SNPs (see above). The final density of SNPs in the genome was between 60,000 and 100,000 per Mb. This is indeed the number of SNPs across all 32 haplotypes, including singletons.

Do these represent unique, repeat masked regions of the genome?

Answer: We did not do any repeat-masking before read alignment. We inspected the alignment statistics of the long read and found that only a few long reads had multiple alignments (mapping quality = 0, Supplementary Table 16). Therefore, we have good reason to believe in our current results are not strongly affected by the repetitive regions.

Are these extracted from the merged vcf, or counted from each individual alignment/SNP calls? Is this mainly driven by tetraploid samples?

Answer: The reported SNP counts are from a merged VCF obtained by multi-sample SNP calling with each haplotype considered a “sample” in DeepVariant. The table below shows the total number of pairwise differences between the reference FB19-011-3 haplotype and the other 31 haplotypes. Tetraploid accessions are indeed highly diverged from the reference, as is the diploid accession from Libya (PI365428). This is consistent with our Introblocker results.

ID	Variant Number (ref:FB19-011-3 haplotype 1)	Ploidy
FB190011hap4	21,497,996	Autotetraploid
FB190011hap1	21,783,637	Autotetraploid
FB190113hap2	22,177,139	diploid
FB190011hap3	24,161,293	Autotetraploid
FB190283hap3	24,866,941	Autotetraploid
FB190283hap4	25,688,854	Autotetraploid
FB190011hap2	26,332,198	Autotetraploid
FB200051hap1	26,470,480	diploid
FB200051hap2	26,704,334	diploid
FB190283hap2	27,850,621	Autotetraploid
FB190283hap1	28,570,185	Autotetraploid
FB200297hap2	30,336,659	diploid
FB200297hap1	30,456,455	diploid
A42hap3	31,119,000	Autotetraploid
GRA22561hap2	31,517,638	Autotetraploid
A42hap2	31,720,059	Autotetraploid
A40hap1	31,989,989	Autotetraploid
GRA22561hap1	32,084,040	Autotetraploid
A17hap2	32,110,838	Autotetraploid
A42hap4	32,115,277	Autotetraploid
A40hap4	32,253,246	Autotetraploid
A17hap3	32,509,107	Autotetraploid
A40hap2	32,999,512	Autotetraploid
A17hap1	33,026,341	Autotetraploid
PI365428hap2	33,182,386	diploid
GRA22561hap3	33,248,504	Autotetraploid
A17hap4	33,300,854	Autotetraploid
GRA22561hap4	33,352,270	Autotetraploid
A42hap1	33,523,284	Autotetraploid
PI365428hap1	33,575,402	diploid
A40hap3	33,592,099	Autotetraploid

It is difficult to evaluate what 180,000 SNPs per Mb means in relation to overall diversity. Assuming these are high-quality SNPs, counted from a merged vcf where repeat regions are masked (to avoid inflated counts), then there is a considerable amount of diversity present in the population (and this is ignoring small and large structural variants), which necessitates needing multiple de novo annotated references to capture it. Differences in ploidy across the population compounds the problem.

Answer: We agree that structural variants are abundant and that genome assemblies are needed to capture the full complement of diversity in *H. bulbosum*. This is illustrated by the results from SyRI, a tool for the discovery of SVs from genome alignments. SyRI reports a large number of highly diverged or not aligned regions. This is also evident in the high proportion of singletons sequence in pangenome graphs constructed with Minigraph-Cactus (Extended Data Figure 6e).

Haplotype genome	Syntenic regions	Not aligned (reference)	Highly diverged
A17 haplotype 1	1,496,422,894	1,592,582,033	245,333,554
A17 haplotype 2	1,429,648,590	1,605,956,748	296,737,447
A17 haplotype 3	1,476,257,669	1,597,402,835	251,522,093
A17 haplotype 4	1,502,528,179	1,563,413,537	274,861,327
A40 haplotype 1	1,422,358,352	1,632,995,317	281,705,329
A40 haplotype 2	1,456,293,232	1,559,049,818	309,910,557
A40 haplotype 3	1,489,623,425	1,534,165,582	302,126,424
A40 haplotype 4	1,400,564,433	1,553,395,443	318,044,959
A42 haplotype 1	1,475,261,314	1,572,636,743	265,202,061
A42 haplotype 2	1,398,823,055	1,650,744,434	258,650,000
A42 haplotype 3	1,422,582,511	1,640,807,383	285,531,616
A42 haplotype 4	1,453,829,246	1,591,694,804	291,712,589
FB19_001_1 haplotype 1	2,012,234,055	1,201,571,219	190,173,882
FB19_001_1 haplotype 2	2,100,494,757	1,045,901,051	204,113,127
FB19_001_1 haplotype 3	1,952,226,829	1,147,826,827	202,800,238
FB19_001_1 haplotype 4	1,959,087,692	1,255,224,559	216,866,539
FB19_011_3 haplotype 2	2,616,890,573	654,452,207	160,833,922
FB19_028_3 haplotype 1	1,832,738,955	1,264,132,603	233,973,662
FB19_028_3 haplotype 2	2,025,047,971	963,422,003	290,807,509
FB19_028_3 haplotype 3	1,999,029,659	1,090,405,306	277,448,872
FB19_028_3 haplotype 4	1,999,264,431	1,107,538,634	213,399,486
FB20_005_1 haplotype 1	2,241,927,891	938,128,456	194,964,445
FB20_005_1 haplotype 2	2,175,270,102	935,365,830	211,183,167
FB20_029_7 haplotype 1	2,149,959,720	1,025,715,261	183,819,379
FB20_029_7 haplotype 2	2,122,217,910	1,030,285,071	201,686,130
GRA2256_1 haplotype 1	1,411,113,212	1,670,654,442	246,613,573
GRA2256_1 haplotype 2	1,358,793,149	1,649,162,291	314,961,577
GRA2256_1 haplotype 3	1,446,728,268	1,519,772,491	324,291,662
GRA2256_1 haplotype 4	1,454,560,182	1,498,680,101	344,637,225
PI365428 haplotype 1	1,555,008,851	1,483,951,727	240,321,229
PI365428 haplotype 2	1,548,869,137	1,537,526,145	216,460,963
Average	1,722,117,943	1,358,534,223	253,248,211
MorexV3	178,119,041	2,817,909,759	400,923,425

To address this issue, we used the Minigraph-Cactus pipeline to discover SNPs and compared the results to the reference and long-read based DeepVariant pipeline. We found a decent overlap between both pipelines (see the figure below). Note that the minigraph-cactus pipeline uses “bubbles” (alternative paths) in the graph genome to represent structural variants. These bubbles can be large in size and resolving them to reference-based coordinates is not always possible, complicating the comparison between DeepVariant and Minigraph-Cactus results.

Additional comments:

-What is the evidence supporting autopolyploidy in *H. bulbosum*? Inspection of the kmer frequency histograms in supplemental figure 1 did not suggest the genotypes collected are autopolyploids (although evaluating this was a challenge considering the every subplot's title, axes and legends were in wingdings font and were completely illegible) Kmer histogram plots for autotetraploids typically show kmer peaks at 3-4X coverage because, as autopolyploids are typically derived from intra-chromosomal duplication, all chromosomes are very similar to each other, and thus share haplotypes. The kmer frequency subplots shown in figure 1 show two main peaks, which I would interpret as: heterozygous diploids or allotetraploids. While I can see small shoulder peaks in some of the panels (indicative of tetraploidy), the bulk of the kmers are observed within the 1X (unique haplotype) range. This seems at odds with autopolyploidy where high similarity among chromosomes enables random meiotic chromosome pairs to occur.

Answer: We have corrected the issues with Supplementary Figure 1.

There are several lines of evidence that support the autotetraploid nature of six of the sequenced accessions:

(i) We used a different tool, Genomescope to compute k-mer frequency histograms, which, owing to different smoothing method, show a different pattern that supports autotetraploidy (Supplementary Fig. 15).

(ii) The FISH image in Fig. 1e shows that the tetraploid clone FB19-028-3 has the expected 28 (4x7) chromosomes and FISH probes designed for four distinct haplotypes label that number of chromosomes.

(iii) We obtained flow-cytometric genome size estimates for three tetraploid individuals and added these to Supplementary Table 8.

(iv) The autopolyploid nature of *H. bulbosum* and its chromosome numbers have been known for a long time (Berg, K.H. von (1935) *Züchter* 8, 151-157; Lein, A. (1948) *Züchter* 19, 6-9; Jørgensen, R.B. (1982) *Nord. J. Bot.* 2, 421-434.)

-I found Figure 4 difficult to interpret and not necessarily supportive of claims made in the manuscript. First (and this is a straightforward change), but the colors used for Spain and Greece in panel B are so similar, it makes distinguishing between them difficult. Second, the authors should showcase how Figure 4C supports the hypothesis that autopolyploids arose twice, because it appears from the phylogenetic tree that polyploidy arose once and is shared between Group 1, Israel and Greece genotypes. Furthermore, it is also difficult to compare between panels B and C because a different number of labels/countries are used.

Answer: We have amended the color scheme in the revised version of Figure 4. Note that the selection of our diversity panel (PCA plots) has been moved to Figure 2.

We have aligned the tree with the ADMIXTURE results in panel b.

We agree that Fig. 4c did not make the most convincing argument for multiple origins of tetraploid accessions. Trees (which implicitly assume the absence of gene flow) are only an imperfect representation of the relationship between populations of the same species. That said, we do see evidence for at least two origins of tetraploid cytotypes in the fact that three tetraploid samples sit on a branch otherwise occupied by diploid samples.

We note that Fig. 4c is not our solely line of evidence for multiple origins. We buttress this conclusion with other types of analyses whose outcomes are shown in Figs. 4b-f and Extended Data Fig. 8. Even from transposons, the tetraploid genome from Greece is similar to the diploid genome from that country in terms of TE profile in Extended Fig. 7b.

Minor comments

L74- Giving the assembly size here lacks context. Unless the reader is already familiar with bulbous barley, they would not understand if this represents a substantial capture of the total genome size.

Answer: We have rephrased this sentence:

[...] the assembled contigs amounted to 6.78 Gb of sequence, capturing a substantial proportion of the diploid genome, and the contig N50 was 9.82 Mb.

L75-76- If comparing to other genomes, give the stats so the reader does not have to look them up.

Answer: The revised sentence reads:

This level of contiguity is comparable to those achieved in barley (16.8 Mb), sugar cane (12.6 Mb) and potato (2.1 Mb and 9.1 Mb).

Chromosome 4H- IBD section that spans basically the entire chromosome- how did you map short-read HiC to a region that is completely duplicated?

Answer: Both haplotypes are represented by a single set of contigs in this region. The contigs have double the average coverage, so we can infer that are present in two haplotypes (see Extended Data Fig. 4a). When making haplotype-specific Hi-C maps, we included these contigs in the map construction process for both haplotypes.

L97- 50% SNPs? This needs to be better described because it is confusing until you read the methods later in the manuscript.

Answer: We have changed this sentence to

A total of 60,934 neighbouring SNP pairs (1.01%) had recombination fractions exceeding 50 %, indicating that they were potentially located in wrongly phased regions.

1.1% - what is the total number of SNPs? Are these clustered at a particular position? or are they interspersed across the genome? Interspersed is fine, but clustered could represent a mis-assembly. Would be nice to see these SNPs paired with chromosome coverage along each chromosome.

Answer: We have added Supplementary Figure 4 to show the locations of the potential phase switches. They present on all chromosomes. We note that many of the SNP pairs with high recombination fraction may not owing to errors in assembly, but in calling SNP from short reads.

L112- report this data per haplotype as well.

Answer: Owing to the reorganization of the manuscript, the gene annotation is described as part of the pangenome analysis and we do not report annotation statistics for individual genomes. We report the range of the number of annotated genes in this sentence:

The number of gene models annotated per genotypes identified ranges from 107,037 for diploid PI365428 to 238,246 for tetraploid FB19_028_3.

Gene annotation statistics for all haplotypes are given in Supplementary Table 10.

L119- These values are not not similar. An LAI value > 20 is considered high-quality while 14.7 is simply adequate for a reference genome.

Answer: The revised sentence reads:

The long terminal repeat (LTR) Assembly Index (LAI), which evaluates the contiguity of intergenic and repetitive regions of genome assemblies based on the intactness of LTR retrotransposons, of the FB19-011-3 genome assembly was 20.52, which is adequate for a high-quality genome assembly

We note that the LAI values of recent telomere-to-telomere assemblies of melon barely exceed 10 (Table 1 of Zhang et al. Nature Genetic 2024, doi:10.1038/s41588-024-01823-6). A possible reason for differences in LAI values between species, apart from assembly quality, could be differences in the timing and extent of the most recent amplifications of transposable elements.

L192-196: this should be scaled in terms of repeats per Mb rather than totals.

Answer: Our intention was to illustrate the large impact of individuals families on genome size. This is why we consider the cumulative length of all elements of a family a better metric.

The average densities per Mb are

BARE1: 117 kb/Mb (*H. vulgare* Morex), 82 kb/Mb (*H. bulbosum* FB19-011-3 haplotype 1)
Sabrina: 99 kb per Mb (*H. vulgare* Morex), 136 kb/Mb (*H. bulbosum* FB19-011-3 haplotype 1)

In the methods section, coverages of HiC and HiFi and pollen libraries should be reported, they are directly relevant for determining the quality of the assembly.

Answer: We have added read counts and coverage values to Supplementary Tables 1 and 6.

tetraploid genome assembly- how exactly did the multi-count contigs get placed?

Answer: We have added a schematic diagram in Supplementary Figure 38. We calculated the PCA central point (black star in the figure below) of each haplotype in single coverage contigs. If a contig's (point) coverage is double, we calculated the Euclidean distance (line between red point and black star) between the contig and the central points of the different haplotypes and assign it to the two nearest haplotypes. In the same way, when the coverage is triple, it is assigned to the three nearest haplotypes.

GENESPACE software used to compare gene synteny among references does not appear to be properly referenced

Answer: We added the proper reference for GENESPACE (ref. # 71 in Online Method):

Lovell, J. T. *et al.* GENESPACE tracks regions of interest and gene copy number variation across multiple genomes. *eLife* **11**, e78526 (2022). <https://doi.org:10.7554/eLife.78526>

Editor

We therefore invite you to revise and resubmit your manuscript, taking into account the points raised by the reviewers. Please highlight all changes in the manuscript text file.

Note that we do not agree with reviewer #1's insistence that a full graph pangenome analysis is necessary here (and we would also ask you to disregard their tone in this report). We are generally satisfied with the level of advance provided here, but feel that it is important that the final comments - especially those from reviewer #3 - are addressed before we make a final decision.

Answer: We address the pangenomic aspects of our study in our response to Reviewer #1 and have also fully responded to the concerns raised by Reviewer #3.

Regarding Reviewer #1's assertion that our study does not constitute a pangenome analysis, we reiterate that we constructed pangenome graphs using multiple complementary approaches, including: (i) a single-copy pangenome, (ii) gene-based partitioning into core, shell, and cloud compartments, and (iii) whole-genome pangenome graphs. These analyses are presented in Figure 2, Extended Data Figure 7, and Supplementary Figures 16–23, 35, and 36. We appreciate the editor's perspective that a full graph-based pangenome is not a necessity for our study and remain confident that our work provides meaningful insights into genome structure and diversity.

You will also need to make some editorial changes to your paper so that it is as brief as possible and complies with our Guide to Authors.

Answer: We have shortened the main text to ~4,300 words and moved several panels from the main figures to the Extended Data items. There are now 5 main figures and 11 Extended Data figures. We have also resized the panels of the main figures to use the available space more efficiently.

Referee #1:

The authors have made some small attempts to create a pan genome, they did run minigraph-cactus but then said that interpretation is complex. That is true, as these analyses are complex, however but it looks like they didn't try to optimise. At present none of the analysis in the manuscript relates to the pangenome so the title and abstract still remain misleading. Making and analysing graph pangenomes is challenging and the authors should not be claiming they have done so when it looks like they just tried putting the data through the software without optimisation/cleaning, then claimed it was too complex and did other analysis of the haplotypes.

Answer: The reviewer claims that our use of Minigraph-Cactus and PGGB was insufficient and that we did not optimize our analyses. However, as stated in our original response, we acknowledge the complexity of interpreting graph-based pangenomes and have approached this analysis systematically. Rather than discarding the results due to complexity, we leveraged

these graphs for structural variation analysis, comparisons between diploid and tetraploid genomes, and single-nucleus pollen sequencing.

We reiterate that our pangenome analysis does not merely involve collecting genome assemblies but includes structured comparative analyses that align with recent pangenomic studies in human and crop plants.

This is still not pan genome analysis and trying to change the definition to fit their data is wrong.

Answer: We acknowledge that in our own recent reviews on pangenomes (Schreiber et al. 2024 <https://doi.org/10.1038/s41576-024-00691-4> and Jayakodi et al. 2025 <https://doi.org/10.1146/annurev-arplant-090823-015358>), we defined them as collections of genome sequences without requiring a graph-based representation. This definition is widely used, including in human genomics. For example, the United States National Human Genome Research Institute states (<https://www.genome.gov/genetics-glossary/Pangenome>):

A pangenome is a collection of genome sequences from many individuals of the same species. Scientists generate pangenomes to capture the breadth of genomic variation across populations and use it as a “reference” to compare other genomes to. Researchers can detect genomic variants by comparing a new individual’s genome to a reference genome.

Matthews et al. 2025 (<https://doi.org/10.1093/bib/bbae588>) note that

[t]he term ‘pangenome’ is currently used to describe multiple different types of genomic information, and limited language is available to differentiate between them.

We are not aware of any strict definition of the term pangenome that mandates a graph-based representation. Rather, pangenomes have been described and analyzed using a variety of approaches, including those employed in our study.

The number of genomes assembled is still low even when looking at assembled haplotypes and especially when these are not fully analysed as pan genomes. This is not addressed by stating they have resolved multiple haplotypes as the number is still low.

Answer: The reviewer maintains that our dataset is too small for a pangenome study, despite our previous clarification that our dataset consists of 10 genome assemblies comprising 32 haplotypes. We emphasized that these haplotypes exhibit greater diversity than a similar number of inbred barley or wild barley genomes (Fig. 2g). Additionally, our selection of accessions was based on genotyping-by-sequencing data from over 200 individuals, ensuring that our assembled genomes capture key diversity across the species' distribution. To further illustrate the representativeness of our dataset, we provided new PCA plots and a geographic diversity map in Figure 2. These additions address the concern that our sample size is too limited for meaningful analysis.

There is still very little biology presented, this is coming across very much as a resource of some genomes rather than having any biological relevance. the authors do not argue well the

case for the pan genome for the virus resistance, there is very little information provided and this is the only real biology presented, which is very poor for a pan genome paper, however, as already stated this is not a pan genome paper. The authors have just moved previous analysis round in the manuscript and the interval for the BYDV is the same, showing no new analysis has been done because there was no real pan genome construction.

Answer: The reviewer states that our study provides little biological insight beyond being a resource. However, we have demonstrated the biological relevance of our work in multiple ways:

1. Genomic Basis of Introgression Breeding: We identified candidate genes for a sublethal factor linked to virus resistance, which is critical for understanding barriers to introgression breeding. This could not have been achieved without a comparative genomic approach using multiple assembled genomes.
2. Diversity in Resistance Genes: Our analysis of nucleotide-binding leucine-rich repeat (NLR) genes demonstrated that these genes predominantly reside in recombinogenic regions, making them accessible for breeding applications.
3. Structural Variation Analysis: We revealed extensive structural variation at the Ryd4 locus, shedding light on resistance gene evolution and highlighting the importance of haplotype-resolved assemblies.

These findings extend beyond being a genomic resource and provide novel biological insights into breeding and adaptation.

The authors have selectively referred to genomes for the comparison which are still not appropriate - they can clearly find things much more related to bulbosa than sugarcane and potato than barley. they are clearly trying to find assemblies more close to theirs in stats rather than what is realistic.

Answer: Reviewer #1 criticizes our comparisons to sugarcane and potato assemblies. However, as we previously explained, our choice of comparative references is based on achieving a fair and meaningful benchmarking of assembly metrics, including N50 and completeness. Although we also reference barley as a close relative, we included sugarcane and potato due to their complex genome structures, which provide additional context for evaluating our assemblies.

The main issue is that this is still not a pan genome, the authors have made a poor attempt with no optimisation of producing a pan genome and it seems to be a more resource than having a significant biological story

Answer: We acknowledge that pangenome analysis is an evolving field and that various methodologies exist. Our study embraces an inclusive definition of a pangenome and employs rigorous comparative genomic analyses to investigate structural variation, gene content, and introgression breeding applications. We respectfully disagree with Reviewer #1's assertion that our study does not constitute a pangenome analysis and maintain that our methodology aligns with established standards in the field.

Referee #2:

This study presents a pangenome of bulbous barley comprising ten annotated, haplotype-resolved genome sequence assemblies of diploid and autotetraploid accessions. They show that the differential amplification of transposable elements after barley and *H. bulbosum* diverged from each other is responsible for genome size differences between them and that autotetraploids arose twice. Additionally, the authors demonstrate the utility of the pangenome in introgression breeding of barley with this wild relative species.

Having reviewed all the reviewer comments and the responses of the authors to those, I am satisfied that the authors have made significant additions to the study's methodology, analysis and reported outcomes, including improvements to the manuscript's text and figures.

The work reported is now much clearer and robust and as such I recommend its acceptance for publication.

Answer: We thank the reviewer for their encouraging comments.

Referee #3 (Remarks to the Author):

First, I want to commend the authors for their revisions that have greatly improved the manuscript. Their improved annotation strategy and analysis has elevated this work to a proper 'pan-genome' analysis of *H. bulbosum*. The analyses are well-thoughtout and considerate. I find little to critique in terms of technical analyses, with good care to show that the diploid genome assembly was properly phased.

Answer: We thank the reviewer for their encouraging comments.

My only gripe in terms of technical analysis and interpretation relates to the characterization of crop-wild introgressions. Authors completed a competitive mapping analysis of GBS data to a synthetic diploid of MorexV3 and *H. bulbosum* (genotype FB19-011-3). Regions where read-mapping skewed toward wild *bulbosum* were inferred to have been introgressed. While that is certainly possible, there is not enough information given (map quality/ CIGAR string parsing/etc..) to assess the validity of that claim (Also, Line 1122 refers to Supp Figure 13, but I think the authors mean Supp table 13). While introgression of *bulbosum* sequence into the genome could be found using this competitive mapping strategy, additional information is needed to understand how close the sequences in the GBS data match wild barley (kmer masking/ read mapping metrics, etc..). As written, the analysis simply shows these regions map better to wild barley than morex, but could still be quite diverged and derived from another donor.

Answer: We acknowledge that our approach is not designed to identify the precise source of introgressions if the donor species is unknown. However, we do not believe this is a significant limitation for two key reasons:

- 1. Phylogenetic distinction of *H. vulgare* and *H. bulbosum*

H. vulgare and *H. bulbosum* occupy a distinct branch of the *Hordeum* phylogenetic tree and are equidistant from all other *Hordeum* species:

[FIGURE REDACTED]

Figure: **[TEXT REDACTED]**

If an introgression had originated from another *Hordeum* species, reads from that region would be expected to map equally well to both genomes, rather than showing the observed bias toward *H. bulbosum*. Furthermore, there are no known reports of successful hybridization with fertile offspring between *H. vulgare* and any *Hordeum* species other than *H. bulbosum* (von Bothmer et al. 1983, <https://doi.org/10.1111/j.1601-5223.1983.tb00895.x>).

The only other conceivable "wild" donor would be *H. vulgare* subsp. *spontaneum*, which, being conspecific with domesticated barley, would be expected to map preferentially to the MorexV3 genome, not the *H. bulbosum* genome. This further supports our inference that the introgressions originate from *H. bulbosum*.

2. Well-defined provenance of the introgression lines (ILs)

The ILs analyzed in our study were developed by Richard Pickering's lab through controlled crosses between elite barley varieties and defined *H. bulbosum* clones, followed by selection of recombinant progeny (Pickering et al. 2000, <https://doi.org/10.1007/PL00002904>). These introgressions were previously characterized using GBS data by Wendler et al. 2015 (<https://doi.org/10.1016/j.molp.2015.05.004>), and in our study, we reanalyze the same dataset to (i) confirm and refine the positions of known introgressions and (ii) determine haplotypes of origin using our haplotype-resolved assemblies.

To validate our mapping strategy, we also analyzed the GBS data from *H. vulgare* cultivars and *H. bulbosum* clones that served as donors of introgression lines:

Figure: Competitive mapping of GBS reads of *H. vulgare* and *H. bulbosum* genotypes to a concatenated genome sequence.

Additionally, we provide more details on our filtering criteria for read mapping in *Analysis of introgression lines* of the Methods section to ensure full transparency in how the analysis was conducted.

*GBS data of introgression lines (IL) of Wendler et al. were aligned to a synthetic diploid genome combining barley MorexV3 and FB19-011-3 haplotype 1. Alignment of GBS data was conducted with BWA-MEM using a hybrid Hordeum bulbosum (FB19-011-3 haplotype 1) and Hordeum vulgare (MorexV3) genome. Average read depths were calculated in 1 Mb windows along the genome with SAMtools, excluding secondary and non-unique alignments with the command `samtools view -q20 -F3332`. Then we normalized the depth of samples: Normalized 1Mb read depth=(Aligned read number within 1Mb)/(Total aligned read number) $\times 10^6$. Windows of the *H. bulbosum* haplotype with more than 200 (normalized 1 Mb read depth) were considered as introgressed regions (Supplementary Table 13).*

We also uploaded the mapping results of all ILs to github (https://github.com/jia-wu-feng/Pan_Bulbosum/tree/main/06_Introgression_lines).

When considering this manuscript, I think it is important to contrast this work with the other wild barley pan-genome that was recently published in Nature (Jayakodi et al. 2024) by a number of the same authors. While this work has generated new genome assemblies and a pan-genome for a different wild species, there is some overlap between the two papers, with a number of similar analyses being completed. This is unavoidable considering the nature of pan-genome analysis, but the title: "Structural variation in the pangenome of wild and domesticated barley" lends itself to stricter scrutiny. I found the previous manuscript that constructs a pan-genome using the wild progenitor of barley (*H. vulgare* subsp. *spontaneum*), quite comprehensive and thus wonder if there are enough impactful findings that the readers of Nature would be interested in another wild barley pan-genome months later, particularly with such a heavy focus on the genome assembly itself (phasing/TE content, etc..)

Answer: We would first like to clarify the term “wild barley”, as it can refer to two distinct concepts:

1. The wild progenitor of domesticated barley (*Hordeum vulgare* subsp. *spontaneum*), often colloquially referred to as *Hordeum spontaneum*. These populations are fully interfertile with domesticated barley.

2. Barley wild relatives in a broader sense, which includes ~30 species in the *Hordeum* genus. These are distinct species that are reproductively isolated from *Hordeum vulgare* and fertile crosses with barley have been obtained only with *H. bulbosum*

Our study focuses on *Hordeum bulbosum*, a wild *Hordeum* species distinct from *H. vulgare* and its progenitor *H. spontaneum*. While there is some methodological overlap with the recent *H. vulgare* pangenome paper (Jayakodi et al. 2024), our study differs significantly in scope, biological insights, and key findings.

1. Emphasis on genome assembly and phasing
A major focus of our study is the haplotype-resolved assembly of *H. bulbosum*, which remains a technical challenge. We implemented innovative validation approaches, including FISH and single-nucleus pollen sequencing, which have not been applied in *H. vulgare*. These phased assemblies were critical to several of our biological conclusions.
2. Origins of autotetraploid *H. bulbosum*
Unlike *H. vulgare*, *H. bulbosum* includes diploid and autotetraploid cytotypes. Our study provides a detailed investigation into the origins of tetraploid cytotypes, which has no equivalent in the *H. vulgare* pangenome study. This analysis relied on haplotype-resolved assemblies to track genome duplication and divergence.
3. Comparative evolutionary analysis of transposable elements (TEs)
A key limitation noted by reviewers of the *H. vulgare* pangenome (see peer review reports: https://static-content.springer.com/esm/art%3A10.1038%2Fs41586-024-08187-1/MediaObjects/41586_2024_8187_MOESM3_ESM.pdf) was the lack of an evolutionary perspective on TEs. This was because TE variation within *H. vulgare* is relatively recent and does not correlate well with population structure. By contrast, *H. bulbosum* diverged from *H. vulgare* ~4.5 million years ago, allowing us to explore evolutionary trends in TE dynamics across a greater timescale, providing insights into their role in genome evolution.
4. Loci of agronomic importance: distinct focus
While both studies analyze loci of agronomic importance, they focus on different aspects. The *H. vulgare* pangenome paper examined loci involved in resistance, inflorescence development, and grain properties, often focusing on recently evolved diversity (e.g., a gene duplication in a fertility regulator that arose post-domestication). Our *H. bulbosum* pangenome focuses on the *ryd4* resistance locus, where we uncovered deeply diverged haplotypes and greater haplotype diversity than in *H. spontaneum*, highlighting the untapped genetic potential of *H. bulbosum* for barley breeding.

While both studies involve pangenome construction, they have distinct biological questions, evolutionary timescales, and applications. Our work on *H. bulbosum* provides novel insights into genome evolution, polyploidy, and introgression breeding that were not possible in the *H. vulgare* study. Given these differences, we believe our study makes a meaningful and timely contribution to the field.

To acknowledge the relation between both studies, we have rephrased the last sentence of the introduction:

*Complementing the pangenome of domesticated barley and its wild progenitor *H. vulgare* subsp. *spontaneum*, we report here a haplotype-resolved pangenome of the barley wild relative *H. bulbosum* and illustrate its applications in evolutionary research and trait mapping.*